# Classification of magnetohydrodynamic transport
# at strong magnetic field

B. Benenowski[1], N. Poovuttikul[2]*,

**1** Instituut-Lorentz for Theoretical Physics, ΔITP, Leiden University, Niels Bohrweg 2,
Leiden 2333CA, The Netherlands
**2** University of Iceland, Science Institute, Dunhaga 3, IS-107, Reykjavik, Iceland
* nickpoovuttikul@hi.is

## Abstract

**Magnetohydrodynamics is a theory of long-lived, gapless excitations in plasmas. It was argued from the point of view of fluid with higher-form symmetry that magnetohydrodynamics remains a consistent, non-dissipative theory even in the limit where temperature is negligible compared to the magnetic field. In this limit, leading-order corrections to the ideal magnetohydrodynamics arise at the second order in the gradient expansion of relevant fields, not at the first order as in the standard hydrodynamic theory of dissipative fluids and plasmas. In this paper, we classify the non-dissipative second-order transport by constructing the appropriate non-linear effective action. We find that the theory has eleven independent charge and parity invariant transport coefficients for which we derive a set of Kubo formulae. The relation between hydrodynamics with higher-form symmetry and the theory of force-free electrodynamics, which has recently been shown to correspond to the zero-temperature limit of the ideal magnetohydrodynamics, as well as simple astrophysical applications are also discussed.**

---

[1]benenowski@lorentz.leidenuniv.nl
[2]nickpoovuttikul@hi.is

# 1 Introduction

Hydrodynamics is a theory of gases, fluids and other collective systems at long time scales and long distances [1, 2]. The framework in which it is formulated is that of a gradient expansion written in terms of local hydrodynamic fields, resulting in an infinite series of zeroth-order hydrodynamics, first-order hydrodynamics, second-order hydrodynamics [3, 4], and higher orders [5–7].

In the absence of dissipation, one can formally attempt to write a hydrodynamic theory in the language of the standard action and use the variational principle to derive the dynamical equations of motion. Such approaches have been successfully implemented in the study of equilibrium fluids [8, 9] and can even be used out-of-equilibrium to compute, for example, the thermodynamical transport coefficients, which generate no entropy [10, 11]. Beyond equilibrium, however, standard field theory necessarily fails as it is unable to correctly account for dissipative effects, which generate entropy. What has transpired in recent years, however, is that hydrodynamics can be consistently formulated as an effective dissipative field theory by using the language of the Schwinger-Keldysh formalism [12–21].

The question of whether a non-dissipative fluid could in principle exist arose with the work of [11], which analysed constraints on conformal second-order transport imposed by the absence of dissipation (or entropy production). A natural example of such a system is the holographic fluid dual to the Einstein-Gauss-Bonnet theory. In this theory, it is known that a "formal" limit exists that takes the shear viscosity to zero, $\eta \to 0$ [22]. Since the entropy production in a conformal fluid is dominated by a single first-order term proportional to $\eta$, this implies that in such a limit, only subleading effects, if any, could generate entropy. The fact that even in the absence of first-order effects, second-order hydrodynamics can indeed continue to generate entropy was observed through a detailed, non-perturbative analysis of second-order transport coefficients in this theory in Refs. [23–25]. These investigations pointed to the fact that a genuine nondissipative fluid requires additional structure in order for it to be realisable. In this work, we propose that such scenario could exist in the context of plasma physics at extremely strong magnetic field regime.

Plasma is an ionised gas described by the theory of magnetohydrodynamics (MHD) [26–28]. In the standard language, MHD is a collective theory of coupled hydrodynamic and electromagnetic degrees of freedom. In the work of Ref. [29, 30], MHD was recently reformulated and generalised to describe any plasma by using the language of higher-form (or generalised global) symmetries [31]. The theory uses the fact that beyond conserved energy and momentum, the conserved number of magnetic flux lines crossing a two-dimensional spatial surface gives rise to a conserved two-form current, or $\nabla_\mu J^{\mu\nu} = 0$, where $J^{\mu\nu}$ is an antisymmetric tensor. This two-form current is the Noether current of a $U(1)$ one-form symmetry, that is ensured by the absence of magnetic monopoles, and is treated in the same footing as energy-momentum tensor in the hydrodynamic gradient expansion scheme. The conservation for the electric flux, on the other hand, is explicitly broken by the presence of ionised medium and is irrelevant in the hydrodynamic setup[3]. The connection between this formalism and the one where the derivative expansion procedure is applied to the *gauged* ordinary $U(1)$ current $j^\mu \sim \nabla_\nu F^{\nu\mu}$ in [33] can be found in [34, 35].

Unlike in an ordinary relativistic fluid, ref. [30] argued that their formulation allowed one to take the zero temperature limit $(T \to 0)$ of MHD and end up with a consistent,

---

[3]In other words, we assume that the life-time of the electric field excitation is much shorter than the characteristic time scale set by temperature and magnetic field. For higher-form symmetry formalism where the conservation of electric flux is only slightly broken, or equivalently when the mentioned life-time is comparable to the macroscopic time scale, see e.g. [32]

hydrodynamic theory of a dissipationless plasma. Since the only dimentionless parameter is $B/T^2$, where $B$ is the strength of the magnetic field, one can equivalently think of this limit as the limit of an extremely strong magnetic field, $B \to \infty$. In this limit, the theory enjoys enhanced spacetime symmetry, which is manifest in emergent boost-invariance along the magnetic field lines. This additional symmetry, along with other symmetries of the plasma allow one to write down hydrodynamic constitutive relations, which permit no first-order gradient terms and no (vector) entropy current. Transport beyond the ideal limit is dominated by second-order hydrodynamics. In effect, Ref. [30] predicted that all first-order transport coefficients in a plasma necessarily have to vanish as $T^2/B \to 0$. This prediction was verified in a dual holographic model by [36].

Motivated by the question of better understanding this enhanced symmetry limit of the cold plasmas, in this paper, we extend the work of [30] and construct a fully non-linear theory of a zero temperature plasma. We do this by writing down a dissipationless effective action, which automatically ensures that the system is closed and produces no entropy. Beyond the verification of the linearised sector of the theory from [30] and the constraints that appeared there, here we will obtain the full set of (non-linear) second-order transport coefficients as well as the relevant Kubo formulae. Due to the complexity of the combinatorics involved in constructing the relevant set of tensors, we will implement a computer algebra algorithm — an extensions of the one used to construct third-order hydrodynamics in [5] — which employs the xAct library in Wolfram Mathematica [37].

The second motivation for this work is an interesting connection between the $T = 0$ theory of magnetohydrodynamics and Force-free Electrodynamics (FFE) known mainly from astrophysics [38–40]. Beyond the observation that both theories posses the same global symmetries, the fact that the equations of motion of FFE are identical to the zero-temperature limit of magnetohydrodynamics with a one-form global $U(1)$ symmetry was recently shown in Refs. [41, 42].

FFE has been widely used to describe the phenomenology of the magnetospheres of compact astrophysical objects such as neutron stars, Kerr black holes [43], active galactic nuclei and, more recently, binary black holes [44]. In these scenarios, one can think of the magnetosphere as consisting of an electromagnetic field coupled to plasma. The plasma is, on the one hand, dilute enough so that its contribution to the equation of state is negligible. For example, its energy density is about fifteen order of magnitude lower than the one of the electromagnetic field in the pulsar's magnetosphere (see e.g. [45] for a review of the astrophysical setup). At the same time, the plasma density is high enough to screen the electric field. In the language of global symmetries, this means that the conservation of the electric flux is explicitly broken so that the only conserved charges in the IR dynamics are energy, momentum and magnetic flux as in the above hydrodynamic setup. The equations of motion for FFE, however, are not written in terms of the conservation laws but in the following way:

$$\nabla_\mu \left( \epsilon^{\mu\nu\rho\sigma} F_{\rho\sigma} \right) = 0, \qquad j^\mu F_{\mu\nu} = \left( \nabla_\lambda F^{\lambda\mu} \right) F_{\mu\nu} = 0, \qquad \epsilon^{\mu\nu\rho\sigma} F_{\mu\nu} F_{\rho\sigma} = 0. \qquad (1.1)$$

The first equation is the conservation of the magnetic flux, which upon the standard identification of the two-form current associated with a one-form symmetry, i.e. $J^{\mu\nu} = \frac{1}{2}\varepsilon^{\mu\nu\rho\sigma} F_{\rho\sigma}$, becomes the conservation equation $\nabla_\mu J^{\mu\nu} = 0$ used in the construction of MHD by [30]. The second equation is the *force-free condition* indicating that the force $j^\mu F_{\mu\nu}$ exerted on the plasma by the electromagnetic field vanishes, with $j^\mu \propto \nabla_\nu F^{\nu\mu}$ being the gauged $U(1)$ current. The last equation, called *the degeneracy condition*, indicates that a probe charge cannot be accelerated along the magnetic field lines in the magnetosphere (since $\mathbf{E} \cdot \mathbf{B} = 0$). The degeneracy condition together with the condition that the magnetic field dominates, $F^{\mu\nu} F_{\mu\nu} = -|\mathbf{E}|^2 + |\mathbf{B}|^2 > 0$, allows one to write $F_{\mu\nu} = \partial_{[\mu}\sigma^1 \partial_{\nu]}\sigma^2$, where

$\{\sigma^1, \sigma^2\}$ are the coordinates orthogonal to the "worldsheet" of the magnetic field lines. This treatment of the magnetic field lines as of strings was implemented in the context of FFE by [39]. A more formal geometric approach to this formalism, as well as various astrophysical applications of it, can be found in a review by Gralla and Jacobson [46]. Note also that FFE description neither depends on the microscopic details of the charge sector $j^\mu$ nor on how it is coupled to the electromagnetic sector. This already hints at the connection between FFE and hydrodynamics as they are both macroscopic effective theories that are independent of the microscopic details.

What is apparent from the above discussion is that both the extreme limit of MHD and FFE work at negligible temperatures, have the same conserved charges and are independent of the microscopic details. However, the formulation with higher-form symmetries provides us with several advantages. Firstly, it allows us to systematically couple a plasma to the external background field $b_{\mu\nu}$ (which parametrises the external charge injected into the system). The other potential improvement comes from the fact that FFE has a built-in assumption: the degeneracy of the magnetic field lines parametrised by $\mathcal{P} \sim \epsilon^{\mu\nu\rho\sigma} F_{\mu\nu} F_{\rho\sigma}$, vanishes. While this makes the system of equations in (1.1) well-behaved and relatively easy to solve, it fails to describe many of the phenomena that happen in the magnetospheres of compact astrophysical objects. In particular, the inability to accelerate charged particles implies that the magnetosphere in FFE description cannot lose its energy in terms of photons. This contradicts the fact that we do observe radio-wave emissions from pulsars (see e.g. [47]). In addition, it also means that FFE cannot account for the observed phenomena such as jets and cosmic ray bursts. In the astrophysics literature, the condition $\mathcal{P} > 0$ is lifted by phenomenologically introducing resistivity to the system by various approximations, resulting in multitudes of models, see e.g. [45] for discussion on origin of the emission and [48] for a review of various models of this type. On the other hand, the systematic gradient expansion of conserved currents in hydrodynamics with higher-form global symmetries allows the possibility of having non-zero $\mathcal{P}$ from derivative corrections, namely $\mathcal{P} = \mathcal{P}(\partial, \partial^2, ...)$. This possibility was pointed out in [42]. Additionally, as already discussed above, if one constrains the temperature of the system to be low compared to the scale of interest, it was argued in [30] that the first derivative corrections must vanish as well. With the classification of second-order transport, we can systematically single out terms which are responsible for the charge acceleration along the field lines. Together with the Kubo formulae, the transport coefficients (analogous to the viscosity) can then be obtained from microscopic theory. Given that FFE also makes appearance in various situation other than compact astrophysical objects (such as solar corona [49] and topological insulators [41]), we hope that the classification presented here will provide a systematic way to analyse force-free electrodynamics and its connection to the underlying microscopic theory.

The remaining sections of the paper are organised as follows. We start by briefly reviewing the construction of the hydrodynamic effective action and discuss how to organise the derivative expansion in section 2. We explain the relevant hydrodynamic variables which are analogous to the fluid velocity and the chemical potential in an ordinary fluid as well as how to organise them into the effective action in section 2.1. And we outline the procedure and algorithm we use to classify all the possible terms in the effective action with two derivatives in section 2.2. There are eleven possible terms in the effective action that contribute to the conserved currents $T^{\mu\nu}, J^{\mu\nu}$. This is our main result and it is presented in the same section. We then study how these new second-order transport coefficients affect the correlation functions in section 3. In subsections 3.1 and 3.2, we study the long-lived modes analogous to Alfvén and magnetosonic waves in the strong magnetic field limit and identify the correlation functions which encode the corresponding sound poles. The Kubo

formulae, which relate the transport coefficients controlling the second-derivative terms to the two-point and three-point functions, are presented in section 3.3. A short discussion on the applications of this formalism, including the transport coefficient responsible for the aceeleration along the field lines of a simple model of the magnetosphere is presented in section 4. We conclude our work and discuss some immediate open problems in section 5. Four appendices containing useful formulae and computational details are also provided.

## 2 Effective action

Effective action is an organised way to construct hydrodynamics given the global symmetries. In this work, where we consider a theory of a conserved energy and momentum $T^{\mu\nu}$ as well as a conserved two-form current $J^{\mu\nu}$, the generating function is obtained by coupling the theory to a background metrics $g_{s\mu\nu}$ and the two-form background gauge fields $b_{s\mu\nu}$ in the Schwinger-Keldysh formalism,

$$Z[g_{s\mu\nu}, b_{s\mu\nu}] = \left\langle \exp\left[ i \sum_{s=1,2} (-1)^{s+1} \int d^4x \sqrt{-g_s} \left( \frac{1}{2} T_s^{\mu\nu} g_{s\mu\nu} + J_s^{\mu\nu} b_{s\mu\nu} \right) \right] \right\rangle_{SK} . \quad (2.1)$$

Here the label $s = 1, 2$ denotes the source which couples to two sets of degrees of freedom, one evolving forwards on the complex time contour while the other one evolving backwards. This generating function is the result of integrating out the soft degrees of freedom from the effective action $\tilde{W}$, namely

$$Z[g_{s\mu\nu}, b_{s\mu\nu}] = \int_{SK} \mathcal{D}[\Phi_s] \exp\left( i\tilde{W}[g_{s\mu\nu}, b_{s\mu\nu}, \Phi_s] \right) , \quad (2.2)$$

where $\Phi_s$ denotes two sets of soft *hydrodynamic* degrees of freedom. In the *classical limit*, where one can ignore the statistical fluctuations (such as in large $N$ theories), the path integration can be performed with the saddle point approximation. The coupling between $\Phi_1$ and $\Phi_2$ results in dissipative effects (such as viscosity) and in their absence one can split $W$ into two pieces that only depend on $s = 1$ and $s = 2$ fields, respectively,

$$\tilde{W} = W[g_{1\mu\nu}, b_{1\mu\nu}, \Phi_1] - W[g_{2\mu\nu}, b_{2\mu\nu}, \Phi_2] . \quad (2.3)$$

We will argue that the action for the theory with strong dynamical magnetic field can be written in the above form and this will be justified in the next section.

The variables $\{g_{s\mu\nu}, b_{s\mu\nu}, \Phi_s\}$ are to be combined into objects which are invariant under diffeomorphisms and gauge transformations of the background fields,

$$g_{s\mu\nu} \to g_{s\mu\nu} + 2\nabla_{(\mu}\xi_{s\nu)}, \qquad b_{s\mu\nu} \to b_{s\mu\nu} + 2\nabla_{[\mu}\lambda_{s\nu]} , \quad (2.4)$$

as well as internal symmetries of $\Phi_s$. These objects will be referred to as *hydrodynamic variables*. Demanding that the action can only depend on such variables, one can proceed to write down all the possible combinations of them that form scalars to construct the effective action, up to the desired order in the derivative expansion. Once $W$ is obtained, the constitutive relations can be obtained in the following way:

$$T^{\mu\nu} = \frac{2}{\sqrt{-g}} \frac{\delta W}{\delta g_{\mu\nu}} , \qquad J^{\mu\nu} = \frac{1}{\sqrt{-g}} \frac{\delta W}{\delta b_{\mu\nu}} . \quad (2.5)$$

The invariance of the generating function $Z$ under the background field transformations in Eq. (2.4) implies that these two currents satisfy the following Ward identities:

$$\nabla_\mu T^{\mu\nu} = H^\nu_{\ \rho\sigma} J^{\rho\sigma} , \qquad \nabla_\mu J^{\mu\nu} = 0 , \quad (2.6)$$

where $H \equiv db$ is the 3-form field strength of the 2-form background field $b_{\mu\nu}$. Note that since we are not working with two copies of the hydrodynamic variables, we will drop the subscript $s$ for the rest of this work. We would also like to point out that, as apparent from the above equation (2.6), the hodge dual of $(db)_{\mu\nu\lambda}$ plays the role of the external vector current $j^{\mu}_{\text{external}}$ that is injected into the system (see e.g. section II.A. of [30] for more details).

With this formalism in place, let us summarise our strategy for constructing the hydrodynamic theory of MHD in the strong magnetic field limit. Firstly, we identify the hydrodynamic variables, constructed from (a single copy of) $g_{\mu\nu}, b_{\mu\nu}$ and $\Phi$. Then we write down all possible scalars which constitute the effective action $W$ up to the second order in the derivative expansion. And the constitutive relations are obtained by varying this effective action. This procedure has also been applied to obtain the effective action for dissipationless relativistic fluid in [11], which we follow. Once this is done we can then use the effective action to find the linearised and non-linear solutions of the theory.

This approach implies that the theory is dissipationless and we shall justify this assumption in the next section. In the following sections, we will show that the strong magnetic field limit $B/T^2 \to \infty$ forbids terms at the first order in the derivative expansion. Moreover, the entropy current vanishes thus justifying the decoupling between the two sets of Schwinger-Keldysh degrees of freedom in (2.3) as well as our construction with non-dissipative effective action.

## 2.1 Formalism for non-dissipative fluid with one-form global symmetry

There are several ways to arrive at the dynamical variables for the zero temperature MHD employed in this work. From the point of view of a fluid with conserved number of strings [30] (see also [29]), the constitutive relations at the zeroth order in the derivative expansion are:

$$T^{\mu\nu} = (\varepsilon + p)u^{\mu}u^{\nu} + pg^{\mu\nu} - \mu\rho h^{\mu}h^{\nu} , \tag{2.7a}$$

$$J^{\mu\nu} = \rho\left(u^{\mu}h^{\nu} - u^{\nu}h^{\mu}\right) , \tag{2.7b}$$

where $u^{\mu}$ is the fluid four-velocity and $h^{\mu}$ is a unit vector parametrising the direction of the string. The thermodynamic quantities satisfy the first law and extensivity condition,

$$dp = sdT + \rho d\mu , \qquad \varepsilon + p = sT + \mu\rho , \tag{2.8}$$

where besides the usual energy density $\varepsilon$, pressure $p$, temperature $T$ and entropy $s$, we have an equilibrium string/magnetic flux density $\rho$ and its corresponding chemical potential $\mu$. As one enters the regime where the temperature $T$ is negligible, the usual fluid variables can be combined into a specific form which preserves the $SO(1,1)$ rotation between $u^{\mu}$ and $h^{\mu}$, namely

$$T^{\mu\nu} = -\varepsilon\Omega^{\mu\nu} + p\Pi^{\mu\nu}, \qquad J^{\mu\nu} = \rho u^{\mu\nu} \tag{2.9}$$

where, in terms of the original variables, we have:

$$u^{\mu\nu} = u^{\mu}h^{\nu} - u^{\nu}h^{\mu} , \tag{2.10a}$$

$$\Omega^{\mu\nu} = -u^{\mu}u^{\nu} + h^{\mu}h^{\nu} = u^{\mu}{}_{\alpha}u^{\alpha\nu} , \tag{2.10b}$$

$$\Pi^{\mu\nu} = g^{\mu\nu} - \Omega^{\mu\nu} . \tag{2.10c}$$

It is postulated in [30] that the corrections to MHD at low temperature can therefore be obtained by writing down the higher-derivative tensors constructed from $u^{\mu\nu}, \Omega^{\mu\nu}$ and $\Pi^{\mu\nu}$ which preserve the boost symmetry between $u^{\mu}$ and $h^{\mu}$. Insisting on using these variables has several physical consequences:

- there is no first derivative rank-two tensor (both symmetric and anti-symmetric) that can be constructed out of $\{u^{\mu\nu}, \Omega^{\mu\nu}, \Pi^{\mu\nu}\}$. This is in agreement with the fact that the system which remains at zero temperature does not dissipate heat. This observation is also confirmed in the case of strongly interacting holographic plasma [36] where all the transport coefficients at the first order in the derivative expansion vanish. *Consequently, the leading-order corrections to the system can only appear at the second order in the derivative expansion.*

At this point, one may proceed to write down all possible combinations of both symmetric and anti-symmetric rank two tensors constructed from the second derivatives of $\{u^{\mu\nu}, \Omega^{\mu\nu}, \Pi^{\mu\nu}\}$. Note however, that not all tensors one can construct are independent as derivatives of certain variables are related to one another via the conservation law (2.4) at the zeroth order in the derivative expansion. In terms of hydrodynamic variables, these relations are

$$\nabla_\lambda \mu = \mu u^{\alpha\beta} \Pi_\lambda{}^\gamma \nabla_\alpha u_{\beta\gamma} - \frac{\rho(\mu)}{\rho'(\mu)} u_{\lambda\alpha} \Pi^\beta{}_\gamma \nabla_\beta u^{\alpha\gamma} + H_{\alpha\beta\gamma} u^{\alpha\beta} \Pi_\lambda{}^\gamma \ , \tag{2.11a}$$

$$\Pi_\lambda{}^\gamma \Omega^{\alpha\beta} \nabla_\alpha u_{\beta\gamma} = 0 \ . \tag{2.11b}$$

Procedure outlined above has been employed to construct the higher-derivative expansion for charged neutral fluid [3–5]. A slight drawback of this approach is that one is also required to construct the non-equilibrium entropy current which constrains certain combinations of transport coefficients to either vanish or be positive definite (see e.g. [4,50,51]). Instead, one can use an additional crucial property of the zero temperature MHD to bypass this step, namely that

- The fact that the free energy is independent of temperature implies that the equilibrium entropy density, $s$, vanishes. Moreover, the entropy current which can be constructed from the Schwinger-Keldysh effective action is

$$s^\mu = V^\mu - \left(\frac{u_\nu}{T}\right) T^{\mu\nu} - \left(\frac{\mu h_\nu}{T}\right) J^{\mu\nu} \ , \qquad \nabla_\mu V^\mu = \mathcal{L}_{KMS} - \mathcal{L} \tag{2.12}$$

  where $\mathcal{L}$ is the effective Lagrangian associated to the Schwinger-Keldysh effective action $\tilde{W}$ and $\mathcal{L}_{KMS}$ is the $\mathbb{Z}_2$-KMS conjugate of the Lagrangian [16,52].

  In the enhanced symmetry system, the entropy current is not invariant under the $SO(1,1)$ rotation and has to vanish. *This implies that there is no entropy production ($\nabla_\mu s^\mu = 0$) and the effective action (2.2) splits into two copies* as in Eq. (2.3) due to the general argument of [19,52] [4].

As a result, we argue that the effective action for strong magnetic field limit of MHD can be described by the following effective action:

$$W = \int d^4x \sqrt{-g} \, \mathcal{L}_{eff} = \int d^4x \sqrt{-g} \Big( p(\mu) + \mathcal{L}_2 \Big) + \mathcal{O}(\partial^3) \tag{2.13}$$

where $\mathcal{L}_2$ is a combination of linearly independent two-derivative scalars constructed with $\mu, u^{\mu\nu}, \Omega^{\mu\nu}$ and $\Pi^{\mu\nu}$. Here we assume that the theory admits a gradient expansion and we will be focusing on the leading correction, which is at the second order in derivatives.

A sharper statement can be made using an effective action construction [42] (for the discussion of the effective action for a more conventional hydrodynamics see e.g. [54,55]).

---

[4]While the vanishing of the entropy production generally implies the decoupling between Schwinger-Keldysh copies, there are exceptions like a parity odd fluid or a system with an anomaly, as shown in [53].

A first step in this approach is to specify the relevant light degrees of freedom $\Phi$ alluded to in the introduction. The first relevant degrees of freedom are the two fiducial coordinates $\sigma^i = \{\sigma^1, \sigma^2\}$ which label the string/magnetic flux lines on the plane perpendicular to them. Additionally, each magnetic flux line is associated with a phase, $\exp\left(i \int_L a_\mu dx^\mu\right)$, where $L$ is the spatial curve parametrising this flux line, as illustrated in Fig 1.

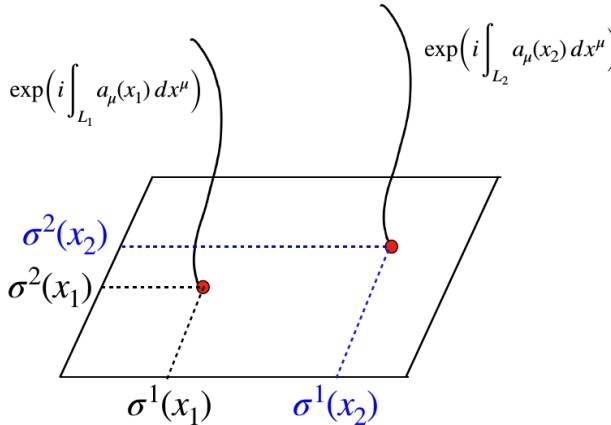

Figure 1: Labeling of the magnetic field lines by the coordinates $\sigma^i$ on the plane perpendicular to the "strings". Each field line is parametrised by a $U(1)$ phase $\exp\left(\int_L a_\mu dx^\mu\right)$.

Here the phase $a_\mu$ transforms together with the background field $b_{\mu\nu}$, see (2.4), in the following way:

$$b_{\mu\nu} \to b_{\mu\nu} + (\partial_\mu \Lambda_\nu - \partial_\nu \Lambda_\mu) , \qquad a_\mu \to a_\mu - \Lambda_\mu . \tag{2.14}$$

There are two advantages for choosing these light degrees of freedom. Firstly, it makes a line operator which is charged under the one-form $U(1)$ manifest. Namely, one can define a 't Hooft line $W(L) \equiv \exp\left(iq \int_L a_\mu(y)dy^\mu\right)$ of charge $q$ and show that it satisfies the Ward identity [5]

$$\left(\partial_\mu J^{\mu\nu}(x)\right)W(L) = iq\frac{\partial}{\partial x_\nu}\delta^4(x - y)W(L), \qquad \text{where} \qquad y \in L . \tag{2.15}$$

Secondly, from a more practical point of view, the invariance of the effective action under the shift (2.14) implies that Euler-Lagrange equation for $a_\mu$ is nothing more than the conservation of magnetic flux or the number of strings i.e. $\partial_\mu J^{\mu\nu} = 0$.

One can now construct the effecting action $W$ from $\{\sigma^i, a_\mu\}$. Note however that these variables cannot appear in an arbitrary form due to their spurious nature. This can be taken into account by demanding that the effective action has to be invariant under additional *internal symmetries* of the fields $\sigma^i$ and $a_\mu$. This means that the effective action can only contain certain combinations of $\{\sigma^i, a_\mu\}$, which turns out to be the hydrodynamic variables $\mu$, $u^{\mu\nu}$, $\Omega^{\mu\nu}$ and $\Pi^{\mu\nu}$ discussed at the beginning of this section. This procedure

---

[5]This is analogous to the Ward identity of the local operator $\mathcal{O}(y)$ with charge $q$ under the ordinary (zero-form) $U(1)$ global symmetry i.e.

$$\left(\partial_\mu j^\mu(x)\right)\mathcal{O}(y) = i\,q\,\delta^4(x - y)\mathcal{O}(y) ,$$

where $j^\mu$ is the conserved current of the ordinary global symmetry. For more details on how this is related to canonical quantisation see e.g. [56].

has been done for similar construction of the effective action for superfluid [54], ordinary fluid [55, 57, 58] and recently extended to fluid with higher-form global symmetry in [42]. For completeness, we summarise the setup in [42] in the remaining of this section where the internal symmetries are:

(i) *Reparametrisation symmetry :* This is due to the fact that the physical quantities cannot depend on the choice of parametrisation $\{\sigma^1, \sigma^2\}$ in the plane perpendicular to the strings. Thus, one demands that the action has to be invariant under the following reparametrisation symmetry:

$$\sigma^i \to \sigma'^i(\sigma^j), \tag{2.16}$$

This is analogous to the volume preserving diffeomorphisms for the ordinary fluid [55]. From this one can define an object akin to the fluid velocity in the following way:

$$u^{\mu\nu} = \epsilon^{\mu\nu\rho\sigma} \left( \frac{S_{\rho\sigma}}{\sqrt{S_{\alpha\beta} S^{\alpha\beta}/2}} \right), \qquad S_{\mu\nu} = 2\partial_{[\mu}\sigma^1 \partial_{\nu]}\sigma^2 . \tag{2.17}$$

This definition has the same property as the one defined via $u^\mu$ and $h^\mu$ in Eq. (2.10a) and satisfies $u^{\mu\nu} u_{\mu\nu} = -2$. However, it is only invariant up to a sign of $\det[\partial\sigma'/\partial\sigma]$. Thus, only products of even numbers of $u^{\mu\nu}$ or combination with odd power of $u^{\mu\nu}$ and $\mu$ can enter the effective action. The reason for the latter scenario will be apparent in the discussion below. At zero-derivative order, the only two non-trivial combinations of $u^{\mu\nu}$ are

$$\Omega^{\mu\nu} = u^\mu{}_\alpha u^{\alpha\nu}, \qquad \Pi^{\mu\nu} = g^{\mu\nu} - \Omega^{\mu\nu} . \tag{2.18}$$

By construction, one can see that these hydrodynamic variables satisfy the following relations:

$$\Omega^\mu{}_\alpha u^{\alpha\nu} = u^{\mu\nu}, \qquad \Pi^\mu{}_\alpha u^{\alpha\nu} = 0, \qquad \Omega^\mu{}_\alpha \Pi^{\alpha\nu} = \Pi^\mu{}_\alpha \Omega^{\alpha\nu} = 0 \tag{2.19}$$

and, of course, $\Omega^{\mu\nu}\Omega_{\mu\nu} = \Pi^{\mu\nu}\Pi_{\mu\nu} = 2$. One can therefore think of $\Omega^{\mu\nu}$ and $\Pi^{\mu\nu}$ as projectors of a vector onto a plane along and perpendicular to the string worldsheet, respectively.

(ii) *One-form chemical shift symmetry :* Due to the fact that the 't Hooft line $W$ is define via an integral along the string, one can shift the one-form phase $a_\mu$ by $\omega_\mu(\sigma^1, \sigma^2)$ that only depends on the coordinates perpendicular to the string:

$$a_\mu \to a_\mu + \omega_\mu(\sigma^1, \sigma^2) , \tag{2.20}$$

which yields the line operator with the same charge, via (2.15). For the effective theory to be independent of such ambiguity and the shift symmetry of the background fields (2.14), $W$ can only depend on the following combination:

$$\tilde{f}^{\mu\nu} = \Omega^{\mu\alpha}\Omega^{\nu\beta} \left( b_{\alpha\beta} + 2\partial_{[\alpha}a_{\beta]} \right) . \tag{2.21}$$

One can then define a scalar quantity out of $\tilde{f}^{\mu\nu}$, which turns out to be the chemical potential $\mu$ for the one-form $U(1)$ symmetry,

$$\mu^2 = -\frac{1}{2}\tilde{f}_{\mu\nu}\tilde{f}^{\mu\nu} . \tag{2.22}$$

Notice that $\tilde{f}^{\mu\nu}/\mu$ has the same property as $u^{\mu\nu}$ when acted upon by the projectors $\Omega^{\mu\nu}$ and $\Pi^{\mu\nu}$. At the zeroth order in the derivative expansion one may concluded

that they are identical. This is not generally true as they can differ by a derivative correction. This is nothing but the manifestation of the fact that the fluid variables are defined up to derivative corrections, commonly known as the frame choice [1] (see also [2] for more recent discussion). It is nevertheless convenient to impose this condition at all orders in the derivative expansion [6]:

$$\tilde{f}^{\mu\nu} = \mu u^{\mu\nu} \ . \tag{2.23}$$

Note that, from the definition in Eq. (2.21), $\tilde{f}^{\mu\nu}$ is invariant under $\sigma^i \to \sigma'^i$ with $\det[\partial\sigma^i/\partial\sigma'^j] = -1$. Thus, the combination $\mu u^{\mu\nu}$ is invariant under the reparametrisation of $\sigma^i$. This implies that the terms with odd number of $u^{\mu\nu}$ can be made invariant if they are accompanied by odd powers of $\mu$.

Before moving on, we shall comment on the relation between this choice of variables and the more conventional FFE formulation in e.g. [39]. There, the field strength tensor $F^{\mu\nu} = (\star\tilde{f})^{\mu\nu}$ is simply written as $F^{\mu\nu} = S^{\mu\nu} = 2\partial_{[\mu}\sigma^1\partial_{\nu]}\sigma^2$, where $S^{\mu\nu}$ appears in Eq. (2.17) and the magnetic flux is trivially conserved. The key difference here is that, in the formalism outlined here, there exists a conserved current $J^{\mu\nu}$ representing the conservation of magnetic flux but $J^{\mu\nu} \neq \tilde{f}^{\mu\nu}$ except at the zeroth order in the derivative expansion! This allows one to go beyond the unrealistic assumption of $\mathbf{E} \cdot \mathbf{B} = 0$ in the conventional FFE formulation.

To sum up, we argued that the dynamical variables $\Phi = \{\sigma^i, a_\mu\}$ have to appear in the following combination: even number of $u^{\mu\nu}$, projectors $\Omega^{\mu\nu}$ and $\Pi^{\mu\nu}$, the scalar $\mu^2$ or products of an odd number of $\mu$ with an odd number of $u^{\mu\nu}$. The remaining steps are to organise these quantities order by order in the derivative expansion. Following the approach in [11], we write down all possible scalars at the second order in derivatives, which is the leading-order correction, modulo terms that can be related to one another via ideal limit equations of motion (2.11a)-(2.11b).

## 2.2 Classification of the second-order effective action

In this section, we will outline the procedure to construct the effective action for $T^2 \ll B$ MHD up to the second order in the derivative expansion and summarise the result. Using the formalism outlined above, the resulting effective action build from $\sigma^i$, $a_\mu$ and the background fields $g_{\mu\nu}$, $b_{\mu\nu}$ is

$$\mathcal{L}_{eff} = p(\mu) + \alpha s^{(\alpha)} + \sum_{i=1}^{2} \beta_i s_i^{(\beta)} + \sum_{i=1}^{8} \gamma_i s_i^{(\gamma)} + \mathcal{O}(\partial^3) \ , \tag{2.24}$$

where $p(\mu)$ is a scalar function of the chemical potential $\mu$, which turns out to be the thermodynamic pressure. The higher derivative terms are composed of independent scalars $s^{(\alpha)}$, $s_i^{(\beta)}$ and $s_i^{(\gamma)}$, each multiplied by a function $\alpha$, $\beta_i$ and $\gamma_i$, respectively, referred to as *transport coefficients*. To justify the gradient expansion, we shall restrict our setup to a situation where there is a hierarchy of scales, namely the characteristic IR length scale set by $\partial \sim \omega, k$, the thermodynamic scale set by the chemical potential $\mu$ and a microscopic length scale $\ell_{\mathrm{micro}}$ satisfy:

$$\frac{\omega}{\sqrt{\mu}}, \frac{k}{\sqrt{\mu}} \quad \ll \quad 1 \quad \ll \quad \frac{1}{\sqrt{\mu}\ell_{\mathrm{micro}}} \ . \tag{2.25}$$

---

[6]Similar choice has also been used in the construction of the effective action of the charge-neutral fluid in [11] where the entropy current is chosen to have no derivative corrections. We will discuss the frame choice for this setup again in section 2.3.

Note that $\mu$ is the chemical potential for the magnetic flux density and therefore has the dimension of [length]$^{-2}$. In addition, for the gradient expansion to make sense, one requires that the scalars $\{\alpha, \beta_i, \gamma_i\}$ are nonsingular in the limit where $\sqrt{\mu}\ell_{\mathrm{micro}} \to 0$. In short, we require that the transport coefficients are finite and only depend on $\mu$ in the limit where the microscopic length scale is infinitesimally small. We will return to this issue at the end of this section with an explicit example where the implication of this assumption becomes more transparent.

We shall proceed to outline the derivation of the effective action (2.24). Let us first consider the possible structures at the zeroth order in the derivative expansion. There is only one scalar at this order, namely the chemical potential $\mu$. Thus the effective action can only be a scalar $p(\mu)$ at this order. As we proceed to higher-order corrections, it is useful to write down all the possible (un-contracted) tensors at a given order. For the first-order terms, we can build a scalar out of the following objects

$$\{\nabla_\rho \mu, \ \nabla_\rho u^{\mu\nu}, \ H_{\alpha\beta\gamma}\} \ . \tag{2.26}$$

We can see that all of the first-order derivative-terms have an odd number of indices and the terms without derivatives all have an even number of indices so there are no scalars at the first order in $T = 0$ MHD. Note also that not all quantities listed above are independent as the derivatives of thermodynamic quantities are related to certain divergences of $u^{\mu\nu}$ via the equations of motion at the zeroth order (2.11a)-(2.11b). Note also that, because there is no vector at the zeroth order, one cannot even build either symmetric or antisymmetric rank-two tensor at the first order in the derivative expansion. This implies that all the transport coefficients at this order must vanish as pointed out in [30].

The second-order terms are the main result of this work. Out of the list of all hydrodynamic variables, the second derivative scalars can be obtained by contractions of the following tensors:

$$\begin{aligned}
\{\nabla_\rho \mu \nabla_\sigma \mu, \ \nabla_\rho \nabla_\sigma \mu, \ \nabla_\rho \mu \nabla_\sigma u^{\mu\nu}, \ \nabla_\rho \nabla_\sigma u^{\mu\nu}, \ \nabla_\rho u^{\mu\nu} \nabla_\sigma u^{\alpha\beta}, \\
\nabla_\rho \mu \, H_{\alpha\beta\gamma}, \ \nabla_\rho u^{\mu\nu} H_{\alpha\beta\gamma}, \ H_{\alpha\beta\gamma} H_{\rho\sigma\lambda}, \ \nabla_\rho H_{\alpha\beta\gamma}, \ R_{\alpha\beta\gamma\delta}\} \ .
\end{aligned} \tag{2.27}$$

Combinatorially, there are about over two hundred combinations of contractions. Of course, not all of the scalars constructed in such a way are independent. One can reduce the number of scalars by requiring that the scalars are not related to one another via ideal limit Ward identity Eq. (2.11a) and that they do not differ from one another by a total derivative. The latter condition came from the fact that the total derivative pieces do not contribute to the constitutive relation and was also employed in [11] for charge neutral fluid. In addition, one can use properties of hydrodynamic variables

(i) Normalisation and projective properties of $u^{\mu\nu}$ (see Appendix A.2)

(ii) Projective properties of $H = db$ (see Appendix A.3)

(iii) Jacobi identities for $u^{\mu\nu}$ (see Appendix A.4)

(iv) We also impose that all the scalars are invariant under all the fundamental discrete symmetries: the charge conjugation ($\mathcal{C}$), time-reversal ($\mathcal{T}$) and parity ($\mathcal{P}$). The discrete charge assignments of the hydrodynamic variables is discussed in Appendix B.2.

By implementing this procedure, we find that the effective action at the second order is captured by eleven independent scalars (more details regarding this procedure are presented in Appendix B.1 while the Mathematica code used to implement these steps can

be found in [59]). There is no first-order derivative terms and the second order derivative pieces can be categorised into three classes with respect to the power of the three form field strength of the 2-form source $H = db$. There is one scalar which depends quadratically in $H$ with the transport coefficient $\alpha$,

$$s^{(\alpha)} = H_{\alpha\gamma\lambda}H_{\beta\delta\kappa}\Pi^{\alpha\beta}\Pi^{\gamma\delta}\Omega^{\lambda\kappa} \ . \tag{2.28a}$$

Similarly, there are two terms which depend linearly on $H$ with the corresponding transport coefficients $\beta_i$,

$$s_1^{(\beta)} = H_{\beta\delta\kappa}u^{\alpha\beta}u^{\gamma\delta}\Pi^{\lambda\kappa}\nabla_\gamma u_{\alpha\lambda} \ ,$$
$$s_2^{(\beta)} = H_{\beta\delta\kappa}\Pi^{\alpha\beta}\Pi^{\gamma\delta}\Omega^{\lambda\kappa}\nabla_\gamma u_{\alpha\lambda} \ . \tag{2.28b}$$

And lastly, there are eight scalars that do not depend on $H$ whose transport coefficients are $\gamma_i$,

$$s_1^{(\gamma)} = R_{\alpha\gamma\beta\delta}u^{\alpha\beta}u^{\gamma\delta} \ , \qquad\qquad s_2^{(\gamma)} = R_{\alpha\gamma\beta\delta}\Pi^{\alpha\beta}\Pi^{\gamma\delta} \ ,$$
$$s_3^{(\gamma)} = R_{\alpha\gamma\beta\delta}\Pi^{\alpha\beta}\Omega^{\gamma\delta} \ ,$$
$$s_4^{(\gamma)} = u^{\alpha\beta}u^{\gamma\delta}\Pi^{\lambda\kappa}\nabla_\beta u_{\delta\kappa}\nabla_\gamma u_{\alpha\lambda} \ , \qquad s_5^{(\gamma)} = u^{\alpha\beta}u^{\gamma\delta}\Pi^{\lambda\kappa}\nabla_\beta u_{\alpha\lambda}\nabla_\delta u_{\gamma\kappa} \ , \tag{2.28c}$$
$$s_6^{(\gamma)} = \Pi^{\alpha\beta}\Pi^{\gamma\delta}\Omega^{\lambda\kappa}\nabla_\beta u_{\delta\kappa}\nabla_\gamma u_{\alpha\lambda} \ , \qquad s_7^{(\gamma)} = \Pi^{\alpha\beta}\Pi^{\gamma\delta}\Omega^{\lambda\kappa}\nabla_\beta u_{\alpha\lambda}\nabla_\delta u_{\gamma\kappa} \ ,$$
$$s_8^{(\gamma)} = \Pi^{\alpha\beta}\Pi^{\gamma\delta}\Omega^{\lambda\kappa}\nabla_\gamma u_{\alpha\lambda}\nabla_\delta u_{\beta\kappa} \ .$$

The transport coefficients $\alpha$, $\beta_i$ and $\gamma_i$ associated to these higher-order derivative-terms have to be determined by the microscopic correlation functions via Kubo formulae (derived in section 3.3). These transport coefficients are dimensionful quantities whose units can be easily determined from the effective action i.e.

$$\alpha \sim [\text{length}]^2, \qquad \beta \sim [\text{length}]^0, \qquad \gamma \sim [\text{length}]^{-2} \tag{2.29}$$

In typical hydrodynamic setup e.g. [1, 3, 5], the transport coefficients only depend on the thermodynamic quantities thus implying that combinations $\alpha|\mu| \equiv \bar{\alpha}$, $\beta_i$ and $\gamma_i/|\mu| \equiv \bar{\gamma}_i$ are dimensionless quantities independent of $\mu$. This assumption also implies that $\beta_1 = \beta_2 = 0$ due to the fact that $s_1^\beta$ and $s_2^\beta$ are not invariant under the reparametrisation symmetry (2.16).

This strict $\mu$-dependence can be relaxed as one allows $\bar{\alpha}$, $\beta_i$ and $\bar{\gamma}_i$ to also depend on $\mu\ell_{\text{micro}}^2$ i.e. the microscopic theory's length scale in the units of macroscopic length scale [7]. However, one has to restrict how the transport coefficients depend on $\ell_{\text{micro}}$ for the gradient expansion to be well-defined. For example, it could happen that

$$\gamma_i = |\mu| \left( \frac{1}{(\mu\ell_{\text{micro}})^n} + ... \right), \qquad \text{for} \qquad n > 0 \tag{2.30}$$

which diverges in the limit where the microscopic energy scale $1/\ell_{\text{micro}} \to \infty$. We shall restrict our analysis to the case where this does not happen otherwise the gradient expansion will breakdown. Consequently, one now allows $s_2^\beta$ and $s_3^\beta$ to be added to the effective action, e.g.

$$\beta_1 = \mu\ell_{\text{micro}}^2 \left( c_2 + \mathcal{O}(\ell_{\text{micro}}^2) \right) , \qquad \beta_2 = \mu\ell_{\text{micro}}^2 \left( c_3 + \mathcal{O}(\ell_{\text{micro}}^2) \right) \tag{2.31}$$

for some constants $c_2$ and $c_3$. It is important to note that these coefficients depend explicitly on $\mu$ and not its absolute value. This is done so so that the combinations $\beta_i s_i^\beta$ are invariant under the reparametrisation symmetry (2.16) as pointed out in [42].

---

[7]This situation can happen in e.g. D3/D7 branes where the Landau pole scale can appear in the thermodynamic quantities [60] and in the transport coefficients of weakly coupled QED at finite temperature [61].

Lastly, we check how the second-order terms transform under the standard discrete symmetries: the charge conjugation ($\mathcal{C}$), parity ($\mathcal{P}$) and time reversal ($\mathcal{T}$). It turns out that all scalars listed above are invariant under all $\mathcal{C}, \mathcal{P}$ and $\mathcal{T}$. This property can be easily derived using the discrete charge assignments for the hydrodynamic variables which we report on in Appendix B.2.

These are the main results in this work so let us summarise them here. Assuming that the gradient expansion can be performed, we find that there are eleven second-order corrections to the effective action for plasma at strong magnetic field. They consist of one term $\alpha$ which depends quadratically on $H = db$, two terms $\beta_i$ which depend linearly on $H$ and eight terms $\gamma_i$ that only depend on the curvature and derivatives of the "fluid velocity" $u^{\mu\nu}$. All the terms presented here are invariant under all discrete $\mathcal{C}, \mathcal{P}, \mathcal{T}$ symmetries (the rest of the allowed independent structures that are odd under these discrete symmetries are also presented in Appendix B.1. The rest of this paper will explore the consequences of these second-order transport coefficients.

## 2.3 Constitutive relation and frame choice

Upon varying the effective action with respect to the background metric $g_{\mu\nu}$ and the two-form gauge field $b_{\mu\nu}$, one finds the constitutive relations which can be written in the following form:

$$
\begin{aligned}
T^{\mu\nu} &= -(\varepsilon + \delta\varepsilon)\,\Omega^{\mu\nu} + (p + \delta p)\,\Pi^{\mu\nu} + t^{\mu\nu}_{SO(1,1)} + t^{\mu\nu}_{SO(2)} + t^{\mu\nu}_{v\otimes v}\,, \\
J^{\mu\nu} &= (\rho + \delta\rho)\,u^{\mu\nu} + s^{\mu\nu}_{SO(2)} + s^{\mu\nu}_{v\otimes v}\,,
\end{aligned}
\tag{2.32}
$$

where $\delta\varepsilon$, $\delta p$, $\delta\rho$, $t^{\mu\nu}$, $s^{\mu\nu}$ are scalars and (traceless) rank-two tensors at the second order in the derivative expansion. Different subscripts under the tensors $t^{\mu\nu}$, $s^{\mu\nu}$ in (2.32) represent how they transform under $SO(1,1)$ and $SO(2)$ symmetries. More precisely, the tensors $t^{\mu\nu}_{SO(1,1)}$, $t^{\mu\nu}_{SO(2)}$, $t^{\mu\nu}_{v\otimes v}$ transform as tensor representations of $SO(1,1)$ and $SO(2)$, and a vector representation of $SO(1,1) \otimes SO(2)$, respectively. In practice, they can be obtained from $T^{\mu\nu}$ via the following projections:

$$
\left(\Omega^\mu{}_\alpha \Omega^\nu{}_\beta\right) T^{\alpha\beta} = -(\varepsilon + \delta\varepsilon)\Omega^{\mu\nu} + t^{\mu\nu}_{SO(1,1)},
\tag{2.33a}
$$

$$
\left(\Pi^\mu{}_\alpha \Pi^\nu{}_\beta\right) T^{\alpha\beta} = -(p + \delta p)\Pi^{\mu\nu} + t^{\mu\nu}_{SO(2)},
\tag{2.33b}
$$

$$
\frac{1}{2}\left(\Omega^\mu{}_\alpha \Pi^\nu{}_\beta + \Omega^\nu{}_\alpha \Pi^\mu{}_\beta\right) T^{\alpha\beta} = t^{\mu\nu}_{v\otimes v},
\tag{2.33c}
$$

and taking the trace of Eq.(2.33a) and Eq.(2.33b) enables us to separate $\delta\varepsilon$, $\delta p$ and the traceless parts $t^{\mu\nu}_{SO(1,1)}$ and $t^{\mu\nu}_{SO(2)}$. Similar procedure can be used to obtain $\delta\rho$, $s^{\mu\nu}_{SO(2)}$ and $s^{\mu\nu}_{v\otimes v}$ from $J^{\mu\nu}$ [8]. The full constitutive relations at non-linear level with curvature and the background field $b_{\mu\nu}$ turned on can be found in Appendix D and the linearised constitutive relations with flat metric and vanishing background field can be found in Appendix C. These constitutive relations are cumbersome in practice and we believe that it is much more convenient to work with the effective action directly.

Before discussing the advantages of this decomposition, let us discuss one subtle issue of hydrodynamic description. As emphasised in the relativistic hydrodynamics' literature (see e.g. [1] and, for modern review [2, 62]), the out of equilibrium values of the chemical

---

[8]The constitutive relation for $J^{\mu\nu}$ does not contain the $s^{\mu\nu}_{SO(1,1)}$ part because it always vanishes. $s^{\mu\nu}_{SO(1,1)}$ would be the $SO(1,1)$ part of $J^{\mu\nu}$ without the scalar part proportional to $u^{\mu\nu}$ so it can be defined as $s^{\mu\nu}_{SO(1,1)} \equiv \frac{1}{2}\left(\Omega^\mu{}_\alpha \Omega^\nu{}_\beta - \Omega^\nu{}_\alpha \Omega^\mu{}_\beta + u^{\mu\nu} u_{\alpha\beta}\right) J^{\alpha\beta}$. But this projector vanishes because of the Jacobi identity (A.21) so $s^{\mu\nu}_{SO(1,1)} = 0$.

potential $\mu$ and the two-index velocity $u^{\mu\nu}$ have no unique definition. While in the effective action construction we impose the condition that $\tilde{f}^{\mu\nu} = \mu u^{\mu\nu}$ without any derivative corrections for convenience, this is not a necessary condition [9]. In fact, at the level of the constitutive relations we have a freedom to redefine the chemical potential and $u^{\mu\nu}$ by second-derivative quantities in the following way

$$\mu \to \mu + \delta\mu_F(\partial^2), \qquad u^{\mu\nu} \to u^{\mu\nu} + \delta u_F^{\mu\nu}(\partial^2) \tag{2.34}$$

where $\delta\mu_F$ and $\delta u_F^{\mu\nu}$ are a scalar and a tensor of our choice, usually chosen to simplify the constitutive relations. Let us first see what happens to the constitutive relations when we redefine the chemical potential. It turns out that the only terms affected by the choice of $\mu$ are scalars $\delta\varepsilon$, $\delta p$ and $\delta\rho$, namely:

$$\delta\varepsilon \to \delta\varepsilon + \frac{\partial\varepsilon}{\partial\mu}\delta\mu_F, \qquad \delta p \to \delta p + \rho\delta\mu_F, \qquad \delta\rho \to \delta\rho + \frac{\partial\rho}{\partial\mu}\delta\mu_F . \tag{2.35}$$

This indicates that we can choose $\delta\mu_F$ to eliminate one second-order correction to $\varepsilon$, $p$ or $\rho$. The choice of $u^{\mu\nu}$ is more subtle. Firstly, one has to realise that $\delta u^{\mu\nu}$ transforms as a product of vector representations in $SO(1,1)$ and $SO(2)$ while $u^{\mu\nu}$ in equilibrium only transforms under $SO(1,1)$, see [30] and Appendix A.2 . We find that the second-order tensors that can be affected by this choice are

$$t_{v\otimes v}^{\mu\nu} \to t_{v\otimes v}^{\mu\nu} - (\varepsilon + p)\left(u^\mu{}_\alpha \delta u_F^{\alpha\nu} + \delta u^\mu{}_\alpha u_F^{\alpha\nu}\right), \tag{2.36a}$$

$$s_{v\otimes v}^{\mu\nu} \to s_{v\otimes v}^{\mu\nu} + \rho\delta u_F^{\mu\nu} , \tag{2.36b}$$

indicating that one can remove either $t_{v\otimes v}^{\mu\nu}$ or $s_{v\otimes v}^{\mu\nu}$ by appropriate choice of $\delta u_F^{\mu\nu}$. This freedom is referred to as a *frame choice* and various choices of hydrodynamic variables have been employed in the literature [10]. In [30], the choice where $\mathcal{R} = \rho$ received no second-order derivative-corrections was made. However, in the current work, it is more convenient to use the constitutive relations obtained directly from the effective action without making additional redefinition.

It is worth mentioning that the frame choice is not innocuous. As shown in the case of the fluid with ordinary $U(1)$ global symmetry, inappropriate frame choice yields unphysical non-hydrodynamic mode that can lead to instabilities even in the stationary fluid [63] (see also [64,65] for recent discussion). We will soon see in Section 3 that here the linearised perturbation also contains non-hydrodynamic modes but, fortunately, they do not lead to instability. Furthermore, the pole we found cannot be removed by the frame choice.

For our purpose, the decomposition (2.32) singles out the terms that are unaffected by the frame choice, namely $s_{SO(2)}^{\mu\nu}$. This $SO(2)$ component of $J^{\mu\nu}$ is responsible for the acceleration of the charged particle along the magnetic field $P = \mathbf{E} \cdot \mathbf{B} \sim \epsilon_{\mu\nu\rho\sigma}J^{\mu\nu}J^{\rho\sigma}$, namely

$$\epsilon_{\mu\nu\rho\sigma}J^{\mu\nu}J^{\rho\sigma} = \rho\,\epsilon_{\mu\nu\rho\sigma}u^{\mu\nu}s_{SO(2)}^{\mu\nu} + \mathcal{O}(\partial^3) \tag{2.37}$$

as $s_{SO(2)}^{\mu\nu}$ is the only component of $J^{\mu\nu}$ that is orthogonal to $u^{\mu\nu}$. This allows us to single out the terms in the effective action that are responsible for $\mathcal{P} > 0$ in a strong magnetic field.

---

[9]Similar issues have been discussed in the context of the effective action for the charge-neutral relativistic fluid in [11,55] (see also [16,21]) and equilibrium partition function of a fluid with one-form global symmetry in [35]

[10]For example, in the case of a fluid with ordinary (zero-form) $U(1)$ global symmetry, the choice of temperature $\delta T_F$, chemical potential $\delta\mu_F$ and fluid velocity $\delta u_F^\mu$ is commonly used to eliminate $\delta\varepsilon$, $\delta\rho$ so that $T^{\mu\nu}u_\nu = -\varepsilon u^\mu$, and is known as *Landau frame* [1]. See also [2] for discussion concerning different frame choices.

# 3 Linearised perturbation and correlation functions

We follow the approach of extracting two-point (and three-point) correlation functions of [66] (see also [2, 67]). This approach, known as variational method, can be done by defining the one-point generating function

$$\mathbb{T}^{\mu\nu} = 2\frac{\delta S_{eff}}{\delta g_{\mu\nu}}, \qquad \mathbb{J}^{\mu\nu} = \frac{\delta S_{eff}}{\delta b_{\mu\nu}} \, . \tag{3.1}$$

The retarded correlation functions are then obtained by varying the one-point generating function. Following the convention of [66], we write down one-point generating functions up to the cubic order in the perturbations $\delta g$, $\delta b$ as

$$
\begin{aligned}
\delta\mathbb{T}^{\mu\nu}(x) = &-\frac{1}{2}\int d^4x' \left( G_{TT}^{\mu\nu,\rho\sigma}(x,x')\,\delta g_{\rho\sigma}(x') + 2G_{TJ}^{\mu\nu,\rho\sigma}(x,x')\,\delta b_{\rho\sigma}(x') \right) \\
&+\frac{1}{4}\int d^4x'd^4x'' \left( G_{TTT}^{\mu\nu,\rho\sigma,\alpha\beta}(x,x',x'')\delta g_{\rho\sigma}(x')\delta g_{\alpha\beta}(x'') \right. \\
&\left. +\frac{1}{2}G_{TTJ}^{\mu\nu,\rho\sigma,\alpha\beta}(x,x',x'')\delta g_{\rho\sigma}(x')\delta b_{\alpha\beta}(x'') + G_{TJJ}^{\mu\nu,\rho\sigma,\alpha\beta}\delta b_{\rho\sigma}(x')\delta b_{\alpha\beta}(x'') \right) \, , \\
\delta\mathbb{J}^{\mu\nu}(x) = &-\frac{1}{2}\int d^4x' \left( G_{JT}^{\mu\nu,\rho\sigma}(x,x')\,\delta g_{\rho\sigma}(x') + 2G_{JJ}^{\mu\nu,\rho\sigma}(x,x')\,\delta b_{\rho\sigma}(x') \right) \\
&+\frac{1}{4}\int d^4x'd^4x'' \left( G_{JTT}^{\mu\nu,\rho\sigma,\alpha\beta}(x,x',x'')\delta g_{\rho\sigma}(x')\delta g_{\alpha\beta}(x'') \right. \\
&\left. +\frac{1}{2}G_{JJT}^{\mu\nu,\rho\sigma,\alpha\beta}(x,x',x'')\delta b_{\rho\sigma}(x')\delta g_{\alpha\beta}(x'') + G_{JJJ}^{\mu\nu,\rho\sigma,\alpha\beta}\delta b_{\rho\sigma}(x')\delta b_{\alpha\beta}(x'') \right) \, ,
\end{aligned}
\tag{3.2}
$$

where $G_{AB}^{\mu\nu,\rho\sigma}, G_{ABC}^{\mu\nu,\rho\sigma,\alpha\beta}$ with $A, B, C = T, J$ are fully retarded two- and three-point correlation functions evaluated in flat space with vanishing external background gauge field (for the procedure to obtained different kind of real-time correlation functions see e.g. [68]).

The fluctuations and eigenmodes of the theory can be studied in two different ways. A more conventional one is to vary the effective action w.r.t. the metric and gauge field to obtain the stress-energy tensor $T^{\mu\nu}$ and $J^{\mu\nu}$. Then one applies the Ward identity to find the spectrum. The second way is to utilise the effective action formalism by writing the fields $\sigma^i$ and $a_\mu$ as

$$
\begin{aligned}
\sigma^i &= x^i + \pi^i(t,x,z) \, , \\
a_\mu &= \frac{1}{2}\mu z dt + \mathfrak{a}_\mu dx^\mu \, ,
\end{aligned}
\tag{3.3}
$$

for $i = x, y$. Upon implementing (3.3) in the action and varying with the background fields $\delta g_{\mu\nu}$, $\delta b_{\mu\nu}$, obtaining the Euler-Lagrange equations for $\pi_i$ and $\mathfrak{a}_\mu$ becomes extremely efficient. These two approaches compliment one another and, given the lengthly effective action, serve as a good consistency check. It is also helpful to note the relations between variables in these two approaches, namely

$$\delta\mu = \partial_t\mathfrak{a}_z - \partial_z\mathfrak{a}_t \, , \qquad u^{ti} = -\partial_z\pi^i \, , \qquad u^{zi} = \partial_t\pi^i \, . \tag{3.4}$$

## 3.1 Propagation along the magnetic field line

When the perturbation is a function of $(t, z)$, the relevant equations of motion that contain nontrivial modes are the Ward identities of the transverse channel, namely

$$\nabla_\mu T^{\mu i} = H^{i\alpha\beta}J_{\alpha\beta} \, , \qquad \nabla_\mu J^{\mu i} = 0 \, , \tag{3.5}$$

where $i = \{x, y\}$. The fluctuations of $u^{\mu\nu}$ that constitute the above equations of motion are $\delta u^{ti}$ and $\delta u^{zi}$. Turning off the metric and the two-form gauge field fluctuations, we find that the conservation of two-form current gives

$$\partial_t u^{ti} + \partial_z u^{zi} = 0 . \tag{3.6}$$

This equation is automatically satisfied in the effective action setup where

$$u^{ti} = -\partial_z \pi_i , \qquad u^{zi} = \partial_t \pi_i . \tag{3.7}$$

The conservation of transverse momentum then yields the wave equation for the field $\pi_i$

$$\rho\mu(\partial_t^2 + \partial_z^2)\pi_i - 2(\gamma_4 - \gamma_5)(\partial_t^2 + \partial_z^2)^2 \pi_i = 0 . \tag{3.8}$$

The above equation can be obtained in two independent ways. One can either substitute the solution (3.7) into the linearised constitutive relations in Appendix C and plug $T^{\mu i}$ into the Ward identity [11]. Equivalently, one can find the Euler-Lagrange equation of the effective action (2.24). The spectrum of this sector is

$$\omega^2 = k_z^2 , \qquad \omega^2 = \frac{\mu\rho}{2(\gamma_4 - \gamma_5)} + k_z^2 . \tag{3.9}$$

This indicates that the first mode, which is the zero temperature limit of the Alfvén wave, received no correction from the second-order derivative-corrections. On the other hand, the new gapped mode sets the scale where the hydrodynamic expansion breaks down.

It is not uncommon that the second-order hydrodynamic contains non-hydrodynamic modes. We can argue that the gapped mode is outside the regime of validity of hydrodynamics since, at $k_z = 0$, the spectrum becomes $\omega/\mu \sim \rho/\mu$, assuming that $(\gamma_4 - \gamma_5)/\mu$ is of $\mathcal{O}(1)$. It is also possible that this mode can be removed upon the field redefinition procedure.

To compute the correlation functions for the fluctuations of this type, it is useful to couple the theory to background metric and gauge field. The solution for $\pi_i$, in Fourier space, can be written in terms of the perturbations of $g_{\mu\nu}$ and $b_{\mu\nu}$, namely

$$\pi_i(\omega, k_z) = \frac{i}{2\mathcal{P}_A^\perp(\omega, k_z)} \Bigg\{ (\omega\delta g_{ti} + k_z\delta g_{iz}) - 2(\omega\delta b_{zi} + k_z\delta b_{ti}) $$
$$- (\omega^2 - k_z^2)\Big[ 2(\gamma_4 - \gamma_5)(\omega\delta g_{ti} + k_z\delta g_{zi})) + \beta_1 (k_z\delta b_{tx} - \omega\delta b_{xz})\Big] \Bigg\} , \tag{3.10}$$

where the polynomial $\mathcal{P}_A^\perp$ is

$$\mathcal{P}_A^\perp(\omega, k) = (\omega^2 - k^2)\left(\mu\rho - 2(\gamma_4 - \gamma_5)(\omega^2 - k^2)\right) \tag{3.11}$$

With this information at hand, one can proceed to extract the correlation functions which contain the pole describing the spectrum (3.9). Only the correlators involving $J^{ti}, J^{zi}, T^{ti}$ and $T^{zi}$ encode the propagating mode. This can be seen by considering the one-point generating functions in the presence of small metric and gauge field fluctuations:

$$\mathbb{T}^{ti} = \left( p - k^2(\gamma_1 - \gamma_4) + \frac{\omega^2\mu\rho}{\omega^2 - k^2} \right)\delta g_{ti} - \omega k \left( \gamma_1 - \gamma_4 + \frac{\mu\rho}{\omega^2 - k^2} \right)\delta g_{xz}$$
$$+ \frac{2\omega\rho}{\omega^2 - k^2}(k\delta b_{ti} + \omega\delta b_{zi}) , \tag{3.12}$$
$$\mathbb{J}^{ti} = \frac{k}{\omega^2 - k^2}\left[ (\omega\delta g_{ti} + k\delta g_{zi}) - \left( \frac{2\rho^2 - \beta_1(\omega^2 - k^2)}{4\mathcal{P}_A^\perp(\omega, k)} \right)(k\delta b_{ti} + \omega b_{zi})\right]$$

---

[11]This is consistent with the equation obtained in the linearised constitutive relations in flat space in [30] where we can identify $\nu_1 = 2(\gamma_4 - \gamma_5)$

We can extract the correlation functions which contain the pole using the prescription (3.2). For example, we have

$$G_{TT}^{ti,ti}(\omega, k_z) = p - k^2(\gamma_1 - \gamma_4) + \frac{\omega^2 \mu \rho}{\omega^2 - k^2} \, , \qquad G_{JT}^{ti,zi} = \frac{k^2}{\omega^2 - k^2} \, . \tag{3.13}$$

Interestingly, there is no pole in the energy density and one-form charge density correlation functions. This may seem odd at first but it can be understood as a consequence of string reparametrisation symmetry. Effectively, this symmetry freezes the temperature to zero. This, together with the fact that there is no fluctuations in "string number density" along the direction of the string, indicates that there is no propagating mode in the longitudinal channel. This can be explicitly seen as the only relevant degree of freedom in the longitudinal channel, namely $\delta\mu = 2(\partial_z \mathfrak{a}_t - \partial_t \mathfrak{a}_z)$, can be solved in terms of the sources as

$$\delta\mu = 2\delta b_{tz} + \frac{1}{2}\left(\mu - \frac{2\gamma_1' k_z^2}{\chi}\right)\delta g_{tt} + \frac{\rho}{2\chi}(\delta g_{xx} + \delta g_{yy}) + \frac{1}{2}\left(\mu + \frac{2\gamma_1' \omega^2}{\chi}\right)\delta g_{zz} - \frac{2\gamma_1'}{\chi}\omega k_z \delta g_{tz} \, . \tag{3.14}$$

where $\chi$ denotes the susceptibility $\chi = \partial\rho/\partial\mu$. One can see that, unlike $\pi_i$, the solution for $\delta\mu$ contains no pole for the propagating mode.

Before moving to a different perturbation channel, let us point out that the gapped mode in (3.9) cannot be removed by the frame choice. This can be seen in the following way. Instead of using the effective action, one can equivalently use the constitutive relations for the linearised perturbation in Appendix C. We will find that for the second derivative correction listed in Appendix C the only non-zero contributions are

$$t_{v\otimes v}^{ti} = \nu_1\left(\partial_t^2 - \partial_z^2\right)\delta u^{iz} \, , \qquad t_{v\otimes v}^{zi} = \nu_1(\partial_t^2 - \partial_z^2)\delta u^{ti} \, , \tag{3.15}$$

for $i = x, y$ in the transverse direction. The equations of motion for this system yield

$$\partial_\mu T^{\mu i} = (\varepsilon + p)(\partial_t u^{iz} - \partial_z u^{ti}) + \partial_t t_{v\otimes v}^{ti} + \partial_z t_{v\otimes v}^{zi} = 0 \, , \tag{3.16a}$$

$$\partial_\nu J^{\mu i} = \rho(\partial_t u^{ti} - \partial_z u^{iz}) = 0 \, . \tag{3.16b}$$

These equations can be solved and one finds that the transverse mode's spectrum is governed by

$$(\omega^2 - k^2)\left(1 - \frac{\nu_1}{\varepsilon + p}(\omega^2 - k^2)\right) = 0 \tag{3.17}$$

and one finds that the gapped mode is governed by $1/\nu_1$. One might think that by choosing Landau frame we will be able to get rid of this mode but this turns out not to be the case. By changing the frame choice $u^{\mu\nu} \to u^{\mu\nu} + \delta u_L^{\mu\nu}$ where $\delta u_L^{\mu\nu}$ is in the $v \otimes v$ representation, we find that the appropriate $\delta u_L^{\mu\nu}$ that will remove the $v \otimes v$ part of the stress-energy tensor according to (2.36a) is

$$\delta u_L^{ti} = \frac{\nu_1}{\varepsilon + p}(\partial_t^2 - \partial_z^2)\delta u^{iz} \, , \qquad \delta u_L^{iz} = \frac{\nu_1}{\varepsilon + p}(\partial_t^2 - \partial_z^2)\delta u^{iz} \, . \tag{3.18}$$

This leads to the new additional structure in $s_{v\otimes v}^{\mu\nu} = \rho\delta u_L^{\mu\nu}$ and the new equation of motion in this new frame is

$$\partial_\mu T^{\mu i} = (\varepsilon + p)(\partial_t u^{iz} - \partial_z u^{ti}) = 0 \, , \tag{3.19a}$$

$$\partial_\nu J^{\mu i} = \rho\left(\partial_t u^{ti} - \partial_z u^{iz}\right) + \rho\left(\partial_t \delta u_L^{ti} + \partial_z \delta u_L^{zi}\right) = 0 \, , \tag{3.19b}$$

yielding the same spectrum with the gapped mode in Eq. (3.17). In addition, one can choose the frame in which $\delta\rho = 0$ and $s_{v\otimes v}^{\mu\nu} = 0$ as in [30] and obtain the same spectrum discussed here.

We should note that while the gapped mode is outside the regime of validity of hydrodynamics, it is a mode that generically appears in the gradient expansion of this type. One example that shares close similarity with our construction is the effective theory of long strings in the context of confining flux tubes in gauge theory (see e.g. [69, 70]). In a formulation presented in e.g. [71, 72], the effective theory describes the dynamics of the string displacement (analogous to $\sigma^1$, $\sigma^2$ in our context) which depends on the coordinates along the string (which is a $(t, z)$-plane in this case). The derivative expansion for long-string theory is then performed with $\partial\sigma \sim u$ chosen to be a zero-derivative object and the mass gap is also generated by the higher-order derivative-terms similar to our setup.

## 3.2 Propagation perpendicular to the magnetic field line

As the fluctuations become functions of $(t, x)$, one can show that the fluctuation of $u^{tx}$ decouples and the resulting equation of motion for the propagating mode in the ideal limit is

$$\rho\partial_t\delta\mu - (\varepsilon + p)\partial_x u^{zx} = 0 \ ,$$
$$\rho\partial_t u^{zx} + \partial_x\rho = 0 \ . \tag{3.20}$$

This yields a simple wave equation with the speed $v_M^2 = \rho/(\mu\chi)$, where the susceptibility $\chi = \partial\rho/\partial\mu$. Note that the $v_M^2 = 1$ if we use the equation of state $p \propto \mu^2$. The same spectrum can be obtained with the effective action approach by varying the action w.r.t. $\pi_x$ and $\mathfrak{a}_z$, which are the only two relevant degrees of freedom in this configuration. These quantities can be related by

$$u^{zx} = \partial_t\pi_x \ , \qquad \delta\mu = -2\partial_t\mathfrak{a}_z \ . \tag{3.21}$$

Similar procedure can be carried out with the second-order derivative. Note that, in order to extract the correlation function, we couple the theory to the background metric and gauge field. The solutions for $\pi_x$ and $\mathfrak{a}_z$ can be written schematically as

$$\pi_x = \frac{1}{\mathcal{P}_M^\parallel(\omega, k_x)}\left(\mathcal{A}_\pi^{\mu\nu}\delta g_{\mu\nu} + \mathcal{B}_\pi^{\mu\nu}\delta b_{\mu\nu}\right) \ , \tag{3.22a}$$

$$\mathfrak{a}_z = \frac{1}{\mathcal{P}_M^\parallel(\omega, k_x)}\left(\mathcal{A}_\mathfrak{a}^{\mu\nu}\delta g_{\mu\nu} + \mathcal{B}_\mathfrak{a}^{\mu\nu}\delta b_{\mu\nu}\right) \ , \tag{3.22b}$$

where the coefficients $\mathcal{A}_{\pi,\mathfrak{a}}^{\mu\nu}$, $\mathcal{B}_{\pi,\mathfrak{a}}^{\mu\nu}$ are functions of $\omega$, $k_x$, thermodynamic quantities and transport coefficients. It is also worth noting that only metric and gauge field perturbations that are even under $y \to -y$ enter the above expressions. The important part is the zeroes of the polynomial $\mathcal{P}_M^\parallel$ which encode the spectrum of the propagating mode. This can be written explicitly as

$$\mathcal{P}_M^\parallel = \mu\rho\chi\omega^2 - \rho^2 k^2 - 2\chi(\gamma_4 - \gamma_5)\omega^4 + 2\chi(\gamma_6 + \gamma_7 + \gamma_8)\omega^2 k^2 \ . \tag{3.23}$$

The above polynomial has a wave-like solution which can be written as

$$\omega = \pm v_M k\left[1 + \frac{k^2}{\mu\rho}\left(v_M^2(\gamma_4 - \gamma_5) - (\gamma_6 + \gamma_7 + \gamma_8)\right)\right] + \mathcal{O}(k^4. \tag{3.24}$$

This agrees with the spectrum derived from the linearised constitutive relations in [30], further discussed in appendix C. One may also notice that there is a non-hydrodynamic, gapped mode at $\omega^2 = \pm\mu\rho/2(\gamma_4 - \gamma_5)$. This is the same gapped mode discussed in the previous subsection which lies beyond the regime of validity of hydrodynamics.

In the parity odd channel, the relevant hydrodynamic degree of freedom is $u^{zy} = \partial_t\pi_y$. The correlation functions in this channel have no hydrodynamic poles. This can be seen athrough the solution for $u^{zy}$ in the presence of the background sources which is

$$
u^{zy} = \partial_t\pi_y = \frac{1}{2\mathcal{P}_M^\perp(\omega, k_x)}\left[\left(-\mu\rho + 2\omega^2(\gamma_4 - \gamma_5) - k_x^2(2\gamma_2 + 2\gamma_3 - \gamma_6 - \gamma_7 - \gamma_8)\right)\delta g_{ty}\right.
$$
$$
\left. + \omega k_x(-2(\gamma_2 + \gamma_3) + \gamma_6 + \gamma_7 + \gamma_8)\delta g_{xy} - k_x\beta_1(k_x\delta b_{ty} + \omega\delta b_{xy}) - 2\rho\delta b_{yz}\right].
$$
$$(3.25a)$$

The spectrum encoded in the polynomial

$$
\mathcal{P}_M^\perp(\omega, k_x) = \omega^2(\gamma_4 - \gamma_5) - \frac{1}{2}\mu\rho - \gamma_8 k^2, \tag{3.25b}
$$

indicates that there is only a gapped, non-hydrodynamic mode.

## 3.3 Kubo formulae

In this section, we will utilise the resulting generating one-point functions to extract a list of simple Kubo formulae. The general scheme will be to substitute the solution for $\pi_i$ and $\mathfrak{a}_\mu$ obtained in (3.10), (3.14), (3.22) and (3.25a) into the generating one-point functions so that they can be expressed in terms of the sources. Then, applying the definition of two- and three-point functions in (3.2) to obtain the two point correlation functions. The transport coefficients can be extracted from the derivative with respect to $\omega, k_x$ or $k_z$ of these correlation functions in the limit where $\omega, k^i \to 0$. It is convenient to consider the correlation functions which have no poles.

### 3.3.1 Kubo formulae from two-point functions

Firstly, the one-point functions involving stress-energy tensor expanded up to the order $k_x^2$ and $k_z^2$ are

$$
\lim_{\omega\to 0}\lim_{k_x\to 0}\mathbb{T}^{xy} = \left(-p + k_z^2\left(\gamma_2 - \gamma_3 - \gamma_6 - \gamma_8\right)\right)\delta g_{xy} + \mathcal{O}(k_z^3), \tag{3.26a}
$$

$$
\lim_{\omega\to 0}\lim_{k_x\to 0}\mathbb{T}^{xx} = \frac{p}{2}\delta g_{zz} + \left(-\frac{p}{2} + k_z^2\left(\gamma_1 - \frac{\rho\gamma_1'}{\chi}\right)\right)\delta g_{tt} \tag{3.26b}
$$
$$
- \left(\frac{p}{2} + \frac{\rho^2}{2\chi} + k_z^2\left(\frac{\rho\gamma_3'}{\chi} + \gamma_6 + \gamma_7 + \gamma_8\right)\right)\delta g_{xx}
$$
$$
+ \left(\frac{p}{2} - \frac{\rho^2}{2\chi} - k_z^2\left(\gamma_2 - \gamma_3 + \gamma_7 + \frac{\rho}{\chi}\gamma_3'\right)\right)\delta g_{yy},
$$

$$\lim_{\omega \to 0} \lim_{k_x \to 0} \mathbb{T}^{tx} = \left(p + k_z^2(\gamma_1 + \gamma_3 + \gamma_4)\right)\delta g_{tx} \, , \tag{3.26c}$$

$$\lim_{\omega \to 0} \lim_{k_z \to 0} \mathbb{T}^{ty} = \left(p - \mu\rho - k_x^2(\gamma_2 - \gamma_3 - \gamma_6 - \gamma_8)\right)\delta g_{ty} \, , \tag{3.26d}$$

$$- 2\left(\rho + \frac{k_x^2}{\mu}(2\gamma_2 - \gamma_3 - \gamma_6 - \gamma_8)\right)\delta b_{yz} \, ,$$

$$\lim_{\omega \to 0} \lim_{k_z \to 0} \mathbb{T}^{tt} = \frac{1}{2}\left(\varepsilon + \mu^2\chi - 2k_x^2(\gamma_4 - \gamma_5 - \mu\gamma_3')\right)\delta g_{tt} + \frac{1}{2}\varepsilon\,\delta g_{xx} \tag{3.26e}$$

$$+ \frac{1}{2}\left(\varepsilon - 2k_x^2(\gamma_2 - \mu\gamma_2')\right)\delta g_{yy}$$

$$+ \frac{1}{2}\left(\varepsilon - \mu^2\chi - 2k_x^2(\gamma_1 + \gamma_3 + \gamma_5 - \mu\gamma_3')\right)\delta g_{zz}$$

$$+ (2\mu\chi - k_x^2\beta_1 - 2k_x^2\gamma_3')\delta b_{tz} \, .$$

The two-form current one-point functions relevant for Kubo formulae are

$$\lim_{\omega \to 0} \lim_{k_x \to 0} \mathbb{J}^{tx} = -\rho\delta g_{xz} + 2\left(\frac{\rho}{\mu} + k_z^2\left(\frac{\beta_1}{\mu} - \frac{1}{\mu^2}(\gamma_4 - \gamma_5)\right)\right)\delta b_{tx} \, , \tag{3.27a}$$

$$\lim_{\omega \to 0} \lim_{k_x \to 0} \mathbb{J}^{xy} = 8\alpha k_z^2\delta b_{xy} \, , \tag{3.27b}$$

$$\lim_{\omega \to 0} \lim_{k_z \to 0} \mathbb{J}^{ty} = -\frac{\beta_1}{\mu}k_x^2\delta b_{yz} - 2k_x^2\alpha\delta b_{ty} - \frac{\beta_2}{2}k_x^2\delta g_{yz} \, , \tag{3.27c}$$

$$\lim_{\omega \to 0} \lim_{k_z \to 0} \mathbb{J}^{tz} = \left(\mu\chi - k_x^2\left(\gamma_3' - \frac{\beta_1}{2}\right)\right)\delta g_{tt} + \rho\delta g_{xx} \tag{3.27d}$$

$$+ \left(\rho + 2k_x^2\gamma_2'\right)\delta g_{yy} + \left(-\mu\chi + k_x^2\left(\gamma_3' + \frac{\beta_1}{2}\right)\right)\delta g_{zz} \, ,$$

$$\lim_{\omega \to 0} \lim_{k_z \to 0} \mathbb{J}^{yz} = -\left(\rho - \frac{k_x^2}{\mu}(2\gamma_2 - \gamma_3 - \gamma_6 - \gamma_8)\right)\delta g_{ty} + \frac{1}{2}k_x^2\beta_1\,\delta g_{yz} \tag{3.27e}$$

$$- \frac{1}{\mu}k_x^2\beta_1\,\delta b_{ty} + 2\left(-\frac{\rho}{\mu} + 2\frac{k_x^2}{\mu^2}\gamma_8 + k_x^2\alpha + k_x^2\frac{\beta_2}{\mu}\right)\delta b_{yz} \, .$$

These one point functions can be combined and immediately give us the following seven Kubo formulae. Note that the r.h.s. are evaluated at $k_x = 0$ or $k_z = 0$, after taking the derivatives:

$$\alpha = -\frac{1}{2}\partial_{k_x}^2 G_{JJ}^{ty,ty}(\omega = 0, k_x, k_z = 0) \, , \tag{3.28a}$$

$$\beta_1 = \frac{1}{4}\left(\partial_{k_x}^2 G_{JT}^{tz,tt} + \partial_{k_x}^2 G_{JT}^{tz,zz}\right) \, , \tag{3.28b}$$

$$\beta_2 = -\partial_{k_x}^2 G_{JT}^{ty,yz}(\omega = 0, k_x, k_z = 0) \, , \tag{3.28c}$$

$$\gamma_2 = -\frac{\mu}{2}\partial_{k_x}^2 G_{TJ}^{ty,yz}(\omega = 0, k_x, k_z = 0) + \partial_{k_x}^2 G_{TT}^{ty,ty}(\omega = 0, k_x, k_z = 0) \, , \tag{3.28d}$$

$$\gamma_7 = -\gamma_2 + \frac{1}{4}\partial_{k_z}^2 G_{TT}^{xx,yy}(\omega = 0, k_x = 0, k_z) \, , \tag{3.28e}$$

$$\gamma_8 = \frac{\mu^2}{4}\partial_{k_x}^2 G_{JJ}^{yz,yz}(\omega = 0, k_x, k_z = 0) - \frac{\mu}{2}(\mu\alpha + \beta_2) \, . \tag{3.28f}$$

There are five remaining transport coefficients that cannot be determined by the above two-point functions. These remaining coefficients $\gamma_1$, $\gamma_3$, $\gamma_4$, $\gamma_5$, $\gamma_6$ enter the two-point

functions only in the following linear combinations which cannot be disentangled:

$$\gamma_3' = \frac{1}{8}\left(\partial_{k_x}^2 G_{JT}^{tz,zz}(\omega = 0, k_x, k_z = 0) - \partial_{k_x}^2 G_{JT}^{tz,tt}(\omega = 0, k_x, k_z = 0)\right) , \quad (3.29a)$$

$$\gamma_3 + \gamma_6 = \gamma_2 - \gamma_8 - \frac{1}{2}\partial_{k_x}^2 G_{TT}^{xy,xy}(\omega = 0, k_z = 0, k_x) , \quad (3.29b)$$

$$\gamma_4 - \gamma_5 = \mu\beta_1 - \frac{\mu^2}{2}\partial_{k_z}^2 G_{JJ}^{tx,tx}(\omega = 0, k_x = 0, k_z) , \quad (3.29c)$$

$$\gamma_1 + \gamma_3 + \gamma_4 = \frac{1}{2}\partial_{k_z}^2 G_{TT}^{tx,tx}(\omega = 0, k_x = 0, k_z) . \quad (3.29d)$$

At this point, the assumption about the form of $\gamma_3$ can be of use. If one assumes that the transport coefficient $\gamma_3$ can only depend on $\mu$, one immediately finds that $\gamma_3/|\mu| = \gamma_3'$. One may also relax this assumption and allow the transport coefficient to depend on the additional microscopic length scale $\ell$, namely $\gamma_3/|\mu| = \bar\gamma_3(\mu\ell^2)$. Still, this requires that $\bar\gamma_3$ cannot be singular when $\mu\ell^2 \to 0$ allowing us to fully determine $\gamma_3$ from $\gamma_3'$. Once this is obtained, one can determined $\gamma_6$ using Eq.(3.29b). This leaves us with the two remaining linear combinations $\gamma_1 + \gamma_4$ and $\gamma_4 - \gamma_5$ which can be computed via

$$\gamma_1 + \gamma_4 = -\gamma_3 + \frac{1}{2}\partial_{k_z}^2 G_{TT}^{tx,tx}(\omega = 0, k_x = 0, k_z) , \quad (3.30a)$$

$$\gamma_4 - \gamma_5 = \mu\beta_1 - \frac{\mu^2}{2}\partial_{k_z}^2 G_{JJ}^{tx,tx}(\omega = 0, k_x = 0, k_z) . \quad (3.30b)$$

This indicates that if one manages to find one of the coefficients among $\gamma_1$, $\gamma_4$, $\gamma_5$ we can use the two above equations (3.30a) and (3.30b). Unfortunately, we cannot find any of these transport coefficients individually from the two-point functions.

### 3.3.2  Three-point correlation functions

It turns out that the Kubo formula for $\gamma_1$ can be obtained by considering the three-point function of the stress-energy tensor. As the three-point correlation functions are much more involved than the two-point functions, we will simplify the situation slightly. Firstly, it is sufficient to set all the fields to be only $z-$dependent (namely $\omega = 0$ and $k_x = 0$). Secondly, we wish to turn off the background fields which source the fluctuations $\pi_i$ and $\mathfrak{a}_\mu$ in this channel. Using the solutions in Eqs. (3.10) and (3.14), one can see that the sources for these modes at $\omega, k_x = 0, k_z \neq 0$ are

$$\{\delta g_{tt}, \delta g_{xx}, \delta g_{yy}, \delta g_{zz}, \delta g_{xz}, \delta g_{yz}\} \quad \text{and} \quad \{\delta b_{tx}, \delta b_{ty}, \delta b_{tz}\} . \quad (3.31)$$

As a result, we can turn off these background fields and consistently turn off $\pi_i$ and $\mathfrak{a}_\mu$.

The next step is to express the one-point generating functions $\mathbb{T}^{\mu\nu}$ and $\mathbb{J}^{\mu\nu}$ up to the second order in the (remaining) background field perturbations. The required Kubo formula can be obtained from $\mathbb{T}^{tt}$, which can be written as

$$\mathbb{T}^{tt}(z) = ... + \frac{1}{2}(p - \mu\rho)\,\delta g_{xy}(z)^2 - 2(\alpha - \mu\alpha')\delta b_{xy}(z)^2 + $$
$$+ \frac{1}{2}\left(4\gamma_1 + 2\mu\gamma_3'\right)(\partial_z\delta g_{xy})^2 + (2\gamma_1 + \mu\gamma_3')\delta g_{xy}\partial_z^2\delta g_{xy} + \mathcal{O}(\partial_z^3) , \quad (3.32)$$

where the ellipses denote the terms linear in the perturbations of the background fields and the contact term. Applying the definition of the three-point function (3.2) and Fourier transforming into the momentum space, we find that

$$\mathbb{T}^{tt} = \frac{1}{2}G_{TTT}^{tt,xy,xy}(k_z, q_z)\,\delta g_{xy}(k_z)\delta g_{xy}(q_z) = $$
$$= -(2\gamma_1 + \mu\gamma_3')\left(k_z q_z + \frac{1}{2}\left(q_z^2 + k_z^2\right)\right)\delta g_{xy}(q_z)\delta g_{xy}(k_z) . \quad (3.33)$$

The coefficient $\gamma_3$ can be obtained via the two-point function (3.29a) and therefore, we find that the Kubo formula for $\gamma_1$ is

$$2\gamma_1 = -\gamma_3' - \frac{1}{2}\partial_{k_z}\partial_{q_z}G_{TTT}^{tt,xy,xy}(k_z, q_z) \ . \tag{3.34}$$

Once this is known, one can immediately obtaine the transport coefficients $\gamma_4$ and then $\gamma_5$ directly from Eq. (3.30). We thereby conclude the computation for the Kubo formulae for the second-order transport coefficients, which consist of seven transport coefficients $\alpha$, $\beta_1$, $\beta_2$, $\gamma_2$, $\gamma_3$, $\gamma_7$, $\gamma_8$ obtained solely from two-point functions and three coefficients $\gamma_1$, $\gamma_4$, $\gamma_5$ which require one three-point function.

# 4  Applications: Force-free Electrodynamics and the acceleration by a magnetosphere

In this section, we will discuss how the second-order derivative-corrections improve the description of the conventional FFE. The most transparent way to compare the two setups is to look at the effective action. Firstly, the FFE action can be written as [39]

$$\mathcal{L}_{FFE} = -\frac{1}{4}\left(\partial_\mu\sigma^1\partial_\nu\sigma^2 - (1 \leftrightarrow 2)\right)^2 \ , \tag{4.1}$$

which is nothing but $F^2 = F_{\mu\nu}F^{\mu\nu}$ when the field strength is written as $F_{\mu\nu} = 2\partial_{[\mu}\sigma^1\partial_{\nu]}\sigma^2$. There is no one-form $U(1)$ phase $a_\mu$ in this formulation and thus the higher-form global symmetry is not manifest in the FFE formalism. In the formulation presented in this work, we can see that the Lagrangian in (4.1) is nothing but $\mathcal{L} = \frac{1}{2}\mu^2$ with the one-form chemical potential $\mu^2$ defined in Eq. (2.22) of section 2.1. With the new hydrodynamic framework, we consistently identify all the possible ways to couple the external charge $j_{\text{external}} = \star db$ , up to the second order in the derivative expansion (via $s^{(\alpha)}$ and $s_i^{(\beta)}$ in (2.28a) and (2.28b), respectively). And there are also nontrivial terms at the higher orders in the derivative expansion. In terms of the above conventional FFE language, the action presented in Section 2.2 can be written (schematically) as

$$\mathcal{L} = \mathcal{L}_{FFE} + \frac{4\gamma_1(F^2)}{F^2}R_{\mu\nu\rho\sigma}\tilde{F}^{\mu\nu}\tilde{F}^{\rho\sigma} + ... \ , \qquad \text{where} \qquad \tilde{F}^{\mu\nu} = \epsilon^{\mu\nu\rho\sigma}\partial_{[\rho}\sigma^1\partial_{\sigma]}\sigma^2 \tag{4.2}$$

and (...) denotes the other ten structures in the effective action. This should come as no surprise since FFE is applicable to a system which is not a free Maxwell theory but a strong dynamical magnetic field coupled to charged matter. The Lagrangian in (4.2), and Section 2.2, should therefore be thought of as the most general effective Lagrangian for such plasma obtained after integrating out the massive degrees of freedom that are not the fluctuations of the string, $\{\sigma^1, \sigma^2\}$ and the degrees of freedom describing the one-form phase $a_\mu$.

As already pointed out in [42], the framework of higher-form symmetries allows us to move away from the limit where the acceleration along the magnetic field line parametrised by $\mathcal{P} \sim \mathbf{E} \cdot \mathbf{B}$ vanishes. In the relativistic notation, this comes from the fact that

$$\mathcal{P} \sim \epsilon_{\mu\nu\rho\sigma}J^{\mu\nu}J^{\rho\sigma} = \rho\,\epsilon_{\mu\nu\rho\sigma}u^{\mu\nu}s_{SO(2)}^{\rho\sigma} \ ,$$

where $s_{SO(2)}^{\mu\nu}$ is the second-order correction to $J^{\mu\nu}$ which transforms as a tensor in the $SO(2)$-representation, see Section 2.3. This enables us to address the regime beyond the simplistic approximation of FFE and has phenomenological consequences as discussed

in the introduction. We shall focus on the two configurations considered in [42]: uniform magnetic field and the Michel monopole solution. The latter is a toy model approximation to the magnetosphere of compact, conducting objects (such as pulsars). This improves the analysis in [42] as we classify all the possible terms that can enter the effective action at the second order. Note also that $s_{SO(2)}^{\mu\nu}$ is the only component of the constitutive relations (2.32) that, by itself, is independent of the frame choice (discussed in Section 2.3).

It turns out that the transport coefficients which govern $s_{SO(2)}^{\mu\nu}$ originate from only three terms in the effective action, namely the term with coefficient $\alpha$ in Eq.(2.28a) and the terms with coefficients $\beta_i$ in Eq.(2.28b), see Appendix D for the expressions. Additionally, in the absence of the external charge represented by $b_{\mu\nu}$, we find that only the term with coefficient $\beta_2$ controls the electric field parallel to the magnetic field line $\mathcal{P} \sim \mathbf{E} \cdot \mathbf{B}$. This result greatly simplifies our analysis and, as a result, we find that **(i)** the uniform magnetic field has $\mathcal{P} = 0$ up to the second order in the derivative expansion and **(ii)** $\mathbf{E} \cdot \mathbf{B}$ for the Michel monopole is non-zero and has the same form as in [42] [12].

## 4.1 Plane wave

In this case, the dynamical variables have the profile as those in the Section 3, namely

$$\sigma^1 = x, \qquad \sigma^2 = y, \qquad a_\mu = \frac{1}{2}\mu \, dzdt \qquad (4.3)$$

The solution is time-independent and, as expected, $s_{SO(2)}^{\mu\nu} = 0$ along with the whole second-order derivative-correction to $J^{\mu\nu}$. One can also study perturbations around this equilibrium solution, like in Section 3. Substituting the solution for the perturbation, both with the propagation along and perpendicular to the magnetic field line yields $s_{SO(2)}^{\mu\nu} = 0$ in the absence of sources for the background fields $g_{\mu\nu}$, $b_{\mu\nu}$. This statement can also be made for a linearised perturbation aligned in any direction w.r.t. the magnetic field line and is consistent with the analysis that used linearised constitutive relations in [30].

## 4.2 Michel monopole

Michel monopole [73] is a toy model for a magnetosphere of a rotating compact object, such as a star or a pulsar, and serves as a starting point for more realistic setups such as a rotating black hole [43]. In the FFE framework (equivalently, the zero derivative case of our setup), this solution is nothing but a rotating monopole whose magnetic flux can be written in the spherical coordinates in the following way (see e.g. [46]):

$$\star J = F = q \sin\theta \, d\theta \wedge d\Big(d\phi - \Omega d(t-r)\Big) , \qquad (4.4)$$

where the current is trivially conserved. Before analysing the effects of the second-order transport in this system, let us pause to discuss its physical implications. To make this solution realistic one typically constructs a magnetic dipole by replacing $F \to F \, \text{sign}[\cos\theta]$ which flips the sign of the monopole charge between the upper and lower hemispheres [13]. The magnetosphere is assumed to be far from the compact object and the spacetime is approximated to simply be the Minkowski space. By replacing the metric to be that of the Kerr black hole and the time coordinate $t$ in Eq. (4.4) to the outgoing Eddington-Finkelstein coordinate $u$, one recovers the Blandford-Znajek solution [43] (see also [46]

---

[12]The authors of [42] considered only a single derivative-correction in the effective action, which is $\nabla^\alpha \varepsilon^{\beta\gamma}(db)_{\alpha\beta\gamma}$, in their notation. In our notation this term translates to $H_{\alpha\beta\gamma}\nabla^\alpha u^{\beta\gamma} = 2(s_2^{(\beta)} - s_3^{(\beta)})$.

[13]This procedure results in a non-zero current along the equator known as *current sheet* [74], see also [46] for discussion in the language of exterior derivatives.

for other solutions of this class). In the effective action language, the Michel monopole solution translates into the following solution [42]:

$$\sigma^1 = \theta \ , \qquad \sigma^2 = \phi - \Omega(t - r) \ , \qquad a = \frac{q}{2r} dt \ . \tag{4.5}$$

In terms of the hydrodynamic variables $\mu$ and $u^{\mu\nu}$, we have the following non-zero components:

$$\mu = q/r^2 \ , \qquad u^{tr} = 1 \ , \qquad u^{t\phi} = -\Omega \ , \qquad u^{r\phi} = -\Omega \ , \tag{4.6}$$

with $u^{\mu\nu} = -u^{\nu\mu}$.

We will now show that the second-order derivative-term $s^{\mu\nu}_{SO(2)}$ is non-zero for this solution with the source $b_{\mu\nu}$ turned off. This results in a non-zero electric field along the "magnetosphere". The parameter $\mathcal{P}$ can be easily computed with the help of the projective properties of $\delta u^{\mu\nu}$. First of all, we assume that the transport coefficients $\alpha$, $\beta_i$ and $\gamma_i$ are small parameters so that the Michel monopole solution in the presence of the second derivative corrections can be written as

$$\mu = \frac{q}{r^2} + \delta\mu(\alpha, \beta, \gamma_i) \ , \qquad u^{\mu\nu} = u^{\mu\nu}_0 + \delta u^{\mu\nu}(\alpha, \beta, \gamma_i) \ , \tag{4.7}$$

where $u^{\mu\nu}_0$ is the Michel monopole solution at the zeroth order in Eq.(4.6) and $\delta u^{\mu\nu}$ is in $v \otimes v$ representation of $SO(1,1) \otimes SO(2)$. One notices immediately that altering the profile of hydrodynamic variables according to (4.7) does not affect the $SO(2)$ component of $J^{\mu\nu}$, see discussion in Section 2.3 (alternatively, one can check this statement directly from non-linear constitutive relations in Appendix D.3). In addition, the wedge product of $\delta u \wedge s_{SO(2)}$ simply vanishes as $\delta u$ has a component in the $SO(2)$ representation. Thus the $\mathcal{P} \sim \mathbf{E} \cdot \mathbf{B}$ at the second order in the derivative expansion is

$$\mathcal{P} \sim \left(\rho_0 + \frac{\partial\rho}{\partial\mu}\delta\mu\right)\epsilon_{\mu\nu\rho\sigma}\left(u^{\mu\nu}_0 + \delta u^{\mu\nu}\right)s^{\rho\sigma}_{SO(2)} = \rho_0\epsilon_{\mu\nu\rho\sigma}u^{\mu\nu}_0\, s^{\rho\sigma}_{SO(2)} + \mathcal{O}(\alpha^2, \beta^2, ...) \tag{4.8}$$

where, $\rho_0 = \partial p/\partial\mu$ obtained at the zeroth order in the derivative expansion and the ellipsis denote the other products of the second-order transport coefficients. All in all, this means that the parameter $\mathcal{P}$ for the Michel monopole at the leading order in the transport coefficients can be obtained from the zeroth-order solution Eq.(4.6).

The above analysis is rather general and we expect the same argument to be applicable to more realistic solutions. Nevertheless, let us return to the Michel monopole solution. The component $s^{\mu\nu}_{SO(2)}$ can be obtained either by varying effective action w.r.t. $b_{\mu\nu}$ and applying the appropriate $SO(2)$ projections or simply read off from the expressions presented in the Appendix D. Contracting it with the $u^{\mu\nu}_0$ and Levi-Civita tensor in the spherical coordinates yields the following expression for the magnetic flux per unit flux density:

$$\mathcal{P}/\rho \sim \frac{4}{r^3}\, q\Omega \left(\frac{\partial\beta_2}{\partial\mu}\right)\cos\theta \ . \tag{4.9}$$

This result has the same form as proposed in [42]. The Kubo formula which relates this transport coefficient to the microscopic correlation function can be found via the two-point correlation function (3.28c). This result implies that the non-zero $\mathbf{E} \cdot \mathbf{B}$ is strongly tied to the existence of an additional length scale $\ell_{\text{micro}}$ in the transport coefficients. In other words, if the transport coefficients $\beta_i$ can only depend on the thermodynamic variables, it will imply that $\partial\beta_i/\partial\mu = 0$. The way out of this conundrum is that there exists an additional length scale $\ell_{\text{miro}}$ so that $\beta_i$ can be a nontrivial function of $|\mu|\ell^2_{\text{micro}}$. This possibility has already been discussed in [42] and what we did here is to point out the precise terms in all possible second-order derivative-structures that are responsible for this.

# 5   Discussion

There are two ways to read this work. The first story can be seen as an investigation of the strong magnetic field limit $\mu/T^2 \gg 1$ in the higher-form symmetry formulation of magnetohydrodynamics of [30]. At the ideal limit, there is a symmetry enhancement corresponding to the $SO(1,1)$ boost along the magnetic field line which alters the hydrodynamic degrees of freedom. If one insists that such a symmetry persists through the higher orders in the derivative expansion one finds that the leading-order corrections come from the second-derivative terms, as all the first-order structures are not invariant under the emergent $SO(1,1)$ symmetry. In addition, it forces the entropy current to vanish identically, making it a genuine non-dissipative theory in hydrodynamic framework. The goal of this work is therefore to classify the leading-order corrections to the "ideal fluid" limit by utilising the framework of hydrodynamic effective action . We then further explore how they affect the correlation functions and provide the Kubo formulae which link the macroscopic EFT to the data from a microscopic theory.

The other way to read this story is through the lens of force free electrodynamics and its application to magnetospheres of astrophysical objects. These systems have the same global symmetries and exist also in the regime where the temperature is negligible compared to the magnetic flux density. The common way to describe these systems strictly implies that the $\mathcal{P} \sim \mathbf{E} \cdot \mathbf{B}$ is zero which contradicts the fact that we observe the energy emission form objects such as pulsars. As proposed in [42], the second-order derivative-corrections to the fluid with one-form global symmetry may provide a path for a more realistic EFT for this family of systems. To this end, using the classification of the second-order transport, we single out the transport coefficients which are responsible for the non-zero $\mathcal{P}$ of the magnetosphere of the Michel monopole solution and their corresponding Kubo formulae. One key result is that the non-zero $\mathbf{E} \cdot \mathbf{B}$ requires the transport coefficients to depend on at least one additional length scales. This came from the second-order terms similar to the one proposed in [42] and we show that there is no other structures at this order in the derivative expansion that affect this process. It would be very interesting to compute these transport coefficients from a known microscopic theory to better understand the role of such length scale as well as compare it with the observed pulsars' spectra in e.g. [47].

As for the open problems and future directions, an interesting exercise would be to pin point the role of transport coefficients in an interesting physical setup. For example, some transport coefficients of the type $\gamma_i$ influence the leading-order corrections to the propagating modes in the uniform magnetic field while the coefficient $\beta_2$ is responsible for $\mathcal{P} > 0$ in the Michel monopole background. It would also be interesting to understand these effects in a more realistic setup such as the magnetosphere of the Kerr black hole, particularly the leading-order corrections to the Blandford-Znajek process [43] and the stability of such solutions [75, 76].

One should keep in mind that the present construction assumes that the theory admits a gradient expansion. While being a standard practice, this is a very strong assumption and is not always valid. For example, in the typical fluid, the thermal fluctuations generate non-analytic terms which invalidate the derivative expansion beyond the first order in $3+1$ dimensions [77, 78]. Fortunately, as the dissipative terms are not allowed by the emergent $SO(1,1)$ symmetry, one may argue that the fluctuations are negligible due to dissipation-fluctuation theorem [14]. Nevertheless, it would be extremely useful if there would be a

---

[14]The origin of this type of non-analytic property can also be traced back to the coupling between Schwinger-Keldysh partners, see e.g. [20]. In our setup, the vanishing of the entropy production implies that such coupling is zero.

different mechanism that breaks the gradient expansion or a way to systematically prove the validity of the gradient expansion for this type of fluids.

# Acknowledgement

We would like to thank J. Armas, A. Brandenberg S. Gralla, T. Harmark, N. Iqbal, A. Jain, N. Mekareeya, V. Puletti, K. Schalm, P. Szpietowski, W. Sybesma and A. Romero-Bermúdez for helpful discussions and especially S. Grozdanov for collaborating at initial stages of the project. We would also like to thank S. Gralla, S. Grozdanov and N. Iqbal for commenting on the manuscript. This research was supported in part by a VICI award (K. Schalm) of the Netherlands Organization for Scientific Research (NWO), by the Netherlands Organization for Scientific Research/Ministry of Science and Education (NWO/OCW), and by the Foundation for Research into Fundamental Matter (FOM). The work of N. P. is supported by Icelandic Research Fund grant 163422-052. N. P. would like to thank Leiden University, Durham University, NORDITA, Niels Bohr Institute, Max Planck Institute for Physics of Complex Systems and University of Amsterdam for their hospitality. His visit at Durham University was supported by COST Action MP1405 (QSPACE).

# A   Useful identities and properties

## A.1   Notation

Here we summarise our notation. For symmetrisation/antisymmetrisation of indices we use round/square brackets on them with an appropriate $\frac{1}{n!}$ factor in front, e.g. $t^{(\mu\nu)} \equiv \frac{1}{2}(t^{\mu\nu} + t^{\nu\mu})$, $t^{[\mu\nu]} \equiv \frac{1}{2}(t^{\mu\nu} - t^{\nu\mu})$. And the angle-brackets denote traceless symmetrisation with respect to the appropriate projector $t^{\langle\mu\nu\rangle} \equiv \frac{1}{2}(\Omega^\mu{}_\alpha \Omega^\nu{}_\beta + \Omega^\nu{}_\alpha \Omega^\mu{}_\beta - \Omega^{\mu\nu}\Omega_{\alpha\beta}) t^{\alpha\beta}$ or $t^{\langle\mu\nu\rangle} \equiv \frac{1}{2}(\Pi^\mu{}_\alpha \Pi^\nu{}_\beta + +\Pi^\nu{}_\alpha \Pi^\mu{}_\beta - \Pi^{\mu\nu}\Pi_{\alpha\beta}) t^{\alpha\beta}$ depending whether $\mu, \nu$ are $SO(1,1)$- or $SO(2)$-indices.

For the derivatives we use two notations, $\nabla_\rho t^{\alpha \cdots}{}_{\beta \dots} = t^{\alpha \cdots}{}_{\beta \dots ;\rho}$ for covariant derivatives (with respect to the metric) and $\partial_\rho t^{\alpha \cdots}{}_{\beta \dots} = t^{\alpha \cdots}{}_{\beta \dots ,\rho}$ for partial derivatives.

## A.2   Projective properties of $u^{\mu\nu}$

The variable $u^{\mu\nu}$ arises from the enhancement of two directions generated by $u^\mu$ and $h^\mu$, which are independent in MHD at non-zero temperature, to a surface with $SO(1,1)$ symmetry. $u^{\mu\nu}$ is an element of the symmetry group of its complement in the antisymmetric representation. We can build a symmetric tensor out of $u^{\mu\nu}$:

$$\Omega^{\mu\nu} = u^{\mu\alpha} u_\alpha{}^\nu , \tag{A.1}$$

which acts as a metric on the $SO(1,1)$-invariant surface. Additionally, the product structure of the symmetry groups in MHD at $T = 0$ means that the metric on the 4-dimensional, background space-time can be decomposed into:

$$g^{\mu\nu} = \Omega^{\mu\nu} + \Pi^{\mu\nu} \;\Rightarrow\; \Pi^{\mu\nu} = g^{\mu\nu} - \Omega^{\mu\nu} , \tag{A.2}$$

where $\Pi^{\mu\nu}$ is the metric in the $SO(2)$-invariant sector. The fact that $u^{\mu\nu}$ belongs purely to the $SO(1,1)$-sector of the theory defines the first constraint on it:

$$\Omega^\mu{}_\alpha u^{\alpha\nu} = u^{\mu\alpha} \Omega_\alpha{}^\nu = u^{\mu\alpha} u_{\alpha\beta} u^{\beta\nu} = u^{\mu\nu} . \tag{A.3}$$

The second constraint on $u^{\mu\nu}$ is its normalisation [15]:

$$u^{\mu\nu}u_{\mu\nu} = -2 . \tag{A.4}$$

The above relation, together with (A.3), implies that $\Omega^{\mu\nu}\Omega_{\mu\nu} = \Omega^\mu{}_\mu = \Pi^{\mu\nu}\Pi_{\mu\nu} = \Pi^\mu{}_\mu = 2$ (in four-dimensional space-time).

### A.2.1  Properties of $\nabla_\rho u^{\mu\nu}$

Now that we know the full set of constraints on $u^{\mu\nu}$ we can calculate derivatives of these constraints and analyse what they imply for $\nabla_\rho u^{\mu\nu}$.

The first constraint we will analyse is the projective property of $u^{\mu\nu}$ (A.3). Its derivative takes the following form:

$$\nabla_\rho u^{\mu\nu} = \Omega^\mu{}_\alpha \nabla_\rho u^{\alpha\nu} + \nabla_\rho u^{\mu\alpha}\Omega_\alpha{}^\nu - u^\mu{}_\alpha u^\nu{}_\beta \nabla_\rho u^{\alpha\beta} . \tag{A.5}$$

Projecting the above equation onto $SO(2)$-sector we find:

$$\Pi^\mu{}_\alpha \Pi^\nu{}_\beta \nabla_\rho u^{\alpha\beta} = 0 . \tag{A.6}$$

We can also contract (A.5) with $u^\rho{}_\mu \Omega_\nu{}^\lambda$:

$$(u^\rho{}_\mu \Omega_\nu{}^\lambda + \Omega^\rho{}_\mu u_\nu{}^\lambda)\nabla_\rho u^{\mu\nu} = 0 , \tag{A.7}$$

which after using the decomposition:

$$\nabla_\rho u^{\mu\nu} = \Omega^\mu{}_\alpha \Omega^\nu{}_\beta \mathcal{A}_\rho{}^{\alpha\beta} + \Pi^\mu{}_\alpha \Pi^\nu{}_\beta \mathcal{B}_\rho{}^{\alpha\beta} + (\Omega^\mu{}_\alpha \Pi^\nu{}_\beta + \Pi^\mu{}_\alpha \Omega^\nu{}_\beta)\mathcal{C}_\rho{}^{\alpha\beta} \tag{A.8}$$

gives us:

$$\Omega^\mu{}_\alpha \Omega^\nu{}_\beta \nabla_\rho u^{\alpha\beta} = 0 . \tag{A.9}$$

So the constraints (A.9) and (A.6) together imply that:

$$(\Omega^\mu{}_\alpha \Pi^\nu{}_\beta + \Pi^\mu{}_\alpha \Omega^\nu{}_\beta)\nabla_\rho u^{\alpha\beta} = \nabla_\rho u^{\mu\nu} , \tag{A.10}$$

which means that $\nabla_\rho u^{\mu\nu}$ belongs, in the last two indices, to the mixed, vector-vector part of $SO(1,1)\otimes SO(2)$. We will denote this shortly as $\nabla_\rho u^{\mu\nu} \in (v \otimes v)^{\mu\nu}$. Similar derivation can also be made for a perturbation $\delta u^{\mu\nu}$ with *fixed* background fields $g_{\mu\nu}$ and $b_{\mu\nu}$ which shares the same projective property.

The derivative of the norm (A.4), $u_{\mu\nu}\nabla_\rho u^{\mu\nu} = 0$, is a trivial consequence of the fact that $\nabla_\rho u^{\mu\nu} \in (v \otimes v)^{\mu\nu}$ so it does not generate a new constraint.

### A.2.2  Properties of $\nabla_\sigma \nabla_\rho u^{\mu\nu}$

In the case of the second derivative of $u^{\mu\nu}$ we proceed in the same way as in the previous section. We first calculate the second derivative of the constraint (A.3) and using the decomposition of $\nabla_\sigma \nabla_\rho u^{\mu\nu}$ in the last two indices into the three sectors of the $SO(1,1)\otimes SO(2)$ symmetry group:

$$\nabla_\sigma \nabla_\rho u^{\mu\nu} = \Omega^\mu{}_\alpha \Omega^\nu{}_\beta \mathcal{A}_{\sigma\rho}{}^{\alpha\beta} + \Pi^\mu{}_\alpha \Pi^\nu{}_\beta \mathcal{B}_{\sigma\rho}{}^{\alpha\beta} + (\Omega^\mu{}_\alpha \Pi^\nu{}_\beta + \Pi^\mu{}_\alpha \Omega^\nu{}_\beta)\mathcal{C}_{\sigma\rho}{}^{\alpha\beta} \tag{A.11}$$

we find the constraints on the tensors $\mathcal{A}_{\sigma\rho}{}^{\alpha\beta}$, $\mathcal{B}_{\sigma\rho}{}^{\alpha\beta}$ and $\mathcal{C}_{\sigma\rho}{}^{\alpha\beta}$. In this case all three of these tensors are non-zero but the first two can be rewritten in terms of products of $\nabla_\rho u^{\mu\nu}$

---

[15]This normalisation agrees with $u^{\mu\nu} \equiv 2u^{[\mu}h^{\nu]}$.

and the antisymmetric part $\mathcal{C}_{[\sigma\rho]}{}^{\alpha\beta}$ of the last one is controlled by the curvature terms only:

$$\mathcal{A}_{\sigma\rho}{}^{\mu\nu} = 2u^{[\mu|\alpha}\Omega^{|\nu]\beta}\nabla_\sigma u_\alpha{}^\lambda \nabla_\rho u_{\beta\lambda} \,, \tag{A.12}$$

$$\mathcal{B}_{\sigma\rho}{}^{\mu\nu} = -2u^{\alpha\beta}\nabla_\sigma u^{[\mu}{}_\alpha \nabla_\rho u^{\nu]}{}_\beta \,, \tag{A.13}$$

$$2\mathcal{C}_{[\sigma\rho]}{}^{\alpha\beta} = R_{\sigma\rho}{}^{\alpha\gamma}u_\gamma{}^\beta + R_{\sigma\rho}{}^{\beta\gamma}u^\alpha{}_\gamma \,. \tag{A.14}$$

This means that only $\nabla_{(\sigma}\nabla_{\rho)}u^{\mu\nu} \in (v \otimes v)^{\mu\nu}$ contributes a new, independent tensor structure at the second order in derivatives.

The second derivative of the norm (A.4), the same as in the case of its first derivative, does not generate new constraints. The second derivative of the norm is trivially satisfied when we apply the projective properties of $\nabla_\sigma \nabla_\rho u^{\mu\nu}$ and $\nabla_\rho u^{\mu\nu}$.

## A.3  Projective properties of $H_{\alpha\beta\gamma}$

As in the case of $u^{\mu\nu}$, we can also write down constraints on $H_{\alpha\beta\gamma}$ coming from its projective properties. These properties come from the fact that $H_{\alpha\beta\gamma}$ is antisymmetric in its three indices and both projectors $\Omega^{\mu\nu}$ and $\Pi^{\mu\nu}$ live in a two-dimensional submanifolds of the four-dimensional space-time. This means that there exists a set of coordinates in which $\Omega^{\mu\nu}$ and $\Pi^{\mu\nu}$ are non-zero only if their indices take values in a two-coordinate subset (different for each projector) of the four coordinates describing the full space-time. And this implies that:

$$\Omega^{\mu\alpha}\Omega^{\nu\beta}\Omega^{\lambda\gamma}H_{\alpha\beta\gamma} = 0 \,, \tag{A.15}$$

$$\Pi^{\mu\alpha}\Pi^{\nu\beta}\Pi^{\lambda\gamma}H_{\alpha\beta\gamma} = 0 \,, \tag{A.16}$$

because there will always be a pair of repeated indices on $H_{\alpha\beta\gamma}$ [16].

We can also take derivatives of the above constraints to obtain the projective properties of the derivatives of $H_{\alpha\beta\gamma}$. For $\nabla_\rho H_{\alpha\beta\gamma}$ we find:

$$\Omega^{\mu\alpha}\Omega^{\nu\beta}\Omega^{\lambda\gamma}\nabla_\rho H_{\alpha\beta\gamma} = 2H_{\alpha\beta\gamma}u^{[\mu|\delta}\Pi^{\alpha\sigma}\Omega^{|\nu]\gamma}\Omega^{\lambda\beta}\nabla_\rho u_{\delta\sigma} - H_{\alpha\beta\gamma}u^{\lambda\delta}\Pi^{\alpha\sigma}\Omega^{\mu\beta}\Omega^{\nu\gamma}\nabla_\rho u_{\delta\sigma} \,, \tag{A.17}$$

$$\Pi^{\mu\alpha}\Pi^{\nu\beta}\Pi^{\lambda\gamma}\nabla_\rho H_{\alpha\beta\gamma} = 2H_{\alpha\beta\gamma}u^{\delta\alpha}\Pi^{\lambda\beta}\Pi^{[\mu|\sigma}\Pi^{|\nu]\gamma}\nabla_\rho u_{\delta\sigma} - H_{\alpha\beta\gamma}u^{\delta\alpha}\Pi^{\lambda\sigma}\Pi^{\mu\beta}\Pi^{\nu\gamma}\nabla_\rho u_{\delta\sigma} \,. \tag{A.18}$$

## A.4  Jacobi identities for $u^{\mu\nu}$

Apart from its normalisation (A.4) and the projective property (A.3), the variable $u^{\mu\nu}$ satisfies also Jacobi identities. This can be understood as either a property of the antisymmetric product which defines $u^{\mu\nu}$ in terms of $u^\mu$ and $h^\mu$ ($u^{\mu\nu} = 2u^{[\mu}h^{\nu]}$) or as a property of the antisymmetric representation of $SO(2)$, which $u^{\mu\nu}$ is.

The lowest-order Jacobi identity for $u^{\mu\nu}$ takes the following form:

$$3u^{[\alpha\beta}u^{\gamma]\lambda} = u^{\alpha\beta}u^{\gamma\lambda} + u^{\gamma\alpha}u^{\beta\lambda} + u^{\beta\gamma}u^{\alpha\lambda} = 0 \,. \tag{A.19}$$

Contracting the above with $u_\lambda{}^\mu$ gives:

$$u^{\alpha\beta}\Omega^{\gamma\mu} + u^{\gamma\alpha}\Omega^{\beta\mu} + u^{\beta\gamma}\Omega^{\alpha\mu} = 0 \,, \tag{A.20}$$

---

[16]Constraint (A.15) is always true as the $SO(1,1)$-sector described by $\Omega^{\mu\nu}$ is always two-dimensional. However, the orthogonal sector characterised by $\Pi^{\mu\nu}$ has dimension $d - 2$ in $d$-dimensional space-time so constraint (A.16) does not exist in higher dimensions than $d = 4$.

and contracting this identity further with $u_\beta{}^\rho$ and changing $\rho \to \beta$:

$$\Omega^{\alpha\beta}\Omega^{\gamma\mu} - u^{\gamma\alpha}u^{\beta\mu} - \Omega^{\beta\gamma}\Omega^{\alpha\mu} = 0 \ . \tag{A.21}$$

There are also higher-order Jacobi identities [79] involving products of more variables $u^{\mu\nu}$ and more indices interchanged cyclically, as well as more possible contractions of them. At the level of the products of three $u$-variables we have [17]:

$$u^{[\alpha\beta}u^{\gamma][\delta}u^{\lambda]\mu} = 0 \ , \tag{A.22}$$

together with all the possible contractions, similarly to (A.20) and (A.21).

We will not analyse any higher-order Jacobi identities here as they are not needed for our study of 2nd-order MHD. But it should be kept in mind that corrections at three-derivative-order and higher may require them. It is also important to mention that derivatives of these Jacobi identities do not generate any new constraints. This statement was only checked at the level of one and two derivatives. But it seems natural that this statement would generalise to any number of derivatives.

The power of Jacobi identities comes from the fact that they can be treated like projectors that annihilate any tensor that they are projected onto. This way contracted with any tensor structure they produce many new identities for those tensor structures. This is the most involved part of the process of generating lists of independent scalars, vectors and tensors at any derivative-order in MHD at zero-temperature.

## A.5 Variations of hydrodynamic variables

Following their definitions in terms of the massless degrees of freedom in terms of $\sigma^i, \mathfrak{a}_\mu, g_{\mu\nu}$ and $b_{\mu\nu}$ in Section 2.1, the variations of the physical quantities under metric and gauge field perturbations are [18]

$$\delta u^{\mu\nu} = -\frac{1}{2}u^{\mu\nu}\Omega^{\alpha\beta}\delta g_{\alpha\beta} \ , \tag{A.23a}$$

$$\delta\mu = -\frac{1}{2}\mu\Omega^{\alpha\beta}\delta g_{\alpha\beta} + u^{\alpha\beta}\delta b_{\alpha\beta} \ , \tag{A.23b}$$

$$\delta\Omega^{\mu\nu} = u^{\mu\alpha}\delta g_{\alpha\beta}u^{\beta\nu} - \Omega^{\mu\nu}\Omega^{\alpha\beta}\delta g_{\alpha\beta} \tag{A.23c}$$

and $\delta\Pi^{\mu\nu} = \delta g^{\mu\nu} - \delta\Omega^{\mu\nu}$. Here we use the notation $\delta g^{\mu\nu} = -g^{\mu\alpha}g^{\nu\beta}\delta g_{\alpha\beta}$. We would like to emphasise the role of the spacetime index which is crucial to the constitutive relations derived from the action. Unlike the ordinary fluid four-velocity $u^\mu$ where $\delta u^\mu = -g^{\mu\nu}\delta u_\nu$ (see e.g. [11]), we have

$$\delta u_\mu{}^\nu = -\frac{1}{2}(u_\mu{}^\alpha\omega^{\nu\beta} + \omega_\mu{}^\alpha u^{\nu\beta})\delta g_{\alpha\beta} - \Pi_\mu{}^\alpha u^{\nu\beta}\delta g_{\alpha\beta}, \tag{A.24a}$$

$$\delta u_{\mu\nu} = \frac{1}{2}u_{\mu\nu}\omega^{\alpha\beta}\delta g_{\alpha\beta} + (u_\mu{}^\alpha\Pi_\nu{}^\beta - \Pi_\mu{}^\alpha u_\nu{}^\beta)\delta g_{\alpha\beta}. \tag{A.24b}$$

These variations with respect to the background fields are consistent with the variations of $u^\mu$ and $h^\mu$ at finite temperature that were obtained in [30, 41].

---

[17] This is the only antisymmetrization of indices on $u^{\alpha\beta}u^{\gamma\delta}u^{\lambda\mu}$ that gives a new, independent identity. Others reduce to $u^{[\alpha\beta}u^{\gamma]\lambda} = 0$.

[18] The results we obtain here are a generalization of the constraints on the derivatives of $u^{\mu\nu}$ as the covariant derivative of the background metric $g_{\mu\nu}$ vanishes but variations can have a non-vanishing effect on the metric.

# B  Computations details

## B.1  More details on the classification of the second-order terms in the effective action

In this section, we will further elaborate on the algorithm we use to generate the second-order derivative-terms in the effective action. The code which implements these steps, `Scalars.nb`, can be found in [59]. Let us recall all the structures with two derivatives of the hydrodynamic variables in Eq.(2.27):

$$\{\nabla_\rho\mu\nabla_\sigma\mu,\ \nabla_\rho\nabla_\sigma\mu,\ \nabla_\rho\mu\nabla_\sigma u^{\mu\nu},\ \nabla_\rho\nabla_\sigma u^{\mu\nu},\ \nabla_\rho u^{\mu\nu}\nabla_\sigma u^{\alpha\beta}, \tag{B.1}$$
$$\nabla_\rho\mu\ H_{\alpha\beta\gamma},\ \nabla_\rho u^{\mu\nu}H_{\alpha\beta\gamma},\ H_{\alpha\beta\gamma}H_{\rho\sigma\lambda},\ \nabla_\rho H_{\alpha\beta\gamma},\ R_{\alpha\beta\gamma\delta}\}\ .$$

All of the above terms have an even number of indices so all of them can be contracted into scalars. In order to reduce the number of scalars we generate by considering all the possible contractions of the terms in (B.1) with zeroth-order terms

$$\{u^{\mu\nu},\ \Omega^{\mu\nu},\ \Pi^{\mu\nu}\} \tag{B.2}$$

we will only take the $(v\otimes v)^{\mu\nu}$-part of $\nabla_\rho u^{\mu\nu}$ and $\nabla_{(\rho}\nabla_{\sigma)}u^{\mu\nu}$ because we know from sections A.2.1 and A.2.2 that those are the only independent contributions to (B.1). Furthermore, because we are only considering contractions into scalars here we will project onto the $(v\otimes v)$-sector with $\omega^\mu{}_\alpha\Pi^\nu{}_\beta$ instead of the full projector $(\omega^\mu{}_\alpha\Pi^\nu{}_\beta+\Pi^\mu{}_\alpha\omega^\nu{}_\beta)$ as both parts of that projector generate the same scalars, up to a sign.

After obtaining all the different scalars from all the possible contractions we use the Jacobi identities, as presented in section A.4, to eliminate scalars related by such identities. By applying Jacobi identities as projectors onto (B.1) we generate a set of identities for scalars at the second order in derivatives. And we use these identities, together with the projective properties of $H_{\alpha\beta\gamma}$ from section A.3, to reduce the list of scalars obtained from all the possible contractions of (2.27) down to the following 27 scalars:

$$
\begin{array}{ll}
R_{\alpha\gamma\beta\delta}u^{\alpha\beta}u^{\gamma\delta} & R_{\alpha\gamma\beta\delta}\Pi^{\alpha\beta}\Pi^{\gamma\delta} \\[4pt]
R_{\alpha\gamma\beta\delta}\Pi^{\alpha\beta}\omega^{\gamma\delta} & H_{\alpha\gamma\lambda}H_{\beta\delta\kappa}\Pi^{\alpha\beta}\Pi^{\gamma\delta}\omega^{\lambda\kappa} \\[4pt]
H_{\beta\gamma\delta}u^{\beta\gamma}\Pi_\alpha{}^\delta\nabla^\alpha\mu & H_{\alpha\beta\delta}u^{\alpha\beta}\Pi^{\gamma\delta}\omega^{\lambda\kappa}\nabla_\kappa u_{\gamma\lambda} \\[4pt]
u^{\alpha\beta}\Pi^{\gamma\delta}\omega^{\lambda\kappa}\nabla_\kappa u_{\delta\lambda}\nabla_\beta u_{\alpha\gamma} & \Pi_{\alpha\beta}\nabla^\alpha\mu\nabla^\beta\mu \\[4pt]
\omega_{\alpha\beta}\nabla^\alpha\mu\nabla^\beta\mu & \Pi_{\alpha\beta}\nabla^\beta\nabla^\alpha\mu \\[4pt]
\omega_{\alpha\beta}\nabla^\beta\nabla^\alpha\mu & H_{\beta\delta\kappa}u^{\alpha\beta}u^{\gamma\delta}\Pi^{\lambda\kappa}\nabla_\gamma u_{\alpha\lambda} \\[4pt]
H_{\beta\delta\kappa}\Pi^{\alpha\beta}\Pi^{\gamma\delta}\omega^{\lambda\kappa}\nabla_\gamma u_{\alpha\lambda} & u^{\alpha\beta}u^{\gamma\delta}\Pi^{\lambda\kappa}\nabla_\beta u_{\delta\kappa}\nabla_\gamma u_{\alpha\lambda} \\[4pt]
\Pi^{\alpha\beta}\Pi^{\gamma\delta}\omega^{\lambda\kappa}\nabla_\beta u_{\delta\kappa}\nabla_\gamma u_{\alpha\lambda} & u^{\beta\gamma}\Pi_\alpha{}^\delta\nabla^\alpha\mu\nabla_\gamma u_{\beta\delta} \\[4pt]
\Pi^{\beta\gamma}\omega_\alpha{}^\delta\nabla^\alpha\mu\nabla_\gamma u_{\beta\delta} & u^{\alpha\beta}\Pi^{\gamma\delta}\nabla_\delta H_{\alpha\beta\gamma} \\[4pt]
\Pi^{\alpha\beta}\omega^{\gamma\delta}\omega^{\lambda\kappa}\nabla_\kappa u_{\beta\lambda}\nabla_\delta u_{\alpha\gamma} & \Pi^{\alpha\beta}\Pi^{\gamma\delta}\omega^{\lambda\kappa}\nabla_\gamma u_{\alpha\lambda}\nabla_\delta u_{\beta\kappa} \\[4pt]
u_\alpha{}^\beta\Pi^{\gamma\delta}\nabla^\alpha\mu\nabla_\delta u_{\beta\gamma} & \Pi_\alpha{}^\beta\omega^{\gamma\delta}\nabla^\alpha\mu\nabla_\delta u_{\beta\gamma} \\[4pt]
u^{\alpha\beta}u^{\gamma\delta}\Pi^{\lambda\kappa}\nabla_\beta u_{\alpha\lambda}\nabla_\delta u_{\gamma\kappa} & \Pi^{\alpha\beta}\Pi^{\gamma\delta}\omega^{\lambda\kappa}\nabla_\beta u_{\alpha\lambda}\nabla_\delta u_{\gamma\kappa} \\[4pt]
u^{\alpha\beta}\Pi^{\gamma\delta}\nabla_\delta\nabla_\beta u_{\alpha\gamma} & \Pi^{\alpha\beta}\omega^{\gamma\delta}\nabla_\delta\nabla_\beta u_{\alpha\gamma} \\[4pt]
H_{\beta\delta\kappa}u^{\alpha\beta}\Pi^{\gamma\delta}\Pi^{\lambda\kappa}\nabla_\lambda u_{\alpha\gamma} &
\end{array}
\tag{B.3}
$$

Next we apply the leading equations of motion (2.11a)-(2.11b) to further reduce the

number of independent scalars to 14:

$$
\begin{aligned}
&R_{\alpha\gamma\beta\delta}u^{\alpha\beta}u^{\gamma\delta} &\quad& R_{\alpha\gamma\beta\delta}\Pi^{\alpha\beta}\Pi^{\gamma\delta}\\
&R_{\alpha\gamma\beta\delta}\Pi^{\alpha\beta}\omega^{\gamma\delta} &\quad& H_{\alpha\gamma\lambda}H_{\beta\delta\kappa}\Pi^{\alpha\beta}\Pi^{\gamma\delta}\omega^{\lambda\kappa}\\
&H_{\beta\delta\kappa}u^{\alpha\beta}u^{\gamma\delta}\Pi^{\lambda\kappa}\nabla_\gamma u_{\alpha\lambda} &\quad& H_{\beta\delta\kappa}\Pi^{\alpha\beta}\Pi^{\gamma\delta}\omega^{\lambda\kappa}\nabla_\gamma u_{\alpha\lambda}\\
&u^{\alpha\beta}u^{\gamma\delta}\Pi^{\lambda\kappa}\nabla_\beta u_{\delta\kappa}\nabla_\gamma u_{\alpha\lambda} &\quad& \Pi^{\alpha\beta}\Pi^{\gamma\delta}\omega^{\lambda\kappa}\nabla_\beta u_{\delta\kappa}\nabla_\gamma u_{\alpha\lambda} &\quad& \text{(B.4)}\\
&u^{\alpha\beta}\Pi^{\gamma\delta}\nabla_\delta H_{\alpha\beta\gamma} &\quad& \Pi^{\alpha\beta}\Pi^{\gamma\delta}\omega^{\lambda\kappa}\nabla_\gamma u_{\alpha\lambda}\nabla_\delta u_{\beta\kappa}\\
&u^{\alpha\beta}u^{\gamma\delta}\Pi^{\lambda\kappa}\nabla_\beta u_{\alpha\lambda}\nabla_\delta u_{\gamma\kappa} &\quad& \Pi^{\alpha\beta}\Pi^{\gamma\delta}\omega^{\lambda\kappa}\nabla_\beta u_{\alpha\lambda}\nabla_\delta u_{\gamma\kappa}\\
&u^{\alpha\beta}\Pi^{\gamma\delta}\nabla_\delta\nabla_\beta u_{\alpha\gamma} &\quad& H_{\beta\delta\kappa}u^{\alpha\beta}\Pi^{\gamma\delta}\Pi^{\lambda\kappa}\nabla_\lambda u_{\alpha\gamma}
\end{aligned}
$$

In the last step we consider each scalar in the above list multiplied by an arbitrary function of the chemical potential $f(\mu)$ and integrate them by parts to eliminate the second derivatives of $u^{\mu\nu}$ and the derivatives of $H_{\alpha\beta\gamma}$. And once again applying the leading equations of motion (2.11a)-(2.11b) we find 12 independent scalars:

$$
\begin{aligned}
&R_{\alpha\gamma\beta\delta}u^{\alpha\beta}u^{\gamma\delta} &\quad& R_{\alpha\gamma\beta\delta}\Pi^{\alpha\beta}\Pi^{\gamma\delta}\\
&R_{\alpha\gamma\beta\delta}\Pi^{\alpha\beta}\omega^{\gamma\delta} &\quad& H_{\alpha\gamma\lambda}H_{\beta\delta\kappa}\Pi^{\alpha\beta}\Pi^{\gamma\delta}\omega^{\lambda\kappa}\\
&H_{\beta\delta\kappa}u^{\alpha\beta}u^{\gamma\delta}\Pi^{\lambda\kappa}\nabla_\gamma u_{\alpha\lambda} &\quad& H_{\beta\delta\kappa}\Pi^{\alpha\beta}\Pi^{\gamma\delta}\omega^{\lambda\kappa}\nabla_\gamma u_{\alpha\lambda}\\
&u^{\alpha\beta}u^{\gamma\delta}\Pi^{\lambda\kappa}\nabla_\beta u_{\delta\kappa}\nabla_\gamma u_{\alpha\lambda} &\quad& \Pi^{\alpha\beta}\Pi^{\gamma\delta}\omega^{\lambda\kappa}\nabla_\beta u_{\delta\kappa}\nabla_\gamma u_{\alpha\lambda} &\quad& \text{(B.5)}\\
&\Pi^{\alpha\beta}\Pi^{\gamma\delta}\omega^{\lambda\kappa}\nabla_\gamma u_{\alpha\lambda}\nabla_\delta u_{\beta\kappa} &\quad& u^{\alpha\beta}u^{\gamma\delta}\Pi^{\lambda\kappa}\nabla_\beta u_{\alpha\lambda}\nabla_\delta u_{\gamma\kappa}\\
&\Pi^{\alpha\beta}\Pi^{\gamma\delta}\omega^{\lambda\kappa}\nabla_\beta u_{\alpha\lambda}\nabla_\delta u_{\gamma\kappa} &\quad& H_{\beta\delta\kappa}u^{\alpha\beta}\Pi^{\gamma\delta}\Pi^{\lambda\kappa}\nabla_\lambda u_{\alpha\gamma}
\end{aligned}
$$

Eliminating the last scalar in the above list because it is not $\mathcal{CPT}$-invariant we arrive at scalars in (2.28a)-(2.28c).

There is also the possibility of using the Levi-Civita symbol $\epsilon_{\alpha\beta\gamma\delta}$ in constructing the second-order scalars. This would produce the following scalars in addition to (B.5):

$$
\begin{aligned}
&\epsilon_{\gamma\delta\lambda\nu}R_{\alpha\beta\kappa\mu}u^{\alpha\beta}u^{\gamma\delta}\Pi^{\kappa\lambda}\Pi^{\mu\nu} &\quad& \epsilon_{\gamma\delta\lambda\nu}R_{\alpha\kappa\beta\mu}u^{\alpha\beta}u^{\gamma\delta}\Pi^{\kappa\lambda}\Pi^{\mu\nu}\\
&\epsilon_{\gamma\delta\nu\sigma}H_{\alpha\kappa\mu}H_{\beta\lambda\rho}u^{\alpha\beta}u^{\gamma\delta}\Pi^{\kappa\lambda}\Pi^{\mu\nu}\Pi^{\rho\sigma} &\quad& \epsilon_{\gamma\delta\nu\sigma}H_{\alpha\beta\kappa}H_{\lambda\mu\rho}u^{\alpha\beta}u^{\gamma\delta}\Pi^{\kappa\lambda}\Pi^{\mu\nu}\Pi^{\rho\sigma}\\
&\epsilon_{\kappa\lambda\nu\sigma}H_{\beta\delta\rho}u^{\alpha\beta}u^{\gamma\delta}u^{\kappa\lambda}\Pi^{\mu\nu}\Pi^{\rho\sigma}\nabla_\gamma u_{\alpha\mu} &\quad& \epsilon_{\alpha\beta\lambda\nu}H_{\kappa\mu\sigma}u^{\alpha\beta}\Pi^{\gamma\delta}\Pi^{\kappa\lambda}\Pi^{\mu\nu}\omega^{\rho\sigma}\nabla_\delta u_{\gamma\rho}\\
&\epsilon_{\alpha\beta\lambda\nu}H_{\delta\mu\sigma}u^{\alpha\beta}\Pi^{\gamma\delta}\Pi^{\kappa\lambda}\Pi^{\mu\nu}\omega^{\rho\sigma}\nabla_\kappa u_{\gamma\rho} &\quad& \epsilon_{\gamma\delta\nu\sigma}H_{\beta\mu\rho}u^{\alpha\beta}u^{\gamma\delta}\Pi^{\kappa\lambda}\Pi^{\mu\nu}\Pi^{\rho\sigma}\nabla_\lambda u_{\alpha\kappa}\\
&\epsilon_{\gamma\delta\nu\sigma}H_{\beta\lambda\rho}u^{\alpha\beta}u^{\gamma\delta}\Pi^{\kappa\lambda}\Pi^{\mu\nu}\Pi^{\rho\sigma}\nabla_\mu u_{\alpha\kappa} &\quad& \epsilon_{\gamma\delta\lambda\sigma}H_{\beta\nu\rho}u^{\alpha\beta}u^{\gamma\delta}\Pi^{\kappa\lambda}\Pi^{\mu\nu}\Pi^{\rho\sigma}\nabla_\mu u_{\alpha\kappa} &\quad& \text{(B.6)}\\
&\epsilon_{\alpha\beta\lambda\nu}u^{\alpha\beta}\Pi^{\gamma\delta}\Pi^{\kappa\lambda}\Pi^{\mu\nu}\omega^{\rho\sigma}\nabla_\delta u_{\gamma\rho}\nabla_\mu u_{\kappa\sigma} &\quad& \epsilon_{\alpha\beta\delta\nu}u^{\alpha\beta}\Pi^{\gamma\delta}\Pi^{\kappa\lambda}\Pi^{\mu\nu}\omega^{\rho\sigma}\nabla_\kappa u_{\gamma\rho}\nabla_\mu u_{\lambda\sigma}\\
&\epsilon_{\gamma\delta\lambda\sigma}u^{\alpha\beta}u^{\gamma\delta}\Pi^{\kappa\lambda}\Pi^{\mu\nu}\Pi^{\rho\sigma}\nabla_\mu u_{\alpha\kappa}\nabla_\nu u_{\beta\rho} &\quad& \epsilon_{\gamma\delta\lambda\nu}u^{\alpha\beta}u^{\gamma\delta}\Pi^{\kappa\lambda}\Pi^{\mu\nu}\omega^{\rho\sigma}\nabla_\beta u_{\mu\sigma}\nabla_\rho u_{\alpha\kappa}\\
&\epsilon_{\gamma\delta\nu\sigma}u^{\alpha\beta}u^{\gamma\delta}\Pi^{\kappa\lambda}\Pi^{\mu\nu}\Pi^{\rho\sigma}\nabla_\mu u_{\alpha\kappa}\nabla_\rho u_{\beta\lambda} &\quad& \epsilon_{\gamma\delta\nu\sigma}u^{\alpha\beta}u^{\gamma\delta}\Pi^{\kappa\lambda}\Pi^{\mu\nu}\Pi^{\rho\sigma}\nabla_\lambda u_{\alpha\kappa}\nabla_\rho u_{\beta\mu}\\
&\epsilon_{\gamma\delta\lambda\sigma}u^{\alpha\beta}u^{\gamma\delta}\Pi^{\kappa\lambda}\Pi^{\mu\nu}\Pi^{\rho\sigma}\nabla_\mu u_{\alpha\kappa}\nabla_\rho u_{\beta\nu}
\end{aligned}
$$

It is important to notice that because the Levi-Civita symbol is totally antisymmetric in all of its four indices, all of them have to take different values. This means that in our separation of indices into the $SO(1,1)$ and $SO(2)$ sectors we have that:

$$
\epsilon^{\mu\nu\rho\sigma} = \Pi^{\mu\alpha}\Pi^{\nu\beta}\Omega^{\rho\gamma}\Omega^{\sigma\delta}\epsilon_{\alpha\beta\gamma\delta} = \frac{1}{2}\Pi^{\mu\alpha}\Pi^{\nu\beta}\Big(\Omega^{\rho\gamma}\Omega^{\sigma\delta} - \Omega^{\sigma\gamma}\Omega^{\rho\delta}\Big)\epsilon_{\alpha\beta\gamma\delta} =
$$
$$
= -\frac{1}{2}\Pi^{\mu\alpha}\Pi^{\nu\beta}u^{\rho\sigma}\epsilon_{\alpha\beta\gamma\delta}u^{\gamma\delta} \; . \tag{B.7}
$$

And this last form of the Levi-Civita symbol is what we used to generate (B.6). It is then straightforward to check that all of the terms in (B.6) are not invariant under charge conjugation $\mathcal{C}$ and the parity $\mathcal{P}$ assigned in appendix B.2.

## B.2 Discrete charges $(\mathcal{C}, \mathcal{P}, \mathcal{T})$ of hydrodynamic variables

In this section, we elaborate on the discrete charge assignment of the hydrodynamic variables. This analysis has already been done for fluid with 1-form global symmetry in [30, 41] and we simply specify this information to the limit where there is an emergent $SO(1,1)$ symmetry. We will also follow the conventions in mentioned papers where all thermodynamic quantities are invariant under all $\mathcal{C}, \mathcal{P}, \mathcal{T}$ symmetries.

A few properties is worth mentioning. Firstly, as the action contains the combination $S \supset \int d^4 x J^{\mu\nu} b_{\mu\nu}$, it implies that $b_{\mu\nu}$ and $J^{\mu\nu}$ must transform in the same way under the discrete symmetries. From this, one can deduce how $u^{\mu\nu}$ transforms via the definition $\mu \sim u^{\mu\nu}(b_{\mu\nu} + ...)$ noting that $\mu$ is chosen to be invariant under all $\mathcal{C}, \mathcal{P}, \mathcal{T}$ transformations. This should be contrasted with the string reparametrisation symmetry $\sigma^i \to \sigma'^i$ with $\det[\partial\sigma'/\partial\sigma] < 0$ which acts effectively as an additional discrete transformation for the hydrodynamic variables (which we denote by $\mathcal{R}$ in the table below). In the latter case, the chemical potential $\mu$, density $\rho$ and $u^{\mu\nu}$ switch sign while the current $J^{\mu\nu}$ does not. The table summarising the transformations of relevant hydrodynamic variables is presented below.

| | $\mathcal{C}$ | $\mathcal{P}$ | $\mathcal{T}$ | $\mathcal{R}$ |
|---|---|---|---|---|
| $\partial_\mu$ | $\partial_\mu$ | $(\partial_0, -\partial_i)$ | $(-\partial_0, \partial_i)$ | $\partial_\mu$ |
| $J^{\mu\nu}$ | $-J^{\mu\nu}$ | $(J^{0i}, -J^{ij})$ | $(-J^{0i}, J^{ij})$ | $J^{\mu\nu}$ |
| $u^{\mu\nu}$ | $-u^{\mu\nu}$ | $(u^{0i}, -u^{ij})$ | $(-u^{0i}, u^{ij})$ | $-u^{\mu\nu}$ |
| $\mu$ | $\mu$ | $\mu$ | $\mu$ | $-\mu$ |
| $H_{\mu\nu\lambda}$ | $-H_{\mu\nu\lambda}$ | $(-H_{0ij}, H_{ijk})$ | $(-H_{0ij}, H_{ijk})$ | $H_{\mu\nu\lambda}$ |

# C Linearised constitutive relations

Using the effective Lagrangian (2.24) we find in the flat-space, flat-gauge-field limit at the linear order in $u^{\mu\nu}$ [19] :

$$\delta\varepsilon = \zeta_{(1,1)} \, u^{\alpha\beta} \Pi^{\gamma\delta} u_{\alpha\gamma,\beta\delta} + \mathcal{O}(\partial^3) \, , \tag{C.1a}$$

$$\delta p = \zeta_{(2)} \, u^{\alpha\beta} \Pi^{\gamma\delta} u_{\alpha\gamma,\beta\delta} + \mathcal{O}(\partial^3) \, , \tag{C.1b}$$

$$\delta\rho = \tilde{\zeta} \, u^{\alpha\beta} \Pi^{\gamma\delta} u_{\alpha\gamma,\beta\delta} + \mathcal{O}(\partial^3) \, , \tag{C.1c}$$

$$t^{\mu\nu}_{SO(1,1)} = -2\eta_{(1,1)} \, u^{\langle\mu|\alpha} \Pi^{\beta\gamma} \Omega^{|\nu\rangle\delta} u_{\alpha\beta,\gamma\delta} + \mathcal{O}(\partial^3) \, , \tag{C.1d}$$

$$t^{\mu\nu}_{SO(2)} = 2\eta_{(2)} \, u^{\alpha\beta} \Pi^{\langle\mu|\gamma} \Pi^{|\nu\rangle\delta} u_{\alpha\delta,\beta\gamma} + \mathcal{O}(\partial^3) \, , \tag{C.1e}$$

$$t^{\mu\nu}_{v\otimes v} = -2u^{(\mu|\alpha} \Pi^{|\nu)\beta} \Big( \nu_0 \, \Pi^{\gamma\delta} u_{\alpha\gamma,\beta\delta} + \nu_1 \, \Omega^{\gamma\delta} u_{\alpha\beta,\gamma\delta} + \nu_2 \, \Pi^{\gamma\delta} u_{\alpha\beta,\gamma\delta} \Big) + \mathcal{O}(\partial^3) \, , \tag{C.1f}$$

$$s^{\mu\nu}_{SO(2)} = 0 + \mathcal{O}(\partial^3) \, , \tag{C.1g}$$

$$s^{\mu\nu}_{v\otimes v} = 2\Omega^{[\mu|\alpha} \Pi^{|\nu]\beta} \Big( \tilde{\nu}_0 \, \Pi^{\gamma\delta} u_{\alpha\gamma,\beta\delta} + \tilde{\nu}_1 \, \Omega^{\gamma\delta} u_{\alpha\beta,\gamma\delta} + \tilde{\nu}_2 \, \Pi^{\gamma\delta} u_{\alpha\beta,\gamma\delta} \Big) + \mathcal{O}(\partial^3) \, . \tag{C.1h}$$

The fact that we obtain $\delta\rho, s^{\mu\nu}_{v\otimes v} \neq 0$ in the linearised theory shows that we are working in a different frame than [30]. The transport coefficients $\zeta$, $\eta$ and $\nu$ depend on the chemical potential $\mu$ and they can be related to the coefficients $\alpha$, $\beta_i$ and $\gamma_i$ in the effective

---

[19]To clarify, the linear order is the part that survives an expansion of the effective degrees of freedom around a constant background, $u^{\mu\nu} \to u^{\mu\nu}_0 + \delta u^{\mu\nu}$ and only contains terms up to the first order in $\delta u$. The linearisation and taking the flat-space, flat-gauge-field limit are performed on the most general $T^{\mu\nu}$ and $J^{\mu\nu}$ so on the results in appendix D. This is because the variations of the action and accompanying them integrations by parts can generate linear terms in the flat limit from terms in the action that are nonlinear and/or contain curvature or gauge field.

Lagrangian (2.24) via:

$$\varepsilon(\mu) = -p(\mu) + \mu p'(\mu) , \qquad \rho(\mu) = p'(\mu) , \tag{C.2a}$$

$$\zeta_{(1,1)} = -\frac{\rho(\mu)\gamma_1'(\mu)}{\rho'(\mu)} + 2\gamma_1(\mu) + 2\gamma_5(\mu) - \mu\gamma_3'(\mu) + \gamma_3(\mu) - \gamma_4(\mu) , \tag{C.2b}$$

$$\zeta_{(2)} = -\gamma_6(\mu) - 2\gamma_7(\mu) - \frac{\rho(\mu)\gamma_3'(\mu)}{\rho'(\mu)} + \gamma_3(\mu) - \gamma_8(\mu) + \mu\gamma_2'(\mu) - 2\gamma_2(\mu), \tag{C.2c}$$

$$\tilde{\zeta} = -\frac{1}{2}\beta_1(\mu) , \tag{C.2d}$$

$$\eta_{(1,1)} = -\frac{\rho(\mu)\gamma_1'(\mu)}{\rho'(\mu)} + 2\gamma_1(\mu) + \gamma_3(\mu) + \gamma_4(\mu) , \tag{C.2e}$$

$$\eta_{(2)} = -\gamma_6(\mu) - \gamma_3(\mu) - \gamma_8(\mu) - \mu\gamma_2'(\mu) + 2\gamma_2(\mu) , \tag{C.2f}$$

$$\nu_0 = -\gamma_6(\mu) + \frac{\rho(\mu)\gamma_3'(\mu)}{\rho'(\mu)} - \gamma_3(\mu) + \gamma_8(\mu) + 2\gamma_2(\mu) , \tag{C.2g}$$

$$\nu_1 = -2\gamma_4(\mu) + 2\gamma_5(\mu) , \tag{C.2h}$$

$$\nu_2 = \gamma_6(\mu) + \gamma_3(\mu) - \gamma_8(\mu) - 2\gamma_2(\mu) , \tag{C.2i}$$

$$\tilde{\nu}_0 = -\frac{1}{2}\beta_2(\mu) , \tag{C.2j}$$

$$\tilde{\nu}_1 = -\frac{1}{2}\beta_1(\mu) , \tag{C.2k}$$

$$\tilde{\nu}_2 = \frac{1}{2}\beta_2(\mu) , \tag{C.2l}$$

Note that we can also allows the transsport coefficients to also depends on additional scale as discussed in Section 2.2 and it will not change the conclusion of this appendix.

As mentioned earlier, the appearance of $\tilde{\zeta}$ and $\tilde{\nu}_i$ in our results simply comes from the fact that the variations of the effective action give a different hydrodynamic frame than the one adopted in [30]. One can show that by changing frame, as presented in Eqs. (2.35)-(2.36b), to the one with $\delta\rho, s_{v\otimes v}^{\mu\nu} = 0$ the transport coefficients in $T^{\mu\nu}$ transform as follows:

$$\zeta_{(1,1)} \mapsto \zeta_{(1,1)} - \mu\tilde{\zeta} , \tag{C.3}$$

$$\zeta_{(2)} \mapsto \zeta_{(2)} - \frac{\rho(\mu)}{\rho'(\mu)}\tilde{\zeta} , \tag{C.4}$$

$$\nu_0 \mapsto \nu_0 - \mu\tilde{\nu}_0 , \tag{C.5}$$

$$\nu_1 \mapsto \nu_1 - \mu\tilde{\nu}_1 , \tag{C.6}$$

$$\nu_2 \mapsto \nu_2 - \mu\tilde{\nu}_2 . \tag{C.7}$$

Additionally, the authors of [30] proposed a condition on transport coefficients which arises from the constraint that makes the equations of motion not overdetermined, namely

$$(\nabla_\mu T^{\mu\nu})\,\Omega_{\nu\lambda} + \mu\,(\nabla_\mu J^{\mu\nu})\,u_{\nu\lambda} = 0 . \tag{C.8}$$

While this relation is obviously satisfied at the zeroth order, it imposes a constraint on the second-order transport coefficients. It was shown by [42] (see their appendix C), that this relation follows from the diffeomorphism invariance of the action generated by a vector

field $\xi^\mu$ in the $SO(1,1)$ plane i.e. $\xi^\mu = \Omega^{\mu\nu}\xi_\nu$. In our case the above constraint generates the following condition for the transport coefficients:

$$\nu_0 + \nu_2 - \mu(\tilde{\nu}_0 + \tilde{\nu}_2) = \frac{\rho(\mu)}{\mu\rho'(\mu)}\left(-\zeta_{(1,1)} + \eta_{(1,1)} + \nu_1 + \mu(\tilde{\zeta} - \tilde{\nu}_1)\right) , \qquad \text{(C.9)}$$

which is satisfied by the expressions in (D.2a)-(C.2l). Using the mappings of the transport coefficients under the frame change into the frame with $\delta\rho, s^{\mu\nu}_{v\otimes v} = 0$ we can also show that this condition becomes:

$$\nu_0 + \nu_2 = \frac{\rho(\mu)}{\mu\rho'(\mu)}\left(-\zeta_{(1,1)} + \eta_{(1,1)} + \nu_1\right) , \qquad \text{(C.10)}$$

which agrees with [30]. It is also worth noting that the above condition is also satisfied in our frame choice by the expressions in (D.2a)-(C.2i) since the transport coefficients in $J^{\mu\nu}$ cancel each other out independently, $\tilde{\nu}_0 + \tilde{\nu}_2 = 0$ and $\tilde{\zeta} - \tilde{\nu}_1 = 0$.

# D  Full non-linear constitutive relations with curvature and field strength

While we strongly recommend working at the level of the effective action when possible, we still would like to list the full constitutive relations for completeness. Recall that the decomposition of the constitutive relations (2.32) is:

$$T^{\mu\nu} = -(\varepsilon + \delta\varepsilon)\,\Omega^{\mu\nu} + (p + \delta p)\,\Pi^{\mu\nu} + t^{\mu\nu}_{SO(1,1)} + t^{\mu\nu}_{SO(2)} + t^{\mu\nu}_{v\otimes v} ,$$
$$J^{\mu\nu} = (\rho + \delta\rho)\,u^{\mu\nu} + s^{\mu\nu}_{SO(2)} + s^{\mu\nu}_{v\otimes v} .$$

We first address structures appearing in each second-order piece and then show how they are related to transport coefficients $\alpha$, $\beta_i$ and $\gamma_i$ in the effective action. These results are obtained via the variation of the action with respect to the background metric and two-form gauge field. Its implementation can be found in the file `Action.nb` in [59].

## D.1  Second-order scalars

The scalars $\delta\varepsilon$, $\delta p$ and $\delta\rho$ obtained by varying the effective action are:

$$
\begin{aligned}
\delta\varepsilon &= \zeta_{(1,1)}\,u^{\alpha\beta}\Pi^{\gamma\delta}u_{\alpha\gamma;\beta\delta} + \zeta^{(\gamma_1)}_{(1,1)}\,R_{\alpha\gamma\beta\delta}u^{\alpha\beta}u^{\gamma\delta} + \zeta^{(\gamma_2)}_{(1,1)}\,R_{\alpha\gamma\beta\delta}\Pi^{\alpha\beta}\Pi^{\gamma\delta} \\
&\quad + \zeta^{(\gamma_3)}_{(1,1)}\,R_{\alpha\gamma\beta\delta}\Pi^{\alpha\beta}\Omega^{\gamma\delta} + \zeta^{(\gamma_4)}_{(1,1)}\,u^{\alpha\beta}u^{\gamma\delta}\Pi^{\lambda\rho}u_{\delta\rho;\beta}u_{\alpha\lambda;\gamma} + \zeta^{(\gamma_5)}_{(1,1)}\,u^{\alpha\beta}u^{\gamma\delta}\Pi^{\lambda\rho}u_{\alpha\lambda;\beta}u_{\gamma\rho;\delta} \\
&\quad + \zeta^{(\gamma_6)}_{(1,1)}\,\Pi^{\alpha\beta}\Pi^{\gamma\delta}\Omega^{\lambda\rho}u_{\delta\rho;\beta}u_{\alpha\lambda;\gamma} + \zeta^{(\gamma_7)}_{(1,1)}\,\Pi^{\alpha\beta}\Pi^{\gamma\delta}\Omega^{\lambda\rho}u_{\alpha\lambda;\beta}u_{\gamma\rho;\delta} \\
&\quad + \zeta^{(\gamma_8)}_{(1,1)}\,\Pi^{\alpha\beta}\Pi^{\gamma\delta}\Omega^{\lambda\rho}u_{\alpha\lambda;\gamma}u_{\beta\rho;\delta} + \zeta^{(\beta_1)}_{(1,1)}\,H_{\beta\delta\rho}u^{\alpha\beta}u^{\gamma\delta}\Pi^{\lambda\rho}u_{\alpha\lambda;\gamma} \\
&\quad + \zeta^{(\beta_2)}_{(1,1)}\,H_{\beta\delta\rho}\Pi^{\alpha\beta}\Pi^{\gamma\delta}\Omega^{\lambda\rho}u_{\alpha\lambda;\gamma} + \zeta^{(\beta_3)}_{(1,1)}\,u^{\alpha\beta}\Pi^{\gamma\delta}H_{\alpha\beta\gamma;\delta} \\
&\quad + \zeta^{(\alpha)}_{(1,1)}\,H_{\alpha\gamma\lambda}H_{\beta\delta\rho}\Pi^{\alpha\beta}\Pi^{\gamma\delta}\Omega^{\lambda\rho} + \mathcal{O}(\partial^3) ,
\end{aligned} \qquad \text{(D.1a)}
$$

$$\delta p = \zeta_{(2)}\, u^{\alpha\beta}\Pi^{\gamma\delta}u_{\alpha\gamma;\beta\delta} + \zeta_{(2)}^{(\gamma_1)}\, R_{\alpha\gamma\beta\delta}u^{\alpha\beta}u^{\gamma\delta} + \zeta_{(2)}^{(\gamma_3)}\, R_{\alpha\gamma\beta\delta}\Pi^{\alpha\beta}\Omega^{\gamma\delta}$$

$$+ \zeta_{(2)}^{(\gamma_4)}\, u^{\alpha\beta}u^{\gamma\delta}\Pi^{\lambda\rho}u_{\delta\rho;\beta}u_{\alpha\lambda;\gamma} + \zeta_{(2)}^{(\gamma_5)}\, u^{\alpha\beta}u^{\gamma\delta}\Pi^{\lambda\rho}u_{\alpha\lambda;\beta}u_{\gamma\rho;\delta}$$

$$+ \zeta_{(2)}^{(\gamma_6)}\, \Pi^{\alpha\beta}\Pi^{\gamma\delta}\Omega^{\lambda\rho}u_{\delta\rho;\beta}u_{\alpha\lambda;\gamma} + \zeta_{(2)}^{(\gamma_7)}\, \Pi^{\alpha\beta}\Pi^{\gamma\delta}\Omega^{\lambda\rho}u_{\alpha\lambda;\beta}u_{\gamma\rho;\delta}$$

$$+ \zeta_{(2)}^{(\gamma_8)}\, \Pi^{\alpha\beta}\Pi^{\gamma\delta}\Omega^{\lambda\rho}u_{\alpha\lambda;\gamma}u_{\beta\rho;\delta} + \zeta_{(2)}^{(\beta_1)}\, H_{\beta\delta\rho}u^{\alpha\beta}u^{\gamma\delta}\Pi^{\lambda\rho}u_{\alpha\lambda;\gamma} \tag{D.1b}$$

$$+ \zeta_{(2)}^{(\beta_2)}\, H_{\beta\delta\rho}\Pi^{\alpha\beta}\Pi^{\gamma\delta}\Omega^{\lambda\rho}u_{\alpha\lambda;\gamma} + \zeta_{(2)}^{(\beta_3)}\, u^{\alpha\beta}\Pi^{\gamma\delta}H_{\alpha\beta\gamma;\delta}$$

$$+ \zeta_{(2)}^{(\alpha)}\, H_{\alpha\gamma\lambda}H_{\beta\delta\rho}\Pi^{\alpha\beta}\Pi^{\gamma\delta}\Omega^{\lambda\rho} + \mathcal{O}(\partial^3)\ ,$$

$$\delta\rho = \tilde{\zeta}\, u^{\alpha\beta}\Pi^{\gamma\delta}u_{\alpha\gamma;\beta\delta} + \tilde{\zeta}^{(\gamma_1)}\, R_{\alpha\gamma\beta\delta}u^{\alpha\beta}u^{\gamma\delta} + \tilde{\zeta}^{(\gamma_2)}\, R_{\alpha\gamma\beta\delta}\Pi^{\alpha\beta}\Pi^{\gamma\delta}$$

$$+ \tilde{\zeta}^{(\gamma_3)}\, R_{\alpha\gamma\beta\delta}\Pi^{\alpha\beta}\Omega^{\gamma\delta} + \tilde{\zeta}^{(\gamma_4)}\, u^{\alpha\beta}u^{\gamma\delta}\Pi^{\lambda\rho}u_{\delta\rho;\beta}u_{\alpha\lambda;\gamma}$$

$$+ \tilde{\zeta}^{(\gamma_5)}\, u^{\alpha\beta}u^{\gamma\delta}\Pi^{\lambda\rho}u_{\alpha\lambda;\beta}u_{\gamma\rho;\delta} + \tilde{\zeta}^{(\gamma_6)}\, \Pi^{\alpha\beta}\Pi^{\gamma\delta}\Omega^{\lambda\rho}u_{\delta\rho;\beta}u_{\alpha\lambda;\gamma} \tag{D.1c}$$

$$+ \tilde{\zeta}^{(\gamma_7)}\, \Pi^{\alpha\beta}\Pi^{\gamma\delta}\Omega^{\lambda\rho}u_{\alpha\lambda;\beta}u_{\gamma\rho;\delta} + \tilde{\zeta}^{(\gamma_8)}\, \Pi^{\alpha\beta}\Pi^{\gamma\delta}\Omega^{\lambda\rho}u_{\alpha\lambda;\gamma}u_{\beta\rho;\delta}$$

$$+ \tilde{\zeta}^{(\beta_2)}\, H_{\beta\delta\rho}\Pi^{\alpha\beta}\Pi^{\gamma\delta}\Omega^{\lambda\rho}u_{\alpha\lambda;\gamma} + \tilde{\zeta}^{(\alpha)}\, H_{\alpha\gamma\lambda}H_{\beta\delta\rho}\Pi^{\alpha\beta}\Pi^{\gamma\delta}\Omega^{\lambda\rho} + \mathcal{O}(\partial^3)\ ,$$

The transport coefficients $\zeta_{(1,1)}$, $\zeta_{(2)}$ and $\tilde{\zeta}$ correspond to the corrections to $\delta\varepsilon$, $\delta p$ and $\delta\rho$, respectively. The superscripts on transport coefficients follow and expand on the numbering of scalars used in the effective action. And the coefficients without superscripts correspond to terms that contribute to the linearised theory and follow the notation of [30]. The above transport coefficients correspond to the coefficients in the effective action in the following way:

$$\zeta_{(1,1)} = -\frac{\rho(\mu)\gamma_1'(\mu)}{\rho'(\mu)} + 2\gamma_1(\mu) + 2\gamma_5(\mu) - \mu\gamma_3'(\mu) + \gamma_3(\mu) - \gamma_4(\mu)\ , \tag{D.2a}$$

$$\zeta_{(1,1)}^{(\gamma_1)} = \mu\gamma_1'(\mu)\ , \qquad\qquad\qquad \zeta_{(1,1)}^{(\gamma_2)} = \mu\gamma_2'(\mu) - \gamma_2(\mu)\ , \tag{D.2b}$$

$$\zeta_{(1,1)}^{(\gamma_3)} = -\frac{\rho(\mu)\gamma_1'(\mu)}{\rho'(\mu)} + \gamma_1(\mu) + \mu\gamma_3'(\mu)\ , \qquad \zeta_{(1,1)}^{(\gamma_4)} = \frac{\rho(\mu)\gamma_1'(\mu)}{\rho'(\mu)} - 2\gamma_1(\mu) \tag{D.2c}$$

$$- \gamma_3(\mu) + \mu\gamma_4'(\mu) - \gamma_4(\mu)\ ,$$

$$\zeta_{(1,1)}^{(\gamma_5)} = -\mu\gamma_1'(\mu) - \mu\gamma_5'(\mu) + \gamma_5(\mu) + \mu^2\gamma_3''(\mu) + \mu\gamma_4'(\mu) - \gamma_4(\mu)\ , \tag{D.2d}$$

$$\zeta_{(1,1)}^{(\gamma_6)} = \gamma_1(\mu) + \mu\gamma_6'(\mu) - \gamma_6(\mu) + 2\gamma_5(\mu) - \mu\gamma_3'(\mu) - \gamma_4(\mu) + \gamma_8(\mu) + \gamma_2(\mu)\ , \tag{D.2e}$$

$$\zeta_{(1,1)}^{(\gamma_7)} = \frac{\rho(\mu)^2\gamma_1''(\mu)}{\rho'(\mu)^2} - \frac{\rho(\mu)\gamma_1'(\mu)}{\rho'(\mu)} - \frac{\rho(\mu)^2\gamma_1'(\mu)\rho''(\mu)}{\rho'(\mu)^3} + \gamma_1(\mu) + \mu\gamma_7'(\mu) + \gamma_2(\mu)\ , \tag{D.2f}$$

$$\zeta_{(1,1)}^{(\gamma_8)} = \gamma_6(\mu) + \gamma_3(\mu) + \mu\gamma_8'(\mu) - \gamma_8(\mu) - 2\gamma_2(\mu)\ , \tag{D.2g}$$

$$\zeta_{(1,1)}^{(\beta_1)} = 2\gamma_1'(\mu) + \beta_1(\mu) + 4\gamma_5'(\mu) - 4\mu\gamma_3''(\mu) - 2\gamma_4'(\mu)\ , \tag{D.2h}$$

$$\zeta_{(1,1)}^{(\beta_2)} = \mu\beta_2'(\mu) - \beta_2(\mu) - \beta_1(\mu) - 2\gamma_3'(\mu)\ , \qquad\qquad \zeta_{(1,1)}^{(\beta_3)} = \frac{1}{2}\beta_1(\mu) + \gamma_3'(\mu)\ , \tag{D.2i}$$

$$\zeta_{(1,1)}^{(\alpha)} = \mu\alpha'(\mu)\ , \tag{D.2j}$$

$$\zeta_{(2)} = -\gamma_6(\mu) - 2\gamma_7(\mu) - \frac{\rho(\mu)\gamma_3'(\mu)}{\rho'(\mu)} + \gamma_3(\mu) - \gamma_8(\mu) + \mu\gamma_2'(\mu) - 2\gamma_2(\mu) \, , \tag{D.3a}$$

$$\zeta_{(2)}^{(\gamma_1)} = \gamma_1(\mu) \, , \qquad \zeta_{(2)}^{(\gamma_3)} = -\gamma_6(\mu) - 2\gamma_7(\mu) - \frac{\rho(\mu)\gamma_3'(\mu)}{\rho'(\mu)} + \gamma_3(\mu) - \gamma_8(\mu) - \gamma_2(\mu) \, , \tag{D.3b}$$

$$\zeta_{(2)}^{(\gamma_4)} = -\gamma_1(\mu) + \gamma_6(\mu) + 2\gamma_7(\mu) + \frac{\rho(\mu)\gamma_3'(\mu)}{\rho'(\mu)} - \gamma_3(\mu) + \gamma_8(\mu) + \gamma_2(\mu) \, , \tag{D.3c}$$

$$\zeta_{(2)}^{(\gamma_5)} = -\gamma_1(\mu) - \mu^2\gamma_2''(\mu) + \mu\gamma_2'(\mu) - \gamma_2(\mu) \, , \qquad \zeta_{(2)}^{(\gamma_6)} = \gamma_8(\mu) + \mu\gamma_2'(\mu) \, , \tag{D.3d}$$

$$\zeta_{(2)}^{(\gamma_7)} = \frac{\rho(\mu)\gamma_6'(\mu)}{\rho'(\mu)} - \gamma_6(\mu) + \frac{2\rho(\mu)\gamma_7'(\mu)}{\rho'(\mu)} - \gamma_7(\mu) + \frac{\rho(\mu)^2\gamma_3''(\mu)}{\rho'(\mu)^2} \tag{D.3e}$$
$$- \frac{\rho(\mu)^2\gamma_3'(\mu)\rho''(\mu)}{\rho'(\mu)^3} + \frac{\rho(\mu)\gamma_8'(\mu)}{\rho'(\mu)} - \gamma_8(\mu) + \frac{\rho(\mu)\gamma_2'(\mu)}{\rho'(\mu)} \, ,$$
$$\zeta_{(2)}^{(\gamma_8)} = \gamma_6(\mu) + \gamma_3(\mu) - 2\gamma_2(\mu) \, , \tag{D.3f}$$

$$\zeta_{(2)}^{(\beta_1)} = 4\mu\gamma_2''(\mu) - 2\gamma_2'(\mu) \, , \qquad\qquad \zeta_{(2)}^{(\beta_2)} = 2\gamma_2'(\mu) - \beta_2(\mu) \, , \tag{D.3g}$$
$$\zeta_{(2)}^{(\beta_3)} = -\gamma_2'(\mu) \, , \qquad\qquad \zeta_{(2)}^{(\alpha)} = -\alpha(\mu) \, , \tag{D.3h}$$

$$\tilde{\zeta} = -\frac{1}{2}\beta_1(\mu) \, , \tag{D.4a}$$

$$\tilde{\zeta}^{(\gamma_1)} = \gamma_1'(\mu) \, , \qquad\qquad \tilde{\zeta}^{(\gamma_2)} = \gamma_2'(\mu) \, , \tag{D.4b}$$
$$\tilde{\zeta}^{(\gamma_3)} = \gamma_3'(\mu) \, , \qquad\qquad \tilde{\zeta}^{(\gamma_4)} = \gamma_4'(\mu) \, , \tag{D.4c}$$
$$\tilde{\zeta}^{(\gamma_5)} = \frac{1}{2}\mu\beta_1'(\mu) + \gamma_5'(\mu) \, , \qquad\qquad \tilde{\zeta}^{(\gamma_6)} = -\frac{1}{2}\beta_2(\mu) - \frac{1}{2}\beta_1(\mu) + \gamma_6'(\mu) \, , \tag{D.4d}$$
$$\tilde{\zeta}^{(\gamma_7)} = \gamma_7'(\mu) \, , \qquad\qquad \tilde{\zeta}^{(\gamma_8)} = \frac{1}{2}\beta_2(\mu) + \gamma_8'(\mu) \, , \tag{D.4e}$$

$$\tilde{\zeta}^{(\beta_2)} = \beta_2'(\mu) + 2\alpha(\mu) \, , \tag{D.4f}$$

$$\tilde{\zeta}^{(\alpha)} = \alpha'(\mu) \, , \tag{D.4g}$$

## D.2   Second-order symmetric tensors

There are three symmetric rank-two tensors $t_{SO(1,1)}^{\mu\nu}$, $t_{SO(2)}^{\mu\nu}$ and $t_{v\otimes v}^{\mu\nu}$ which transform as $SO(1,1)$ and $SO(2)$ tensors, and a product of vectors under $SO(1,1) \otimes SO(2)$, respectively.

By varying the effecitve action w.r.t. the background metric, we find:

$$
\begin{aligned}
t_{SO(1,1)}^{\mu\nu} =\ & -2\eta_{(1,1)}\, u^{\langle\mu|\alpha}\Pi^{\beta\gamma}\Omega^{|\nu\rangle\delta}u_{\alpha\beta;\gamma\delta} + 2\eta_{(1,1)}^{(\gamma_3)}\, R_{\alpha\gamma\beta\delta}\Pi^{\alpha\beta}\Omega^{\langle\mu|\gamma}\Omega^{|\nu\rangle\delta} \\
& + 2\eta_{(1,1)}^{(\gamma_5)}\, u^{\beta\gamma}u^{\langle\mu|\alpha}\Pi^{\delta\lambda}\Omega^{|\nu\rangle\rho}u_{\alpha\delta;\rho}u_{\beta\lambda;\gamma} + 2\eta_{(1,1)}^{(\gamma_6)}\, \Pi^{\alpha\beta}\Pi^{\gamma\delta}\Omega^{\langle\mu|\lambda}\Omega^{|\nu\rangle\rho}u_{\delta\rho;\beta}u_{\alpha\lambda;\gamma} \\
& + 2\eta_{(1,1)}^{(\gamma_7)}\, \Pi^{\alpha\beta}\Pi^{\gamma\delta}\Omega^{\langle\mu|\lambda}\Omega^{|\nu\rangle\rho}u_{\alpha\lambda;\beta}u_{\gamma\rho;\delta} + 2\eta_{(1,1)}^{(\gamma_8)}\, \Pi^{\alpha\beta}\Pi^{\gamma\delta}\Omega^{\langle\mu|\lambda}\Omega^{|\nu\rangle\rho}u_{\alpha\lambda;\gamma}u_{\beta\rho;\delta} \quad \text{(D.5a)} \\
& + 2\eta_{(1,1)}^{(\beta_2)}\, H_{\beta\delta\rho}\Pi^{\alpha\beta}\Pi^{\gamma\delta}\Omega^{\langle\mu|\lambda}\Omega^{|\nu\rangle\rho}u_{\alpha\lambda;\gamma} + 2\eta_{(1,1)}^{(\beta_3)}\, u^{\langle\mu|\alpha}\Pi^{\beta\gamma}\Omega^{|\nu\rangle\delta}H_{\alpha\beta\delta;\gamma} \\
& + 2\eta_{(1,1)}^{(\beta_4)}\, H_{\beta\delta\rho}\Pi^{\alpha\beta}\Omega^{\lambda\rho}\Omega^{\langle\mu|\gamma}\Omega^{|\nu\rangle\delta}u_{\alpha\gamma;\lambda} + 2\eta_{(1,1)}^{(\alpha)}\, H_{\alpha\gamma\lambda}H_{\beta\delta\rho}\Pi^{\alpha\beta}\Pi^{\gamma\delta}\Omega^{\langle\mu|\lambda}\Omega^{|\nu\rangle\rho} + \mathcal{O}(\partial^3)\ ,
\end{aligned}
$$

$$
\begin{aligned}
t_{SO(2)}^{\mu\nu} =\ & 2\eta_{(2)}\, u^{\alpha\beta}\Pi^{\langle\mu|\gamma}\Pi^{|\nu\rangle\delta}u_{\alpha\delta;\beta\gamma} + 2\eta_{(2)}^{(\gamma_2)}\, R_{\alpha\gamma\beta\delta}\Pi^{\gamma\delta}\Pi^{\langle\mu|\alpha}\Pi^{|\nu\rangle\beta} + 2\eta_{(2)}^{(\gamma_3)}\, R_{\alpha\gamma\beta\delta}\Pi^{\langle\mu|\alpha}\Pi^{|\nu\rangle\beta}\Omega^{\gamma\delta} \\
& + 2\eta_{(2)}^{(\gamma_4)}\, u^{\alpha\beta}u^{\gamma\delta}\Pi^{\langle\mu|\lambda}\Pi^{|\nu\rangle\rho}u_{\delta\rho;\beta}u_{\alpha\lambda;\gamma} + 2\eta_{(2)}^{(\gamma_5)}\, u^{\alpha\beta}u^{\gamma\delta}\Pi^{\langle\mu|\lambda}\Pi^{|\nu\rangle\rho}u_{\alpha\lambda;\beta}u_{\gamma\rho;\delta} \\
& + 2\eta_{(2)}^{(\gamma_6)}\, \Pi^{\gamma\delta}\Pi^{\langle\mu|\alpha}\Pi^{|\nu\rangle\beta}\Omega^{\lambda\rho}u_{\delta\rho;\beta}u_{\alpha\lambda;\gamma} + 2\eta_{(2)}^{(\gamma_7)}\, \Pi^{\gamma\delta}\Pi^{\langle\mu|\alpha}\Pi^{|\nu\rangle\beta}\Omega^{\lambda\rho}u_{\beta\lambda;\alpha}u_{\gamma\rho;\delta} \\
& + 2\eta_{(2)}^{(\gamma_8,1)}\, \Pi^{\gamma\delta}\Pi^{\langle\mu|\alpha}\Pi^{|\nu\rangle\beta}\Omega^{\lambda\rho}u_{\gamma\lambda;\alpha}u_{\delta\rho;\beta} + 2\eta_{(2)}^{(\gamma_8,2)}\, \Pi^{\gamma\delta}\Pi^{\langle\mu|\alpha}\Pi^{|\nu\rangle\beta}\Omega^{\lambda\rho}u_{\alpha\lambda;\gamma}u_{\beta\rho;\delta} \\
& \qquad\qquad\qquad\qquad\qquad\qquad\qquad\qquad\qquad\qquad\qquad\qquad\qquad\qquad\qquad\qquad \text{(D.5b)} \\
& + 2\eta_{(2)}^{(\beta_1)}\, H_{\beta\delta\rho}u^{\alpha\beta}u^{\gamma\delta}\Pi^{\langle\mu|\lambda}\Pi^{|\nu\rangle\rho}u_{\alpha\lambda;\gamma} + 2\eta_{(2)}^{(\beta_2,1)}\, H_{\beta\delta\rho}\Pi^{\gamma\delta}\Pi^{\langle\mu|\alpha}\Pi^{|\nu\rangle\beta}\Omega^{\lambda\rho}u_{\gamma\lambda;\alpha} \\
& + 2\eta_{(2)}^{(\beta_2,2)}\, H_{\beta\delta\rho}\Pi^{\gamma\delta}\Pi^{\langle\mu|\alpha}\Pi^{|\nu\rangle\beta}\Omega^{\lambda\rho}u_{\alpha\lambda;\gamma} + 2\eta_{(2)}^{(\beta_3)}\, u^{\alpha\beta}\Pi^{\langle\mu|\gamma}\Pi^{|\nu\rangle\delta}H_{\alpha\beta\delta;\gamma} \\
& + 2\eta_{(2)}^{(\alpha)}\, H_{\alpha\gamma\lambda}H_{\beta\delta\rho}\Pi^{\gamma\delta}\Pi^{\langle\mu|\alpha}\Pi^{|\nu\rangle\beta}\Omega^{\lambda\rho} + \mathcal{O}(\partial^3)\ .
\end{aligned}
$$

In the case of the above tensor structures the superscript on the transport coefficients denotes the scalar that is the trace of the corresponding tensor, again following and expanding on the numbering of scalars used in the effective action and the scalars in the constitutive relations earlier. And the numbers after comma number the tensors with the same trace. Because the off-diagonal $(v \otimes v)$-components below are trivially traceless, the superscripts on their transport coefficients carry only a greek later to indicate their structure (number of field strengths $H$) and a number after comma.

$$
\begin{aligned}
t_{v\otimes v}^{\mu\nu} =\ & -2\nu_0\, u^{(\mu|\alpha}\Pi^{\gamma\delta}\Pi^{|\nu)\beta}u_{\alpha\gamma;\beta\delta} - 2\nu_1\, u^{(\mu|\alpha}\Pi^{|\nu)\beta}\Omega^{\gamma\delta}u_{\alpha\beta;\gamma\delta} - 2\nu_2\, u^{(\mu|\alpha}\Pi^{\gamma\delta}\Pi^{|\nu)\beta}u_{\alpha\beta;\gamma\delta} \\
& + 2\nu^{(\gamma,1)}\, R_{\alpha\beta\gamma\delta}u^{\beta\gamma}u^{(\mu|\alpha}\Pi^{|\nu)\delta} + 2\nu^{(\gamma,2)}\, R_{\alpha\beta\gamma\delta}\Pi^{\beta\gamma}\Pi^{(\mu|\alpha}\Omega^{|\nu)\delta} \\
& + 2\nu^{(\gamma,3)}\, u^{\beta\gamma}u^{(\mu|\alpha}\Pi^{\lambda\rho}\Pi^{|\nu)\delta}u_{\beta\rho;\gamma}u_{\alpha\lambda;\delta} + 2\nu^{(\gamma,4)}\, u^{\beta\gamma}u^{(\mu|\alpha}\Pi^{\lambda\rho}\Pi^{|\nu)\delta}u_{\beta\rho;\gamma}u_{\alpha\delta;\lambda} \\
& + 2\nu^{(\gamma,5)}\, u^{\beta\gamma}u^{(\mu|\alpha}\Pi^{\lambda\rho}\Pi^{|\nu)\delta}u_{\alpha\lambda;\rho}u_{\beta\delta;\gamma} + 2\nu^{(\gamma,6)}\, \Pi^{\beta\gamma}\Pi^{(\mu|\alpha}\Omega^{\lambda\rho}\Omega^{|\nu)\delta}u_{\beta\rho;\gamma}u_{\alpha\lambda;\delta} \\
& + 2\nu^{(\gamma,7)}\, \Pi^{\beta\gamma}\Pi^{(\mu|\alpha}\Omega^{\lambda\rho}\Omega^{|\nu)\delta}u_{\beta\lambda;\alpha}u_{\gamma\rho;\delta} + 2\nu^{(\gamma,8)}\, \Pi^{\beta\gamma}\Pi^{(\mu|\alpha}\Omega^{\lambda\rho}\Omega^{|\nu)\delta}u_{\gamma\delta;\rho}u_{\alpha\lambda;\beta} \quad \text{(D.5c)} \\
& + 2\nu^{(\beta,1)}\, u^{(\mu|\alpha}\Pi^{\gamma\delta}\Pi^{|\nu)\beta}H_{\alpha\beta\gamma;\delta} + 2\nu^{(\beta,2)}\, u^{(\mu|\alpha}\Pi^{|\nu)\beta}\Omega^{\gamma\delta}H_{\alpha\beta\gamma;\delta} \\
& + 2\nu^{(\beta,3)}\, H_{\alpha\gamma\rho}u^{\beta\gamma}u^{(\mu|\alpha}\Pi^{\lambda\rho}\Pi^{|\nu)\delta}u_{\beta\lambda;\delta} + 2\nu^{(\beta,4)}\, u^{\alpha\beta}\Pi^{(\mu|\gamma}\Omega^{|\nu)\delta}H_{\alpha\gamma\delta;\beta} \\
& + 2\nu^{(\beta,5)}\, H_{\gamma\delta\rho}\Pi^{\beta\gamma}\Pi^{(\mu|\alpha}\Omega^{\lambda\rho}\Omega^{|\nu)\delta}u_{\alpha\lambda;\beta} + 2\nu^{(\beta,6)}\, H_{\alpha\delta\rho}\Pi^{\beta\gamma}\Pi^{(\mu|\alpha}\Omega^{\lambda\rho}\Omega^{|\nu)\delta}u_{\beta\lambda;\gamma} \\
& + 2\nu^{(\beta,7)}\, H_{\alpha\gamma\rho}\Pi^{\beta\gamma}\Pi^{(\mu|\alpha}\Omega^{\lambda\rho}\Omega^{|\nu)\delta}u_{\beta\lambda;\delta} + 2\nu^{(\beta,8)}\, H_{\alpha\gamma\rho}\Pi^{\beta\gamma}\Pi^{(\mu|\alpha}\Omega^{\lambda\rho}\Omega^{|\nu)\delta}u_{\beta\delta;\lambda} \\
& + 2\nu^{(\alpha)}\, H_{\alpha\beta\lambda}H_{\gamma\delta\rho}\Pi^{\beta\gamma}\Pi^{(\mu|\alpha}\Omega^{\lambda\rho}\Omega^{|\nu)\delta}u_{\beta\delta;\lambda} + \mathcal{O}(\partial^3)\ .
\end{aligned}
$$

The transport coefficients can be written in terms of $\alpha$, $\beta_i$ and $\gamma_i$ as follows:

$$
\eta_{(1,1)} = -\frac{\rho(\mu)\gamma_1'(\mu)}{\rho'(\mu)} + 2\gamma_1(\mu) + \gamma_3(\mu) + \gamma_4(\mu)\ , \tag{D.6a}
$$

$$\eta^{(\gamma_3)}_{(1,1)} = -\gamma_1(\mu) - \gamma_3(\mu) - \gamma_4(\mu) \ , \tag{D.6b}$$

$$\eta^{(\gamma_5)}_{(1,1)} = \frac{\rho(\mu)\gamma'_1(\mu)}{\rho'(\mu)} + \mu\gamma'_1(\mu) - 2\gamma_1(\mu) + \mu\gamma'_3(\mu) - \gamma_3(\mu) + \mu\gamma'_4(\mu) - \gamma_4(\mu) \ , \tag{D.6c}$$

$$\eta^{(\gamma_6)}_{(1,1)} = -\gamma_1(\mu) - \gamma_3(\mu) - \gamma_4(\mu) + \gamma_8(\mu) + \gamma_2(\mu) \ , \tag{D.6d}$$

$$\eta^{(\gamma_7)}_{(1,1)} = -\frac{\rho(\mu)^2\gamma''_1(\mu)}{\rho'(\mu)^2} + \frac{\rho(\mu)\gamma'_1(\mu)}{\rho'(\mu)} + \frac{\rho(\mu)^2\gamma'_1(\mu)\rho''(\mu)}{\rho'(\mu)^3} - \gamma_1(\mu) + \gamma_7(\mu) \tag{D.6e}$$
$$+ \frac{\rho(\mu)\gamma'_3(\mu)}{\rho'(\mu)} - \gamma_3(\mu) + \gamma_2(\mu) \ ,$$

$$\eta^{(\gamma_8)}_{(1,1)} = \gamma_6(\mu) + \gamma_3(\mu) - 2\gamma_2(\mu) \ , \tag{D.6f}$$

$$\eta^{(\beta_2)}_{(1,1)} = -\beta_2(\mu) \ , \qquad\qquad \eta^{(\beta_3)}_{(1,1)} = -\frac{1}{2}\beta_1(\mu) \ , \tag{D.6g}$$

$$\eta^{(\beta_4)}_{(1,1)} = -2\gamma'_1(\mu) - 2\gamma'_3(\mu) - 2\gamma'_4(\mu) \ , \tag{D.6h}$$

$$\eta^{(\alpha)}_{(1,1)} = -\alpha(\mu) \ , \tag{D.6i}$$

$$\eta_{(2)} = -\gamma_6(\mu) - \gamma_3(\mu) - \gamma_8(\mu) - \mu\gamma'_2(\mu) + 2\gamma_2(\mu) \ , \tag{D.7a}$$

$$\eta^{(\gamma_2)}_{(2)} = -\gamma_2(\mu) \ , \qquad \eta^{(\gamma_3)}_{(2)} = -\gamma_6(\mu) - \gamma_3(\mu) - \gamma_8(\mu) + \gamma_2(\mu) \ , \tag{D.7b}$$

$$\eta^{(\gamma_4)}_{(2)} = -\gamma_1(\mu) + \gamma_6(\mu) - \gamma_4(\mu) + \gamma_8(\mu) - \gamma_2(\mu) \ , \tag{D.7c}$$

$$\eta^{(\gamma_5)}_{(2)} = -\gamma_1(\mu) - \gamma_5(\mu) + \mu\gamma'_3(\mu) - \gamma_3(\mu) + \mu^2\gamma''_2(\mu) - \mu\gamma'_2(\mu) + \gamma_2(\mu) \ , \tag{D.7d}$$

$$\eta^{(\gamma_6)}_{(2)} = -\gamma_6(\mu) - \gamma_3(\mu) + \gamma_8(\mu) - \mu\gamma'_2(\mu) + 2\gamma_2(\mu) \ , \tag{D.7e}$$

$$\eta^{(\gamma_7)}_{(2)} = \frac{\rho(\mu)\gamma'_6(\mu)}{\rho'(\mu)} - \gamma_6(\mu) + \frac{\rho(\mu)\gamma'_3(\mu)}{\rho'(\mu)} - \gamma_3(\mu) + \frac{\rho(\mu)\gamma'_8(\mu)}{\rho'(\mu)} - \gamma_8(\mu) \tag{D.7f}$$
$$- \frac{\rho(\mu)\gamma'_2(\mu)}{\rho'(\mu)} + 2\gamma_2(\mu) \ ,$$

$$\eta^{(\gamma_8,1)}_{(2)} = -\gamma_8(\mu) \ , \qquad \eta^{(\gamma_8,2)}_{(2)} = \gamma_6(\mu) + \gamma_3(\mu) - 2\gamma_2(\mu) \ , \tag{D.7g}$$

$$\eta^{(\beta_1)}_{(2)} = -\beta_1(\mu) - 2\gamma'_3(\mu) - 4\mu\gamma''_2(\mu) + 2\gamma'_2(\mu) \ , \quad \eta^{(\beta_2,1)}_{(2)} = \beta_2(\mu) + 2\gamma'_2(\mu) \ , \tag{D.7h}$$

$$\eta^{(\beta_2,2)}_{(2)} = -\beta_2(\mu) \ , \qquad\qquad\qquad \eta^{(\beta_3)}_{(2)} = \gamma'_2(\mu) \ , \tag{D.7i}$$

$$\eta^{(\alpha)}_{(2)} = -2\alpha(\mu) \ , \tag{D.7j}$$

$$\nu_0 = -\gamma_6(\mu) + \frac{\rho(\mu)\gamma'_3(\mu)}{\rho'(\mu)} - \gamma_3(\mu) + \gamma_8(\mu) + 2\gamma_2(\mu) \ , \tag{D.8a}$$

$$\nu_1 = -2\gamma_4(\mu) + 2\gamma_5(\mu) \ , \qquad \nu_2 = \gamma_6(\mu) + \gamma_3(\mu) - \gamma_8(\mu) - 2\gamma_2(\mu) \ , \tag{D.8b}$$

$$\nu^{(\gamma,1)} = 2\gamma_1(\mu) + 2\gamma_5(\mu) + 2\gamma_3(\mu) \ , \qquad \nu^{(\gamma,2)} = 2\gamma_2(\mu) - \frac{\rho(\mu)\gamma_3'(\mu)}{\rho'(\mu)} \ , \tag{D.8c}$$

$$\nu^{(\gamma,3)} = -\mu\gamma_6'(\mu) + \gamma_6(\mu) + \gamma_3(\mu) + \mu\gamma_8'(\mu) - \gamma_8(\mu) + 2\mu\gamma_2'(\mu) - 2\gamma_2(\mu) \ , \tag{D.8d}$$

$$\nu^{(\gamma,4)} = \mu\gamma_6'(\mu) - \gamma_6(\mu) + 2\mu\gamma_3'(\mu) - \gamma_3(\mu) - \mu\gamma_8'(\mu) + \gamma_8(\mu) - 4\mu\gamma_2'(\mu) + 2\gamma_2(\mu) \ , \tag{D.8e}$$

$$\nu^{(\gamma,5)} = -\frac{2\rho(\mu)\gamma_1'(\mu)}{\rho'(\mu)} + 2\gamma_1(\mu) - \frac{2\rho(\mu)\gamma_5'(\mu)}{\rho'(\mu)} + 2\gamma_5(\mu) + \frac{\mu\rho(\mu)\gamma_3''(\mu)}{\rho'(\mu)} \tag{D.8f}$$
$$- \frac{\rho(\mu)\gamma_3'(\mu)}{\rho'(\mu)} - \frac{\mu\rho(\mu)\gamma_3'(\mu)\rho''(\mu)}{\rho'(\mu)^2} + 2\gamma_3(\mu) + 2\mu\gamma_2'(\mu) - 2\gamma_2(\mu) \ ,$$

$$\nu^{(\gamma,6)} = \frac{2\rho(\mu)\gamma_1'(\mu)}{\rho'(\mu)} - 2\gamma_1(\mu) - \gamma_6(\mu) + \frac{\rho(\mu)\gamma_3'(\mu)}{\rho'(\mu)} - 3\gamma_3(\mu) + \frac{2\rho(\mu)\gamma_4'(\mu)}{\rho'(\mu)} \tag{D.8g}$$
$$- 2\gamma_4(\mu) + \gamma_8(\mu) + 4\gamma_2(\mu) \ ,$$

$$\nu^{(\gamma,7)} = \gamma_6(\mu) - \frac{\rho(\mu)\gamma_3'(\mu)}{\rho'(\mu)} + \gamma_3(\mu) - \gamma_8(\mu) - 2\gamma_2(\mu) \ , \tag{D.8h}$$

$$\nu^{(\gamma,8)} = 2\gamma_1(\mu) + 2\gamma_3(\mu) + 2\gamma_4(\mu) - 2\gamma_2(\mu) \ , \tag{D.8i}$$

$$\nu^{(\beta,1)} = -\beta_2(\mu) \ , \qquad\qquad\qquad \nu^{(\beta,2)} = \frac{1}{2}\beta_1(\mu) \ , \tag{D.8j}$$

$$\nu^{(\beta,3)} = -\beta_2(\mu) + 2\gamma_6'(\mu) - 2\gamma_8'(\mu) - 4\gamma_2'(\mu) \ , \qquad \nu^{(\beta,4)} = -\frac{1}{2}\beta_1(\mu) \ , \tag{D.8k}$$

$$\nu^{(\beta,5)} = -\beta_1(\mu) - 2\gamma_6'(\mu) - 4\gamma_3'(\mu) + 2\gamma_8'(\mu) + 8\gamma_2'(\mu) \ , \tag{D.8l}$$

$$\nu^{(\beta,6)} = \beta_2(\mu) - \frac{\rho(\mu)\beta_1'(\mu)}{\rho'(\mu)} + \beta_1(\mu) - \frac{2\rho(\mu)\gamma_3''(\mu)}{\rho'(\mu)} + \frac{2\rho(\mu)\gamma_3'(\mu)\rho''(\mu)}{\rho'(\mu)^2} - 4\gamma_2'(\mu) \ , \tag{D.8m}$$

$$\nu^{(\beta,7)} = -\mu\beta_2'(\mu) + 2\beta_2(\mu) + \beta_1(\mu) \ , \qquad \nu^{(\beta,8)} = \mu\beta_2'(\mu) - \beta_2(\mu) \ , \tag{D.8n}$$

$$\nu^{(\alpha)} = 2\beta_2'(\mu) + 4\alpha(\mu) \ . \tag{D.8o}$$

### D.3   Second-order anti-symmetric tensors

Similarly to the symmetric tensors, the antisymmetric tensor components $s_{SO(1,1)}^{\mu\nu}$, $s_{SO(2)}^{\mu\nu}$ and $s_{v\otimes v}^{\mu\nu}$ can be obtained by varying the action w.r.t. the 2-form source $b_{\mu\nu}$. It turns out that the $SO(1,1)$-components obtained this way are all proportional to $u^{\mu\nu}$ so they are contained in $\delta\rho$, at this order in the derivative expansion. The non-zero components $s_{SO(2)}^{\mu\nu}$ and $s_{v\otimes v}^{\mu\nu}$ are:

$$s_{SO(2)}^{\mu\nu} = 2\tilde{\eta}_{(2)}^{(\gamma,1)} R_{\alpha\gamma\beta\delta} u^{\alpha\beta}\Pi^{[\mu|\gamma}\Pi^{|\nu]\delta} + 2\tilde{\eta}_{(2)}^{(\gamma,2)} u^{\alpha\beta}\Pi^{[\mu|\gamma}\Pi^{|\nu]\delta}\Omega^{\lambda\rho}u_{\gamma\lambda;\alpha}u_{\delta\rho;\beta}$$
$$+ 2\tilde{\eta}_{(2)}^{(\gamma,3)} u^{\alpha\beta}\Pi^{[\mu|\gamma}\Pi^{|\nu]\delta}\Omega^{\lambda\rho}u_{\delta\rho;\beta}u_{\alpha\gamma;\lambda} + +2\tilde{\eta}_{(2)}^{(\gamma,4)} u^{\alpha\beta}\Pi^{\lambda\rho}\Pi^{[\mu|\gamma}\Pi^{|\nu]\delta}u_{\beta\delta;\rho}u_{\alpha\gamma;\lambda} \tag{D.9a}$$
$$+ 2\tilde{\eta}_{(2)}^{(\gamma,5)} u^{\alpha\beta}\Pi^{\lambda\rho}\Pi^{[\mu|\gamma}\Pi^{|\nu]\delta}u_{\beta\lambda;\rho}u_{\alpha\delta;\gamma} + 2\tilde{\eta}_{(2)}^{(\beta,1)} \Pi^{[\mu|\alpha}\Pi^{|\nu]\beta}\Omega^{\gamma\delta}H_{\alpha\beta\gamma;\delta}$$
$$+ 2\tilde{\eta}_{(2)}^{(\beta,2)} H_{\beta\delta\rho}u^{\alpha\beta}\Pi^{\lambda\rho}\Pi^{[\mu|\gamma}\Pi^{|\nu]\delta}u_{\alpha\gamma;\lambda} + 2\tilde{\eta}_{(2)}^{(\beta,3)} H_{\beta\gamma\delta}u^{\alpha\beta}\Pi^{\lambda\rho}\Pi^{[\mu|\gamma}\Pi^{|\nu]\delta}u_{\alpha\lambda;\rho} + \mathcal{O}(\partial^3) \ ,$$

$$s^{\mu\nu}_{v\otimes v} = 2\tilde{\nu}_0 \, \Pi^{\beta\gamma}\Pi^{[\mu|\alpha}\Omega^{|\nu]\delta}u_{\beta\delta;\alpha\gamma} + 2\tilde{\nu}_1 \, \Pi^{[\mu|\alpha}\Omega^{\gamma\delta}\Omega^{|\nu]\beta}u_{\alpha\beta;\gamma\delta} + 2\tilde{\nu}_2 \, \Pi^{\beta\gamma}\Pi^{[\mu|\alpha}\Omega^{|\nu]\delta}u_{\alpha\delta;\beta\gamma}$$

$$+ 2\tilde{\nu}^{(\gamma,1)} \, R_{\alpha\gamma\beta\delta}u^{\alpha\beta}\Pi^{[\mu|\gamma}\Omega^{|\nu]\delta} + 2\tilde{\nu}^{(\gamma,2)} \, u^{\alpha\beta}\Pi^{\delta\lambda}\Pi^{[\mu|\gamma}\Omega^{|\nu]\rho}u_{\alpha\delta;\beta}u_{\lambda\rho;\gamma}$$

$$+ 2\tilde{\nu}^{(\gamma,3)} \, u^{\alpha\beta}\Pi^{\delta\lambda}\Pi^{[\mu|\gamma}\Omega^{|\nu]\rho}u_{\alpha\gamma;\beta}u_{\delta\rho;\lambda} + 2\tilde{\nu}^{(\gamma,4)} \, u^{\alpha\beta}\Pi^{\delta\lambda}\Pi^{[\mu|\gamma}\Omega^{|\nu]\rho}u_{\alpha\gamma;\rho}u_{\beta\delta;\lambda}$$

$$+ 2\tilde{\nu}^{(\gamma,5)} \, u^{\alpha\beta}\Pi^{\delta\lambda}\Pi^{[\mu|\gamma}\Omega^{|\nu]\rho}u_{\beta\lambda;\rho}u_{\alpha\delta;\gamma} + 2\tilde{\nu}^{(\gamma,6)} \, u^{\alpha\beta}\Pi^{\delta\lambda}\Pi^{[\mu|\gamma}\Omega^{|\nu]\rho}u_{\alpha\delta;\beta}u_{\gamma\rho;\lambda} \quad \text{(D.9b)}$$

$$+ 2\tilde{\nu}^{(\beta,1)} \, H_{\beta\gamma\lambda}u^{\alpha\beta}\Pi^{\delta\lambda}\Pi^{[\mu|\gamma}\Omega^{|\nu]\rho}u_{\delta\rho;\alpha} + 2\tilde{\nu}^{(\beta,2)} \, \Pi^{\beta\gamma}\Pi^{[\mu|\alpha}\Omega^{|\nu]\delta}H_{\alpha\beta\delta;\gamma}$$

$$+ 2\tilde{\nu}^{(\beta,3)} \, H_{\beta\lambda\rho}u^{\alpha\beta}\Pi^{\delta\lambda}\Pi^{[\mu|\gamma}\Omega^{|\nu]\rho}u_{\alpha\gamma;\delta} + 2\tilde{\nu}^{(\beta,4)} \, H_{\beta\gamma\rho}u^{\alpha\beta}\Pi^{\delta\lambda}\Pi^{[\mu|\gamma}\Omega^{|\nu]\rho}u_{\alpha\delta;\lambda}$$

$$+ 2\tilde{\nu}^{(\beta,5)} \, H_{\alpha\beta\lambda}u^{\alpha\beta}\Pi^{\delta\lambda}\Pi^{[\mu|\gamma}\Omega^{|\nu]\rho}u_{\delta\rho;\gamma} + 2\tilde{\nu}^{(\beta,6)} \, H_{\beta\gamma\lambda}u^{\alpha\beta}\Pi^{\delta\lambda}\Pi^{[\mu|\gamma}\Omega^{|\nu]\rho}u_{\alpha\delta;\rho}$$

$$+ 2\tilde{\nu}^{(\alpha)} \, H_{\alpha\gamma\delta}H_{\beta\lambda\rho}u^{\alpha\beta}\Pi^{\delta\lambda}\Pi^{[\mu|\gamma}\Omega^{|\nu]\rho} + \mathcal{O}(\partial^3) \,.$$

The transport coefficients of the $SO(2)$-components are related to those in the effective action via

$$\tilde{\eta}^{(\gamma,1)}_{(2)} = \frac{1}{2}\beta_2(\mu) \,, \qquad\qquad \tilde{\eta}^{(\gamma,2)}_{(2)} = -\frac{1}{2}\beta_1(\mu) \,, \qquad\qquad \text{(D.10a)}$$

$$\tilde{\eta}^{(\gamma,3)}_{(2)} = \frac{1}{2}\beta_2(\mu) - \frac{1}{2}\beta_1(\mu) \,, \qquad\qquad \tilde{\eta}^{(\gamma,4)}_{(2)} = -\frac{1}{2}\beta_2(\mu) \,, \qquad\qquad \text{(D.10b)}$$

$$\tilde{\eta}^{(\gamma,5)}_{(2)} = \frac{\rho(\mu)\beta'_2(\mu)}{2\rho'(\mu)} - \frac{1}{2}\beta_2(\mu) \,, \qquad\qquad \text{(D.10c)}$$

$$\tilde{\eta}^{(\beta,1)}_{(2)} = -\alpha(\mu) \,, \qquad\qquad \tilde{\eta}^{(\beta,2)}_{(2)} = \alpha(\mu) \,, \qquad\qquad \text{(D.10d)}$$

$$\tilde{\eta}^{(\beta,3)}_{(2)} = \alpha(\mu) - \frac{\rho(\mu)\alpha'(\mu)}{\rho'(\mu)} \,, \qquad\qquad \text{(D.10e)}$$

and similarly for the $(v \otimes v)$-components:

$$\tilde{\nu}_0 = -\frac{1}{2}\beta_2(\mu) \,, \qquad\qquad \tilde{\nu}_1 = -\frac{1}{2}\beta_1(\mu) \,, \qquad\qquad \text{(D.11a)}$$

$$\tilde{\nu}_2 = \frac{1}{2}\beta_2(\mu) \,, \qquad\qquad \text{(D.11b)}$$

$$\tilde{\nu}^{(\gamma,1)} = -\frac{1}{2}\beta_1(\mu) \,, \qquad\qquad \tilde{\nu}^{(\gamma,2)} = -\frac{1}{2}\beta_2(\mu) + \frac{1}{2}\mu\beta'_2(\mu) \,, \qquad \text{(D.11c)}$$

$$\tilde{\nu}^{(\gamma,3)} = \frac{1}{2}\beta_2(\mu) - \frac{\rho(\mu)\beta'_1(\mu)}{2\rho'(\mu)} + \frac{1}{2}\beta_1(\mu) \,, \qquad\qquad \tilde{\nu}^{(\gamma,4)} = \frac{1}{2}\beta_2(\mu) \,, \qquad\qquad \text{(D.11d)}$$

$$\tilde{\nu}^{(\gamma,5)} = \frac{1}{2}\beta_2(\mu) \,, \qquad\qquad \tilde{\nu}^{(\gamma,6)} = -\frac{1}{2}\mu\beta'_2(\mu) \,, \qquad\qquad \text{(D.11e)}$$

$$\tilde{\nu}^{(\beta,1)} = -2\mu\alpha'(\mu) + 2\alpha(\mu) \,, \qquad\qquad \tilde{\nu}^{(\beta,2)} = 2\alpha(\mu) \,, \qquad\qquad \text{(D.11f)}$$

$$\tilde{\nu}^{(\beta,3)} = -\beta'_2(\mu) + 2\alpha(\mu) \,, \qquad\qquad \tilde{\nu}^{(\beta,4)} = -2\alpha(\mu) \,, \qquad\qquad \text{(D.11g)}$$

$$\tilde{\nu}^{(\beta,5)} = -\frac{1}{2}\beta'_2(\mu) \,, \qquad\qquad \tilde{\nu}^{(\beta,6)} = -2\mu\alpha'(\mu) \,, \qquad\qquad \text{(D.11h)}$$

$$\tilde{\nu}^{(\alpha)} = 4\alpha'(\mu) \,, \qquad\qquad \text{(D.11i)}$$

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
