# Peer review of "Classification of magnetohydrodynamic transport at strong magnetic field"

_SciPost Physics_

## Round 1 · Referee Report · Anonymous (Referee 2) · 2020-9-14

Report

This work is a technically impressive, relevant, timely and important addition to the field of magnetohydrodynamics, which was recently reformulated in the language of higher-form symmetries by Ref. [30]. It was only through this new machinery that it became clear how the zero temperature (or, in other words, the extremely strong magnetic field) limit of MHD could be understood and that this limit was connected to a previously widely studied theory called force-free electrodynamics.

What is interesting about this zero temperature limit is that a consistent truncation of MHD exists which is non-dissipative and, yet, it retains the structure of an infinite gradient expansion series with higher-and-higher derivatives. It is believed that a non-dissipative limit is impossible in standard neutral hydrodynamics, which makes MHD special.

Since the limit is non-dissipative, the leading-order corrections to ideal MHD come from second-order hydrodynamics (the MHD equivalent of the full BRSSS theory). A few of these terms were studied previously (in [30] and then in [42], ...) and their phenomenology discussed. However, this is the first paper that completely classifies all of them. In a sense, this work is equivalent to finding all possible “viscosities” in Navier-Stokes hydrodynamics, which are the leading-order corrections to ideal hydrodynamics. The number in the Navier-Stokes theory is 2, here it is 11. Whatever the full second-order MHD theory is, it should reduce to the results of this paper in the limit of extremely strong magnetic field. Moreover, this paper teaches us how force-free electrodynamics needs to be corrected away from the ideal limit at zero temperature.

One issue with papers that are of this nature, that are this technical and delve into the complexity of classifying higher-order hydrodynamics, is that it is pretty much impossible to know whether the classification presented here is completely correct. It is even harder to determine whether it is optimal (it contains the smallest possible number of terms). This is not a criticism of this paper, only a warning to the community that such classifications require independent verification.

The paper would be even stronger if the authors managed to find a direct new physical consequence of their second-order terms. Naturally, this is very difficult and any such success easily warrants a separate publication.

Taking all this into account, I believe that the paper is certainly of high enough quality, novelty and relevance to warrant publication in SciPost.

I do not think that any major changes are necessary. The list of smaller suggestions follows.

1- At the end of section 3, I suggest that the authors add a complete list of all transport coefficients and show how they can be computed out of two- and three-point function Kubo formulae; i.e., a list of all transport coefficients expressed purely in terms of hydrodynamic n-point functions.

2- I suggest a thorough read through the text to eliminate typos and grammatical errors.

3- Above eq. 2.11a, the authors talk about the conservation law 2.4. I would not call 2.4 a conservation law. Those are diffeomorphisms and gauge transformations.

4- Below eq. 2.30, it should say “break down” not “breakdown”.

5- Regarding these additional gapped modes, the authors say that they cannot be removed by a frame choice (above eq. 2.37). Below eq. 3.9, they then say that it is also possible that this mode can be removed upon the field redefinition. Can it be removed or not? I have found the discussion of these extra modes a bit confusing in places and it would be nice if this could be clarified, possibly with a summary in a single location in the text (maybe in the discussion). Some interesting sounding comments are also added at the beginning of page 19, but the full meaning of that paragraph and the physical significance of such modes in the cited references in not completely clear.

---

## Round 1 · Referee Report · Anonymous (Referee 1) · 2020-9-14

Report

This work deals with formalising force free electrodynamics (FFE) as an effective field theory. In particular, this work extends and improves the work of Refs. [30] and [42] by classifying second order contributions to a proposed effective action and constitutive relations. I find the work very interesting but I think that the description of the state of affairs in this subject and some of the claims need some revision.

1 - FFE from MHD
It is claimed in different parts of the paper (including the abstract) that FFE was recently shown to be derived from MHD. The authors refer to a few references. I do not think this is accurate. For instance, Ref. [41] did not derive FFE from MHD but noted that by imposing additional symmetries in the ideal order MHD action, the equations of motion of FFE followed. This is not a derivation. Additionally, Ref. [30] did not derive it - the arguments are based on symmetry enhancement - in the sense that no formal limit from finite temperature MHD is taken such that FFE is recovered. I believe that these parts of the paper should be clearly rewritten as to better reflect the state of affairs. What seems to be the case is that there is a proposal for the symmetries of FFE from an effective action point of view and those symmetries are assumed to hold in a derivative expansion to higher orders.

2 - The strong magnetic field limit
Related to the previous point, the authors claim that their work can be understood as the B^2/T->Infinity limit of MHD. I cannot understand how this can be claimed. Have the authors, or anyone else for that matter, start from MHD and taken such formal limit in powers of (B^2/T) ? This requires taking MHD and expanding it in powers of (B^2/T) which I believe is a quite nontrivial exercise that has not yet been done. If this is the case, I think the authors should change their claims and in fact pose it as an open problem.

3 - Non-dissipative FFE
Related to the previous point, the assumptions on the symmetries of FFE render the theory non-dissipative even at second order. I find it hard to believe that if I would take MHD and expand it powers of (B^2/T) I would not get a dissipative term at first order in derivatives, let alone second order. Can the authors explain where in their approximation shall we start saying dissipative effects of MHD? And could they estimate if dissipative effects can actually be leading compared to their symmetry assumptions?

4 - A caveat in the counting of transport
It appears to me that there is a caveat in the counting of transport coefficients in the effective action and that this caveat is also present in Ref. [42]. As the authors mention in their section 4, when looking at magnetosphere solutions, one is interested in dipole solutions (as monopole solutions are not physical). What seems to be forgotten is that dipole solutions are constructed by gluing two monopole solutions across the equator. What this means is that, in fact, the relevant solutions are solutions of FFE with an interface. Already at ideal order this requires Israel-type matching conditions across the interface. The analysis done by the authors, in particular in appendix B, ignores this and performs partial integrations assuming the non-existence of such boundaries/interfaces. I wonder if within the symmetry assumptions of the authors there would be a non-trivial first order term if boundaries were considered. From a quick search on arXiv I found that arXiv:1803.00991 (which appears to nevertheless be missing in the references of the present manuscript as it deals with the same reformulation of MHD in terms of higher-form symmetries) contains a preliminary study of MHD with interfaces but I am not sure whether it can answer my question. Regardless, I do not think that the authors need to redo their work but I think it's an issue that should be mentioned, and which has potential implications.

5 - Is the U(1)-1 form symmetry spontaneously broken?
As far as I understand from the Goldstone theorem for higher-form symmetries, the photon is the Goldstone mode of spontaneously broken 1-form symmetry. MHD is a theory with a dynamical photon. Electric fields are screened in this theory so not all components of the field strength are relevant (which allows for a dual description in terms of magnetic field lines) but there are magnetic fields which of course require a photon nevertheless. Isn't the conservation of the two-form current mentioned in this paper that of a spontaneously broken one-form symmetry? I cannot find this mentioned explicitly anywhere.

6 - A trivial question
The authors are arguing that another microscopic length scale is required to make sense of transport coefficients and they link this to the renormalisation group scale that appears in holographic models. I do not fully understand why this additional scale is needed to have a consistent expansion in derivatives and I also don't understand why (4.9) would vanish otherwise. As I understand the transport coefficient \beta_2 is an arbitrary function of \mu which in physical terms is proportional to the modulus of the magnetic field. Why does \beta_2 have to be a constant if there was no extra microscopic scale?

If the authors can clarify these points, I would be happy to recommend the paper for publication.

---

## Editorial Decision

awaiting_resubmission